EMBO
Molecular Medicine

# Targeting PI3K/Akt/mTOR signaling in rodent models of PMP22 gene-dosage diseases

Doris Krauter [1,2,3,6], Daniela Stausberg[1,2], Timon J Hartmann[1,2], Stefan Volkmann [2], Theresa Kungl[4], David A Rasche[1,2,3], Gesine Saher[2], Robert Fledrich [4], Ruth M Stassart[5], Klaus-Armin Nave [2], Sandra Goebbels[2✉], David Ewers [1,2,3✉] & Michael W Sereda [1,2,3✉]

## Abstract

Haplo-insufficiency of the gene encoding the myelin protein PMP22 leads to focal myelin overgrowth in the peripheral nervous system and hereditary neuropathy with liability to pressure palsies (HNPP). Conversely, duplication of PMP22 causes Charcot-Marie-Tooth disease type 1A (CMT1A), characterized by hypomyelination of medium to large caliber axons. The molecular mechanisms of abnormal myelin growth regulation by PMP22 have remained obscure. Here, we show in rodent models of HNPP and CMT1A that the PI3K/Akt/mTOR-pathway inhibiting phosphatase PTEN is correlated in abundance with PMP22 in peripheral nerves, without evidence for direct protein interactions. Indeed, treating DRG neuron/Schwann cell co-cultures from HNPP mice with PI3K/Akt/mTOR pathway inhibitors reduced focal hypermyelination. When we treated HNPP mice in vivo with the mTOR inhibitor Rapamycin, motor functions were improved, compound muscle amplitudes were increased and pathological tomacula in sciatic nerves were reduced. In contrast, we found Schwann cell dedifferentiation in CMT1A uncoupled from PI3K/Akt/mTOR, leaving partial PTEN ablation insufficient for disease amelioration. For HNPP, the development of PI3K/Akt/mTOR pathway inhibitors may be considered as the first treatment option for pressure palsies.

**Keywords** Charcot–Marie–Tooth Neuropathies; Peripheral Myelin Protein of 22 kDa; Myelin; Schwann Cell; PI3K/Akt/mTOR Signaling
**Subject Categories** Genetics, Gene Therapy & Genetic Disease; Neuroscience

## Introduction

Schwann cells wrap myelin around peripheral nerve axons for fast neural transmission (Nave, 2010). Proper expression of the peripheral myelin protein of 22 kDa (PMP22), an integral constituent of the compact myelin sheath, is important for the development and function of peripheral nerve fibers. The *PMP22* gene is located on chromosome 17p11.2 and mutations as well as copy number variations are causative for a group of hereditary neuropathies affecting approximately 1 in 5000 humans (Li et al, 2013; Skre, 1974). Haplo-insufficiency of *PMP22* causes hereditary neuropathy with liability to pressure palsies (HNPP) (Chance et al, 1993). Clinically, patients suffer from sensory loss as well as palsies and paresthesia upon mechanical stress on the nerve (Attarian et al, 2020; Li et al, 2004; Mouton et al, 1999). A hallmark of HNPP is the formation of tomacula, extensive formation of myelin sheaths at cytoplasmic areas, such as paranodes and Schmidt-Lanterman incisures leading to deformed and constricted axons and subsequently mild demyelination (Adlkofer et al, 1997; Mandich et al, 1995; Yoshikawa and Dyck, 1991). Slowed nerve conduction velocity and conduction block can be observed at sites susceptible for compression while other regions are unaffected (Bai et al, 2010; Li et al, 2002).

Complementarily, a duplication on chromosome 17 results in the overexpression of PMP22 and causes Charcot–Marie–Tooth disease type 1A (CMT1A) (Lupsky, 1992; Lupski et al, 1991; Raeymaekers et al, 1991; Timmerman et al, 1990). Patients suffer from a distally pronounced, slowly progressive muscle weakness and sensory symptoms that are accompanied by slowed nerve conduction velocity (NCV) (Pareyson and Marchesi, 2009). Schwann cells from CMT1A patients and animal models show early defects in growth signaling and differentiation (Fledrich et al, 2019; Fledrich et al, 2014; Fornasari et al, 2018; Magyar et al, 1996; Massa et al, 2006) resulting in large caliber axons that are thinly or amyelinated (Sereda et al, 1996) along with reduced internodal length (Saporta et al, 2009) as well as small hypermyelinated axons, later onion bulb formation (Fledrich et al, 2019; Marie and Bertrand, 1918) and secondary axonal loss (Fledrich et al, 2012). While we were able to explain onion bulbs and hypermyelination of small axons by overstimulation of Schwann cells upon induced expression of the soluble growth factor neuregulin 1 type I, the functional impairments in CMT1A are only partly due to this autoparacrine mechanism and importantly, reduced myelin growth is independent of Schwann cell derived neuregulin 1 (Fledrich et al, 2019). Thus, although the genetics

[1]Research Group "Translational Neurogenetics", Max Planck Institute for Multidisciplinary Sciences, Göttingen, Germany. [2]Department of Neurogenetics, Max Planck Institute for Multidisciplinary Sciences, Göttingen, Germany. [3]Department of Neurology, University Medical Center Göttingen, Göttingen, Germany. [4]Institute of Anatomy, University of Leipzig, Leipzig, Germany. [5]Institute of Neuropathology, University of Leipzig, Leipzig, Germany. [6]Present address: Division of Molecular Neurobiology, Department of Medical Biochemistry and Biophysics, Karolinska Institutet, Stockholm, Sweden. ✉E-mail: sgoebbels@mpinat.mpg.de; david.ewers@med.uni-goettingen.de; mwsereda@med.uni-goettingen.de

of *PMP22* gene dosage diseases have been described more than 25 years ago, the molecular mechanisms causative for the abnormal myelination remain largely unknown and no small molecule or pharmacological therapeutic intervention exist.

Myelin growth is regulated by the PI3K/Akt/mTOR signaling pathway in the nervous system (Domenech-Estevez et al, 2016; Figlia et al, 2017; Norrmen et al, 2014; Ogata et al, 2004). As a central growth signaling hub (Kim and Guan, 2019), mTOR complex 1 (mTORC1) promotes expression of myelin genes via activation of downstream targets like eukaryotic translation initiation factor 4E-binding protein 1 (4E-BP1) and 70 kDa ribosomal S6 kinase (S6K), which initiates translation via activation of ribosomal protein S6 (S6) (Beirowski and Wong, 2017; Figlia et al, 2018). We have previously shown that *Pmp22* overexpression reduces PI3K/Akt/mTOR signaling in a CMT1A rat model, starting from early postnatal development (Fledrich et al, 2014). Here, we investigated whether PI3K/Akt/mTOR signaling provides therefore a therapeutic target to treat the consequences of altered Pmp22 gene-dosage early on in development. We found a striking correlation between PMP22 dosage and the steady-state level of the major inhibitor of the PI3K/Akt/mTOR pathway, the enzyme *phosphatase and tensin homolog* (PTEN). Indeed, we found that HNPP, as modeled in haplo-insufficient *Pmp22* mice (*Pmp22*[+/-]), can be clinically alleviated by pharmacologically targeting mTOR signaling. Our results thus identify a pharmacological target for this inherited neuropathy. In contrast, the CMT1A disease course as modeled by *PMP22*[tg] mice was unaltered upon partial ablation of PTEN without an effect on Schwann cell differentiation.

# Results

## PMP22 dosage perturbs the abundance of the growth signaling inhibitor PTEN in animal models of CMT1A and HNPP

The major inhibitor of the PI3K/Akt/mTOR pathway is PTEN, which counteracts PI3K by dephosphorylating phosphatidyinosityl (3,4,5)-trisphosphate (PIP3) to phosphatidyinosityl (4,5)-bisphosphate (PIP2), resulting in a decreased activity of Akt (Maehama and Dixon, 1998; Stambolic et al, 1998). Depletion of *Pten* in Schwann cells thus leads to an overactivation of the PI3K/Akt/mTOR pathway and focal hypermyelination of axons (Figlia et al, 2017; Goebbels et al, 2012), a phenotype similar to the tomacula in nerves from HNPP patients. When we tested by Western Blot analysis the abundance of PTEN in peripheral nerve lysates of HNPP mice, with 50% reduced expression of PMP22, we also found the levels of PTEN consistently reduced throughout development (Figs. 1A,B and EV1A). Moreover, an increase in ribosomal protein S6 phosphorylation indicates overactivation of mTOR already at postnatal day 6 of HNPP mice, when compared to wildtype controls (Fig. 1b). In contrast, in lysates from *Pmp22* transgenic rats (*Pmp22*[tg]) which model CMT1A (Sereda et al, 1996), the overexpression of PMP22 correlated with an increased abundance of PTEN and a decrease of the 70 kDa ribosomal S6 kinase (S6K) phosphorylation downstream of mTOR at postnatal day 6 (Figs. 1C,D and EV1B). Interestingly, we observed a similar correlation of *Pmp22* and *Pten* at the mRNA level suggesting that *Pmp22* gene-dosage affects *Pten* transcription in Schwann cells

(Fig. 1E,F). In view of the direct correlation between PMP22 and PTEN expression levels in the mutants throughout development, we explored the possibility of an interaction between the proteins. By immunoprecipitation experiments we were unable to detect protein-protein interaction between PMP22 and PTEN (Fig. EV2A–C). Furthermore, we performed pull-down experiments using purified PMP22. While we were able to confirm interaction of PMP22 with P0 (D'Urso et al, 1999), we could not detect PTEN in PMP22 eluates (Fig. EV2D), making a direct, high-affinity interaction unlikely. Comparing purified myelin with peripheral nerve lysates, Western Blot analysis showed that PTEN is more abundant in the lysates (Fig. 1G). Indeed, immunohistochemical staining revealed widespread PTEN expression in axons, Schwann cell nuclei and in the non-compacted cytoplasmic myelin compartments including the bands of Cajal (Fig. 1H).

Taken together, *Pmp22* dosage directly correlates with the abundance of PTEN and correlates inversely with the activation level of the PI3K/Akt/mTOR pathway (Fig. 1B,D) (Bolino et al, 2016; Fledrich et al, 2014). The observed alterations coincide with the occurrence of tomacula shortly after the onset of myelination in HNPP (Fig. EV1C,D) as well as early dysmyelination in CMT1A (Fledrich et al, 2014). We therefore hypothesized that in peripheral nerves of *Pmp22*[+/-] mice, which model HNPP, abnormal myelin growth is caused by the enhanced PI3K/Akt/mTOR signaling due to a partial loss of PTEN. Conversely, we hypothesized that reduced PI3K/Akt/mTOR activity due to increased PTEN levels leads to dysmyelination in *Pmp22*[tg] CMT1A nerves. Importantly, our model (Fig. 1I) rendered the PI3K/Akt/mTOR pathway as an interesting target for an experimental therapy in models of HNPP and CMT1A.

## Targeting PI3K/Akt/mTOR signaling ameliorates myelin outfoldings

To investigate the role of PI3K/Akt/mTOR signaling in HNPP myelin pathology, we first studied co-cultures of Schwann cells and dorsal root ganglia neurons (SC-DRG), derived from *Pmp22*[+/-] mice, which were either treated with LY294002, a PI3K inhibitor, or with Rapamycin, an inhibitor of mTOR (Fig. 2A). *Pmp22*[+/-] co-cultures showed more myelin outfoldings, which were particularly prominent at paranodal regions (Fig. 2B) and resembled tomacula, the histological hallmark of HNPP. When quantified two weeks after myelination was induced, we found 35% of myelinated segments with outfoldings in SC-DRG co-cultures from *Pmp22*[+/-] mice (Fig. 2C). Treatment with either one inhibitor for 2 weeks visibly decreased the number of segments with outfoldings by about 50% (Fig. 2B,C). Of note, the total number of myelinated segments was unaltered in mutant cultures (Fig. 2D). Thus, inhibiting the activated PI3K/Akt/mTOR pathway in *Pmp22*[+/-] Schwann cells significantly ameliorated aberrant myelin growth in vitro.

## Rapamycin treatment improves the phenotype of *Pmp22*[+/-] HNPP mice

These promising in-vitro findings prompted us to perform a therapeutic trial by treating *Pmp22*[+/-] mice in vivo. Between postnatal days 21 and 148, mice were given Rapamycin intraperitoneally (5 mg/kg), twice per week (Fig. 3A). As expected, Rapamycin treatment resulted in decreased phosphorylation of

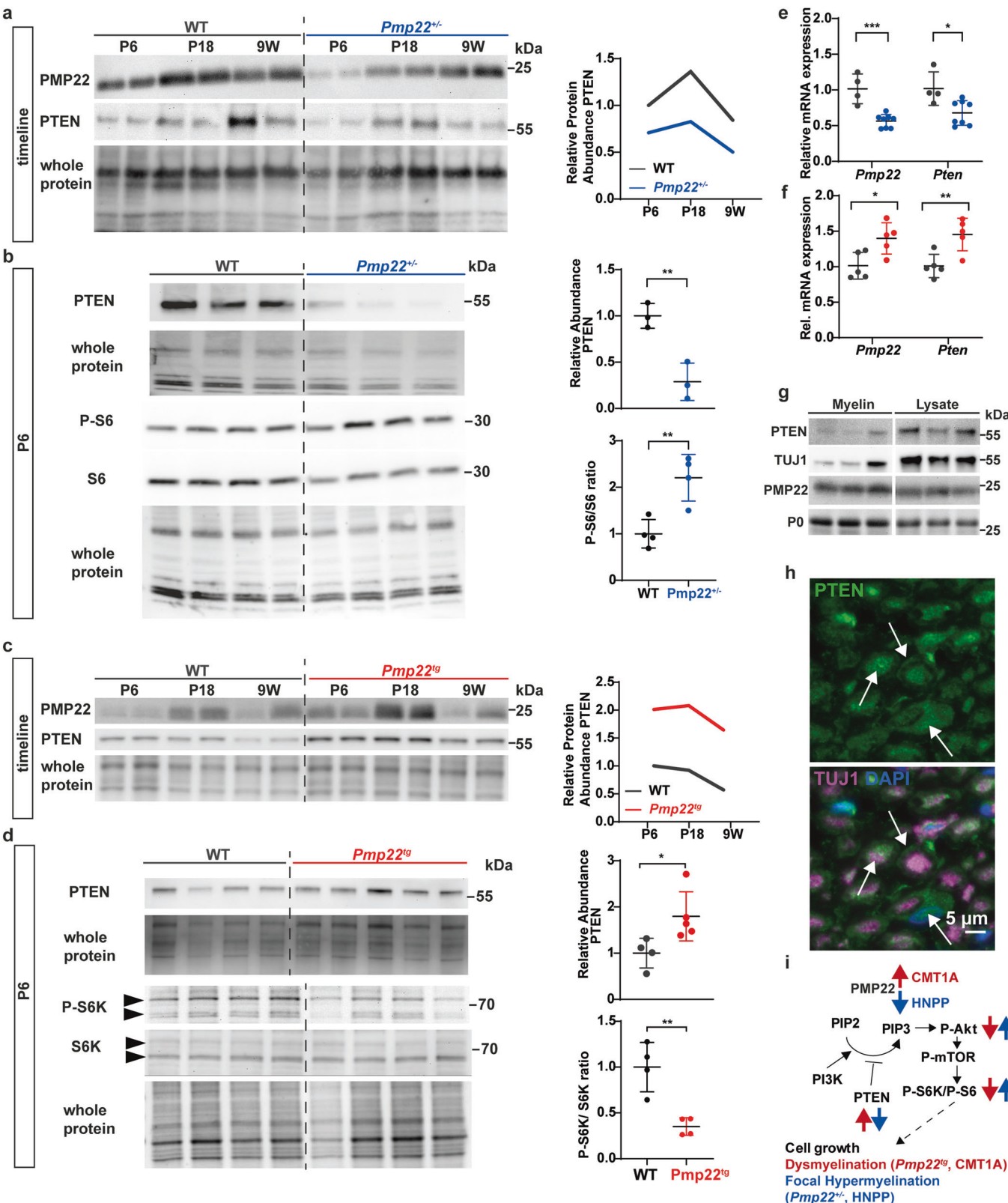

**Figure 1.  PTEN is *Pmp22* gene-dosage dependently altered in animal models of HNPP and CMT1A.**

(A) Western Blot analysis (left panel) showing PTEN and PMP22 protein levels in sciatic nerve lysates of *Pmp22*[+/-] mice at postnatal day 6 (P6), postnatal day 18 (P18) and 9 weeks of age compared to wildtype (WT) control (n = 2 per time point and group). Fast green whole protein staining was used as loading control for the quantification (right panel). (B) Western Blot analysis (left panel) showing a decrease of PTEN protein levels in sciatic nerve lysates of *Pmp22*[+/-] mice (n = 3) at postnatal day 6 and an increase in S6 phosphorylation compared to wildtype (WT) control (n = 4). Whole protein staining was used as loading control for the quantification (right panel). (C) Western Blot analysis (left panel) of PTEN and PMP22 protein levels in sciatic nerve lysates at P6, P18 and 9 weeks of age in *Pmp22*[tg] rats compared to WT control. Whole protein staining served as loading control for the quantification (right panel). (D) Western Blot analysis (left panel) showing an increase of PTEN protein levels in sciatic nerve lysates of *Pmp22*[tg] rats at postnatal day 6 and a decrease in S6K phosphorylation compared to wildtype (WT) control (n = 4). Whole protein staining was used as loading control for the quantification (right panel). (E) Quantitative RT-PCR analysis in tibial nerves from n = 4 WT and n = 7 *PMP22*[+/-] mice shows decreased mRNA levels of *Pmp22* and *Pten* in *Pmp22*[+/-] mice at P18. *Rplp0* and *Ppia* served as housekeeping genes. (F) Quantitative RT-PCR analysis in tibial nerves from n = 5 WT and n = 5 *PMP22*[tg] rats shows increased mRNA levels of *Pmp22* and *Pten* in *Pmp22*[tg] rats at P18. *Rplp0* and *Ppia* served as housekeeping genes. (G) Immunoblot of WT P18 rat whole sciatic nerve lysate and purified myelin. PTEN and TUJ1 are enriched in the lysate while PMP22 and P0 are enriched in the myelin fraction. (H) Femoral nerve cross section of 9-week-old WT rats shows PTEN (green) localization to the axon (magenta, TUJ1), nuclei (blue DAPI) and bands of Cajal (indicated by arrows). Scale bar is 5 µm. (I) Graphical overview of *Pmp22* gene-dosage dependent alterations in the PI3K/Akt/mTOR signaling pathway in animal models of CMT1A and HNPP. PMP22 overexpression leads to increased PTEN protein levels, reduced activation of the downstream PI3K/Akt/mTOR growth signaling pathway and subsequently demyelination (red). In contrast, PMP22 heterozygosity results in decreased PTEN levels, increased activation of the PI3K/Akt/mTOR signaling cascade and hypermyelination (blue). Data information: Mean numbers are displayed ±standard deviation. Statistical analysis was performed using Student's *t* test, *$p < 0.05$,**$p < 0.01$, ***$p < 0.001$. Source data are available online for this figure.

ribosomal protein S6 downstream of mTOR (Fig. 3B). Surprisingly, in nerve lysates from *Pmp22*[+/-] mice that had been treated with Rapamycin also the abundance of PTEN was increased (Fig. 3B). Since mTOR acts downstream of PTEN, this indicates that PTEN itself is negatively controlled by mTOR. This interpretation is in agreement with the observation by others that PI3K signaling upregulates PTEN expression at the translational level, presumably as a regulatory feedback in this signaling pathway (Mukherjee et al, 2021). Histologically, sciatic nerves from *Pmp22*[+/-] mice displayed the disease-typical formation of tomacula, i.e., areas of focal hypermyelination, recurrent loops, and myelin infoldings, already starting at an early stage of myelination (Figs. 3C and EV1C). Quantification of axons showing these in nerve cross sections from Rapamycin treated animals revealed a significant decrease in the pathological hallmarks of HNPP (Fig. 3C–E), while the total axon number was unaltered (Fig. 3F). Importantly, Rapamycin treatment significantly improved the grip strength of *Pmp22*[+/-] mice, although they did not reach wildtype levels (Fig. 3G). As expected, Rapamycin reduced the weight gain of all animals during development, without impact on the grip strength in wildtype mice (Fig. 3G,H). Electrophysiological measurements revealed lower compound muscle action potential (CMAP) amplitudes and lower nerve conduction velocities (NCV) in placebo treated *Pmp22*[+/-] mice. In contrast, Rapamycin treated animals displayed increased amplitudes reflecting improved muscle innervation, whereas conduction velocities remained decreased (Fig. 3I,J).

Taken together, reducing the abnormal activity of the PI3K/Akt/mTOR pathway observed in HNPP Schwann cells improves the histological and clinical phenotype of the disease model.

## Inhibiting PTEN improves myelination in SC-DRG co-culture model of CMT1A

*Pmp22* expression levels correlate with the abundance of PTEN in peripheral nerves (Figs. 1A–D and EV1A,B). *Pmp22* overexpression in CMT1A causes neuropathy by a partial loss of PI3K/Akt/mTOR signaling in Schwann cells, which should be ameliorated by the pharmacologic inhibition of PTEN. To test this hypothesis, we generated *Pmp22*[tg] CMT1A SC-DRG co-cultures and increased the signaling strength of the pathway by inhibiting PTEN. To this end we applied a small vanadium complex (VO-OHpic) to the medium,

a highly potent and specific inhibitor of PTEN that increases PtdIns(3,4,5)P3 and Akt phosphorylation (Rosivatz et al, 2006; Fig. 4A). Similar to the in-vivo situation in patients and CMT1A animal models (Fledrich et al, 2018), SC-DRG co-cultures from *Pmp22*[tg] mice exhibited fewer myelinated axon segments (Fig. 4B,C). However, this number was significantly increased in the presence of 500 nM VO-OHpic as compared to DMSO treated controls (Fig. 4B,C). Interestingly, a tenfold higher inhibition of PTEN with 5 µM VO-OHpic diminished myelination, independent of the *Pmp22* genotype (Figs. 4C and EV3). Similarly, the prolonged inhibition of PTEN with VO-OHpic (for 14 days) caused a dosage-dependent reduction in myelinated segments in wildtype co-cultures (Figs. 4C and EV3). As this may indicate toxicity of VO-OHpic, and as we also cannot rule out a negative effect of PTEN inhibition on DRG neurons, these results have to be taken with care. Nevertheless, they revealed that the CMT1A myelination defect can in principle be ameliorated by reducing the activity of PTEN.

## Targeting *Pten* in Schwann cells to correct myelination in models of CMT1A

To reduce *Pten* function specifically in CMT1A Schwann cells in vivo, we applied a genetic approach (Fig. 5A). As the genetic tools to specifically target Schwann cells were only available in the mouse and not the rat, we used the C61 mouse model of CMT1A (Huxley et al, 1998). We reduced PTEN by about 50% selectively in CMT1A Schwann cells by crossbreeding *Pmp22*[tg] mice with floxed *Pten* and *Dhh-cre* mice, yielding *PTEN*[fl/+]*Dhh*[cre/+]*PMP22*[tg] experimental mutants (Fig. 5B). Western Blot analyses of sciatic nerve lysates confirmed the increase of PTEN in *PMP22*[tg] mice and the reduction of PTEN in the double mutants (Fig. 5C,D). As predicted, this manipulation caused activation of the downstream targets of Akt and mTOR, S6K and ribosomal protein S6, as visualized by protein abundance using Western Blot analysis (Fig. 5C,D) and in immunostained peripheral nerve sections (Fig. 5E). Similar to the VO-OHpic treated cultures, Schwann cell-DRG co-cultures derived from *PTEN*[fl/+]*Dhh*[cre/+]*Pmp22*[tg] mutants displayed an increased number of myelinated segments when compared to co-cultures from *Pmp22*[tg] single mutants (Fig. 5F,G). In contrast, co-cultures from *PTEN*[fl/+]*Dhh*[cre/+] mice showed no alteration in the number of

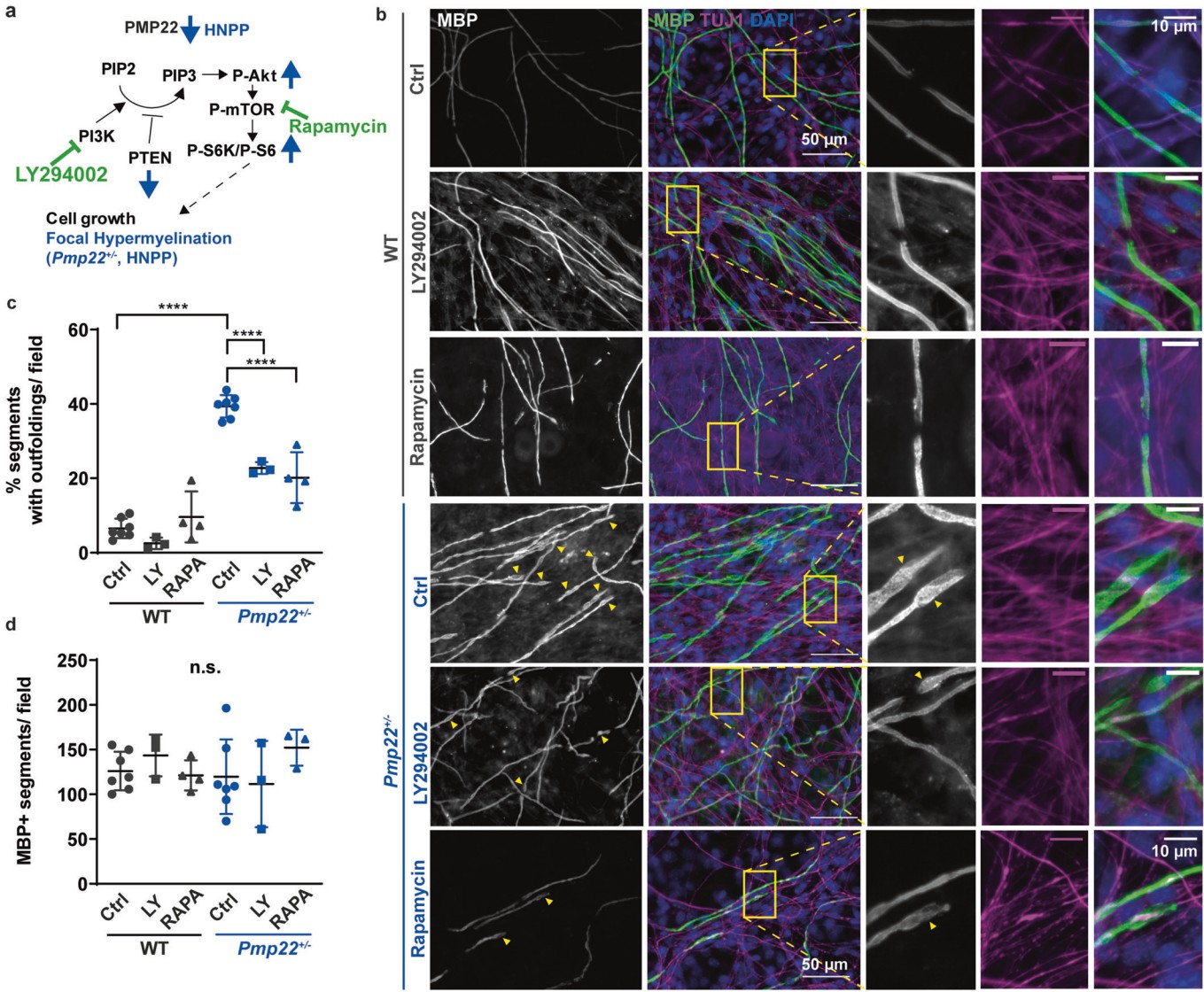

**Figure 2. Inhibition of the PI3K/Akt/mTOR signaling pathway reduces myelin outfoldings in *Pmp22*$^{+/-}$ co-cultures.**

(A) PI3K inhibitor LY294002 and mTOR inhibitor Rapamycin were used to counteract the upregulated PI3K/Akt/mTOR signaling pathway in *Pmp22*$^{+/-}$ mice. (B) Example images displaying Schwann cell-dorsal root ganglia neuron co-cultures from wildtype (WT) and *Pmp22*$^{+/-}$ (HNPP) mice 14 days after induction of myelination. Cells were treated with either DMSO as controls, 10 μM LY294002 or 20 nM Rapamycin. Fixed cells were stained for myelin basic protein (MBP, gray/green) to visualize myelin and beta tubulin III (TUJ1, magenta) for axons. Cell nuclei are stained by DAPI in blue. Yellow arrowheads indicate myelin outfoldings. Scale bar is 50 μm (overview) and 10 μm (blow-up). (C) Quantification of (B) reveals an increased proportion of myelinated segments with outfoldings in *Pmp22*$^{+/-}$ control co-cultures (circles) and a reduction after inhibition of PI3K (squares) or mTOR (triangles). $n = 3-7$ animals with $n = 5$ fields of view (500 × 500 μm) were quantified. (D) The mean number of myelinated segments is unaltered in control and treated *Pmp22*$^{+/-}$ co-cultures. $n = 3-7$ animals with $n = 5$ fields of view (500 × 500 μm) were quantified. Data information: Mean numbers are displayed ±standard deviation. Groups were compared using one-way ANOVA with Sidak's multiple comparison test (****$p \leq 0.0001$). Source data are available online for this figure.

myelinated segments (Fig. 5F,G). Collectively, these data demonstrate that targeting PTEN in PMP22 overexpressing Schwann cells improves myelination.

We then investigated whether PTEN reduction alleviates the phenotype of *Pmp22*$^{tg}$ mice in vivo. *Pmp22*$^{tg}$ CMT1A mice had fewer myelinated axons and smaller axonal diameters when compared to their wildtype littermates at age P18 and 16 weeks (Figs. 6A,B and EV4B,C) (Huxley et al, 1998). Moreover, *g*-ratio analysis revealed hypermyelinated small caliber axons and hypomyelinated large caliber axons in *Pmp22*$^{tg}$ mice (Fig. 6C,E).

While the mere heterozygosity of *Pten* in Schwann cells did not alter the number of myelinated axons, g-ratios or axon diameter, in CMT1A mice the same *Pten* heterozygosity increased the number of myelinated axons at postnatal day 18 (Figs. 6A,B and EV4A,B). However, this normalization of myelin was not maintained in the adult (Fig. 6A,B). Similarly, we noted increased myelin thickness in *PTEN*$^{fl/+}$*Dhh*$^{cre/+}$*Pmp22*$^{tg}$ mice early in development (Fig. 6C,D) but not at 16 weeks of age (Fig. 6E,F). In adult CMT1A mice, also the behavioral and electrophysiological phenotype was unaltered by *Pten* heterozygosity (Figs. 6G,H and EV4D–F). Interestingly, teased

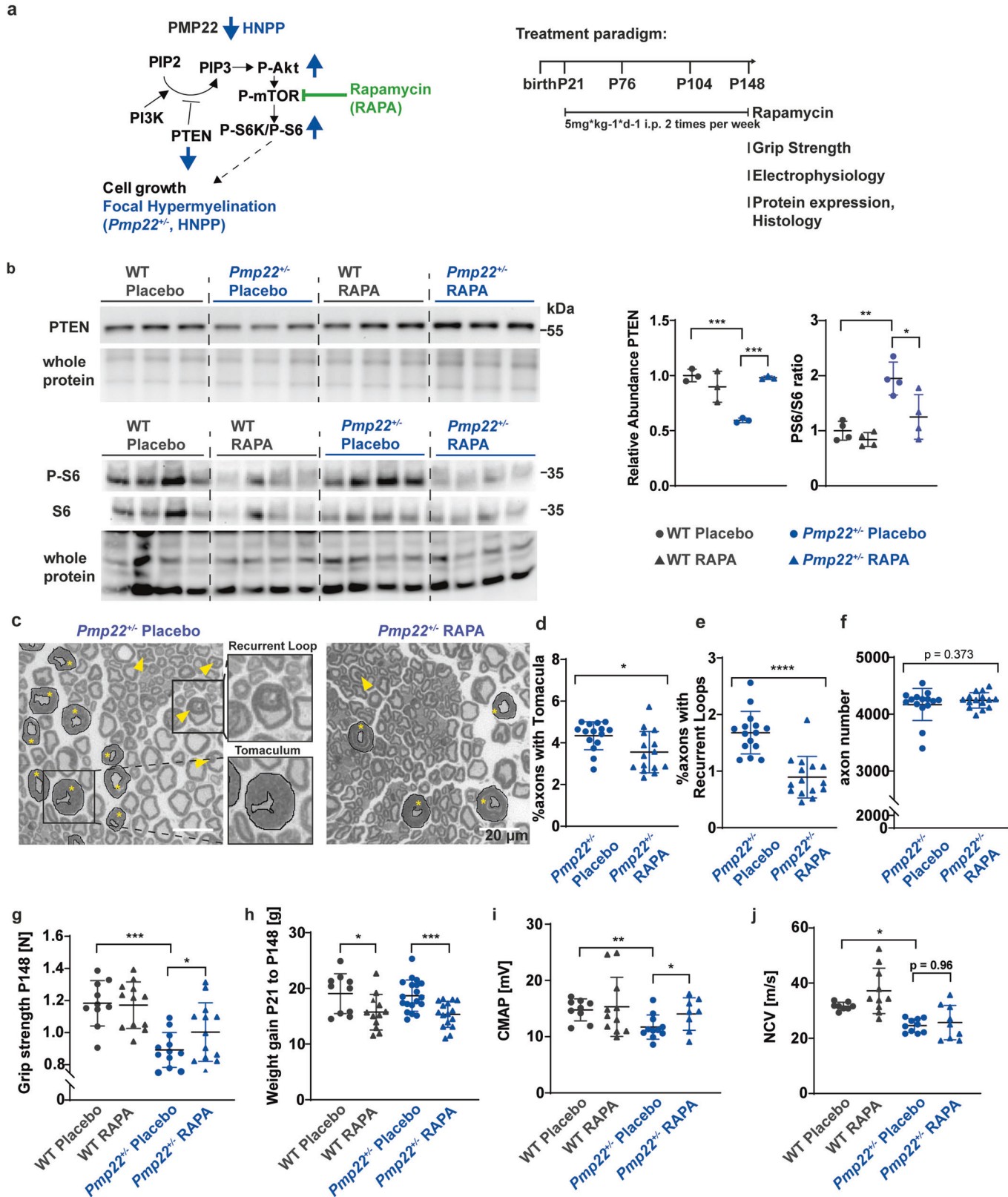

◄

**Figure 3. Rapamycin treatment in *Pmp22*⁺/⁻ mice ameliorates the disease phenotype.**

(A) *Pmp22*⁺/⁻ and wildtype (WT) control mice were injected i.p. with placebo solution or 5 mg Rapamycin per kg bodyweight two times per week from P21 until P148 to reduce mTOR activity. Grip strength analysis, electrophysiology, protein expression analysis and histology were performed at P148. (B) Western Blot analysis of PTEN protein ($n = 3$ animals per group) and phosphorylated and total S6 ($n = 4$ animals per group) (left panel) in whole sciatic nerve lysates. Quantification using whole protein staining as loading control shows increased PTEN protein levels and decreased S6 phosphorylation after Rapamycin treatment in *Pmp22*⁺/⁻ mice (right panel). (C) Sciatic nerve semi-thin sections of *Pmp22*⁺/⁻ placebo and Rapamycin treated mice. Tomacula are encircled in black and marked with asterisks. Yellow arrowheads point to recurrent loops. Scale bar is 20 μm. (D) The percentage of axons showing tomacula is decreased in whole sciatic nerves of Rapamycin treated *Pmp22*⁺/⁻ mice (triangle) compared to placebo controls (circles) at P148, $n = 15$ animals per group. (E) The percentage of axons showing recurrent loops is decreased in whole sciatic nerves of Rapamycin treated *Pmp22*⁺/⁻ mice (triangle) compared to placebo controls (circles) at P148, $n = 15$ animals per group. (F) Total axon number in sciatic nerves is unaltered between Rapamycin treated *Pmp22*⁺/⁻ mice (triangle) and placebo controls (circles) at P148, $n = 15$ animals per group. (G) Forelimb grip strength is decreased in *Pmp22*⁺/⁻ placebo mice ($n = 12$, blue circles) compared to WT placebo mice ($n = 10$, gray circles), whereas Rapamycin treatment improves strength in *Pmp22*⁺/⁻ mice ($n = 13$, blue triangles) and does not affect WT mice ($n = 12$, gray triangles) at P148. (H) Rapamycin-treated *Pmp22*⁺/⁻ and WT animals (triangles) gained less weight than placebo controls (circles). Groups of $n = 10$ WT placebo, $n = 12$ WT Rapamycin, $n = 19$ *Pmp22*⁺/⁻ placebo and $n = 16$ *Pmp22*⁺/⁻ Rapamycin. (I) Electrophysiological analysis shows reduced compound muscle action potential amplitudes (CMAP) in *Pmp22*⁺/⁻ placebo mice and increased CMAP after Rapamycin treatment. $n = 8$ WT placebo, $n = 10$ WT Rapamycin, $n = 10$ *Pmp22*⁺/⁻ placebo and $n = 8$ *Pmp22*⁺/⁻ Rapamycin. (J) Electrophysiological analysis shows reduced nerve conduction velocities (NCV) in *Pmp22*⁺/⁻ placebo mice and no alterations after Rapamycin treatment. $n = 8$ WT placebo, $n = 10$ WT Rapamycin, $n = 10$ *Pmp22*⁺/⁻ placebo and $n = 8$ *Pmp22*⁺/⁻ Rapamycin. Data information: Mean numbers are displayed ±standard deviation. Statistical analysis was performed using Student's $t$ test (D–F) and one-way ANOVA with Sidak's multiple comparison test (B, G–J); *$p \leq 0.05$ **$p \leq 0.01$, ***$p \leq 0.001$, ****$p \leq 0.0001$). Source data are available online for this figure.

fiber preparations of both *Pmp22*ᵗᵍ and *PTEN*ᶠˡ/⁺*Dhh*ᶜʳᵉ/⁺*Pmp22*ᵗᵍ mice at postnatal day 18 showed the same phenotype of shortened internodes (Fig. EV4G,H). Thus, the activated PI3K/Akt/mTOR pathway appears to rescue radial but not longitudinal myelin growth transiently in postnatal development.

## Schwann cell differentiation defects in CMT1A are not rescued by PTEN reduction

CMT1A Schwann cells fail to differentiate normally, partly due to upregulation of the growth factor Neuregulin 1 type I (Fledrich et al, 2014). To test whether reduced PTEN function has an impact on Schwann cell differentiation in *Pmp22*ᵗᵍ CMT1A mice, we compared the expression levels of markers for Schwann cell differentiation (*Hmgcr*) and dedifferentiation (*Ngfr*, *Pou3f1* and *Sox2*). Interestingly, there was no difference in *Pmp22* over-expressing mice cross-bred to *Pten* heterozygotes (Fig. 7A). Importantly, these marker transcripts were never altered in nerves of *Pmp22*⁺/⁻ HNPP mice (Fig. 7B), in line with our previous observation of unaltered Neuregulin 1 type I expression (Fledrich et al, 2014). Also myelin sheath thickness (excluding the tomacula) in peripheral nerve axons of *Pmp22*⁺/⁻ mice was similar to wildtype controls at postnatal day 18, whereas *Pmp22*ᵗᵍ CMT1A nerves displayed a characteristic shift of hypermyelinated small axons and thinly myelinated larger axons at that age (Fig. 7C).

In our resulting working model, *Pmp22* heterozygosity causes HNPP via decreased PTEN levels, leading to a secondary increase of PI3K/Akt/mTOR signaling in Schwann cells. Tomacula formation was improved by inhibiting mTOR using Rapamycin. *Pmp22* overexpression in CMT1A causes a Schwann cell differentiation defect, increased PTEN expression and a reduction of PI3K/Akt/mTOR signaling. Targeting Pten delayed dysmyelination in CMT1A models but failed to overcome the differentiation defect of mutant Schwann cells.

## Discussion

*Pmp22* is a dosage sensitive gene which causes hereditary neuropathy with liability to pressure palsies in haplo-insufficiency and Charcot-Marie-Tooth disease type 1A when overexpressed. We

made the observation that *Pmp22* gene dosage not only determines the expression level of this myelin protein in the peripheral nervous system, but also affects the abundance of PTEN in Schwann cells. The phosphatase PTEN degrades the signaling lipid PIP3. Thus, in *Pmp22*ᵗᵍ (CMT1A) nerve fibers the increased abundance of PTEN most likely explains the previously described decrease of PI3K/Akt/mTOR signaling (Fledrich et al, 2014), underlying reduced myelin growth in CMT1A. Our proposed model of disease is further supported by the observations that ablating Akt or mTOR in Schwann cells leads to hypomyelination (Cotter et al, 2010; Norrmen et al, 2014; Sherman et al, 2012), whereas the transgenic overexpression of Akt in these cells, similar to targeting PTEN directly, causes hypermyelination (Domenech-Estevez et al, 2016; Figlia et al, 2017; Flores et al, 2008; Goebbels et al, 2012). Indeed, depleting PTEN in Schwann cells causes an HNPP-like phenotype with focal myelin hypergrowth (tomacula) (Goebbels et al, 2012). Strikingly, we could show that treatment of *Pmp22*⁺/⁻ (HNPP) mice with Rapamycin, a specific inhibitor of mTOR and clinically approved drug, ameliorated the disease phenotype on the behavioral, electrophysiological and histological level.

*PTEN* is a tumor suppressor gene that mediates growth arrest (Furnari et al, 1998), similar to PMP22's initially described function as a growth arrest specific gene cloned from fibroblasts (Manfioletti et al, 1990). In Schwann cells, PTEN interacts with discs large homolog 1 (Dlg1) at the adaxonal myelin membrane, where the downregulation of PI3K/Akt/mTOR signaling terminates myelination (Cotter et al, 2010). Interestingly, deletion of *Pten* in Schwann cells leads to a delayed onset of myelination early postnatally (Figlia et al, 2017). Similarly, in *Pmp22*⁺/⁻ mice the onset of myelination is delayed, whereas at later stages these mice exhibit the HNPP phenotype with tomacula formation (Adlkofer et al, 1995). Thus, PMP22 serves two functions that affect the timing of myelination. Early in development, PMP22 enables the timely onset of myelination, whereas in later development PMP22 acts as a "break" on myelin growth. We note that also mTOR plays such a dual role, as increased mTOR activity in early development suppresses myelination (following radial sorting), whereas at later stages it is the loss of mTOR activity that interferes with continued myelination (Figlia et al, 2017). Thus, in CMT1A patients, the overexpression of PMP22 causes initially hypermyelination, most strikingly of small caliber axons, but a hypomyelination at later

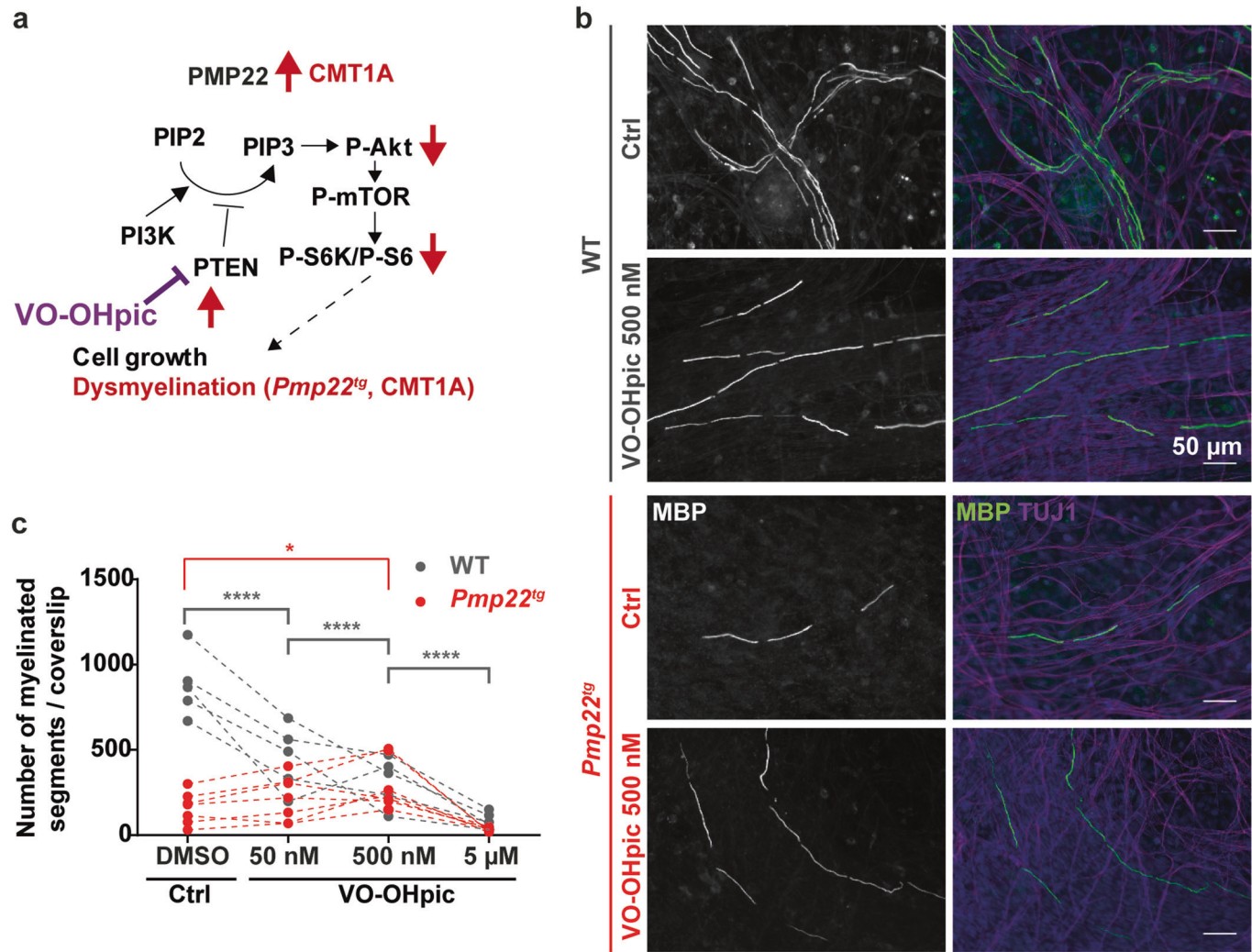

**Figure 4. Inhibition of PTEN improves myelination in *Pmp22^tg* co-cultures in vitro.**

(A) PTEN inhibitor VO-OHpic was used to disinhibit the PI3K/Akt/mTOR signaling pathway in CMT1A. (B) Example images of Schwann cell-dorsal root ganglia neuron co-cultures from wildtype (WT) and *Pmp22^tg* rats treated with DMSO as control (Ctrl) or PTEN inhibitor VO-OHpic (500 nM). Cells were stained for myelin basic protein (MBP) as a marker for myelinated segments (gray/ green) and TUJ1 for neurons (magenta) as well as DAPI for cell nuclei (blue). (C) Quantification of (B) shows a dose-dependent decrease of myelinated segments in WT co-cultures treated with DMSO and different concentrations of VO-OHpic (50 nM, 500 nM, 5 μM) and an increase of myelinated segments in *Pmp22^tg* co-cultures with 500 nM VO-OHpic (WT $n = 5$, CMT1A $n = 7$ animals). Groups were compared using two-way ANOVA with Sidak's multiple comparison test (*$p \leq 0.05$, ****$p \leq 0.0001$). Source data are available online for this figure.

stages. Interestingly, upon strong PMP22 overexpression, Schwann cells cease to myelinate axons already right after sorting, i.e. before myelination has begun, as observed in *Pmp22* transgenic rats bred to homozygosity (Niemann et al, 2000; Sereda et al, 1996).

The mechanisms that link the abundance of PMP22 to that of PTEN are still unclear and we here neither show direct nor indirect control of PTEN expression by PMP22. Immunoprecipitation of PMP22 and PTEN failed to detect protein-protein interactions between PMP22 and PTEN in tissue extracts or when over-expressed in cultured cells. Further, we have not been able to pull down PTEN from sciatic nerve lysate with purified PMP22. Although transient interaction cannot be ruled out, this makes it highly likely that PMP22 stabilizes PTEN indirectly, for example by transcriptional regulation, or on the protein level by sequestration

in membrane microdomains (lipid rafts) that are associated with PTEN interacting lipids, such as PIP2 (Kreis et al, 2014). PTEN could also increase in abundance by binding to scaffolding proteins in PMP22-dependent membrane microdomains (Lee et al, 2014).

The therapeutic effect of the mTOR (mTORC1) inhibitor Rapamycin in a mouse model of HNPP suggests that mTOR signaling in Schwann cells must be considered a pharmacological target in human HNPP for which no therapy is available (Pisciotta et al, 2021). Although generally considered a mild neuropathy, the clinical spectrum of HNPP patients covers more severe disease courses, including recurrent positional short-term sensory symptoms, progressive mononeuropathy, Charcot–Marie–Tooth disease-like polyneuropathy, chronic sensory polyneuropathy, chronic inflammatory demyelinating polyneuropathy-like, and recurrent subacute

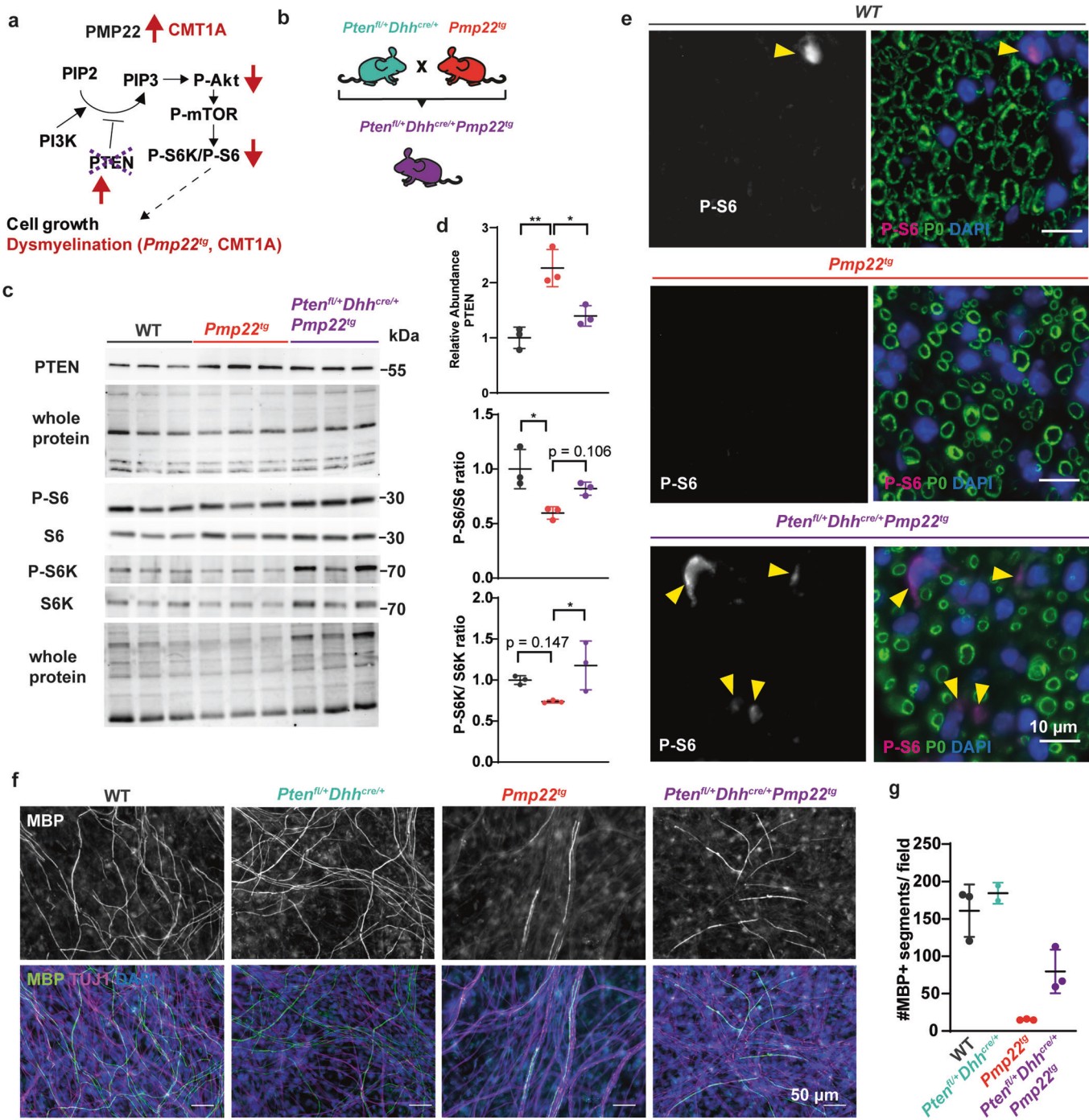

polyneuropathy (Mouton et al, 1999). The range of duration of symptoms can reach from 1 week to 50 years (Kumar et al, 2002). Besides null alleles, also heterozygous PMP22 mutations can lead to HNPP-like neuropathy (Shy et al, 2006) possibly due to reduced trafficking of the mutant protein to the Schwann cell plasma membrane (Stefanski et al, 2022). Currently it is unknown whether PI3K/Akt/mTOR signaling is increased also in those cases. We note that activation of mTOR is also observed in rodent models of other peripheral neuropathies displaying hypermyelination, e.g. recessive CMT4B caused by mutations in the phosphoinositide lipid

phosphatases myotubularin-related protein 2 (*MTMR2*) and 13 (*MTMR13*) genes (Azzedine et al, 2003; Bolino et al, 2000; Guerrero-Valero et al, 2021; Houlden et al, 2001) which are also targets of Rapamycin therapy (Bolino et al, 2016). The macrolid drug Rapamycin is mostly used as an immunosuppressive after organ transplantation and currently investigated in cancer therapy (Benjamin et al, 2011; Ma and Blenis, 2009). Systemic application was generally well tolerated in a phase 2b study in the majority of patients with inclusion body myositis, a rare non-lethal neuromuscular disease (Benveniste et al, 2021). We note that oral application of Rapamycin

◀ **Figure 5.　Genetic depletion of PTEN ameliorates mTOR activity in *Pmp22^tg* nerves and increases myelination in vitro.**

(A) Rationale to breed Schwann cell specific heterozygous *Pten* knockout mice (*Pten^fl/+Dhh^cre/+*) with CMT1A mice (*Pmp22^tg*) to lower Pten expression in CMT1A mice (*Pten^fl/+Dhh^cre/+Pmp22^tg*). (B) In order to reduce Pten genetically in Schwann cells, heterozygous Pten floxed mice under the Dhh^cre driver (*Pten^fl/+Dhh^cre/+*) were crossbred with *Pmp22^tg* mice. (C) Western Blot analysis of sciatic nerve lysate from WT, *PMP22^tg* and *Pten^fl/+Dhh^cre/+Pmp22^tg* at postnatal day 18 against PTEN, P-S6, S6, P-S6K and S6K with whole protein staining as the loading control. (D) Quantification of (C) reveals increased PTEN protein levels in *PMP22^tg* sciatic nerve lysates compared to WT controls as well as decreased PTEN protein levels in *Pten^fl/+Dhh^cre/+Pmp22^tg* sciatic nerve lysates compared to *Pmp22^tg* mice (upper panel) and downregulation of P-S6 (middle panel) activation in *PMP22^tg* mice compared to WT mice, while P-S6K (lower panel) activation is increased in *Pten^fl/+Dhh^cre/+Pmp22^tg* mice as compared to *PMP22^tg* mice. Groups were compared using one-way ANOVA with Sidak's multiple comparison test (*$p \leq 0.05$, **$p \leq 0.01$). (E) Paraffin cross sections of femoral nerves from 18 days old WT, *Pmp22^tg* and *Pten^fl/+Dhh^cre/+Pmp22^tg* mice show an increased signal for Phospho-S6 in double mutants (magenta, indicated by yellow arrowheads). Myelin is visualized by P0 (green) and nuclei by DAPI (blue). Scale bar is 10 µm. (F) Representative example images of Schwann cell dorsal root ganglia neuron co-cultures 14 days after induction of myelination. Myelin basic protein (MBP) indicates myelinated segments (gray/green), TUJ1 neurons (magenta) and DAPI nuclei (blue). Scale bar is 50 µm. (G) Quantification of (F) shows increased numbers of myelinated segments in *Pten^fl/+Dhh^cre/+Pmp22^tg* compared to *Pten^fl/+Dhh^cre/+* co-cultures. Shown are means of 5 fields of view (500 × 500 µm) for each animal (*n* = 2–3) ±standard deviation. Source data are available online for this figure.

has the potential to be repurposed for geroprotection after it was shown to prolong the life span in model organisms (Harrison et al, 2009; Kaeberlein et al, 2005; Kapahi et al, 2004; Vellai et al, 2003). Most recently, the adverse side effects of continuous Rapamycin treatment were circumvented in mice by a brief pulse of the drug given orally in early adulthood (Juricic et al, 2022). In HNPP, the local transdermal application of an mTOR inhibitor could be a preventive strategy and treatment of "pressure palsies", e.g. immediately after a suspected minor nerve injury.

Why is there this striking therapeutic effect of correcting PI3K/Akt/mTOR signaling in *Pmp22^+/-* mice but no such effect in *Pmp22^tg* mice that model CMT1A? One possible explanation is that in CMT1A a mere 50% reduction of PTEN (*PTEN^fl/+Pmp22^tgDhh^cre/+*) may not sufficiently activate mTOR. In contrast, even the complete loss of PTEN from PMP22 overexpressing CMT1A Schwann cells (*Pten^fl/flPmp22^tgDhh^cre/+*) leads at best to the formation of aberrant myelin profiles (Fig. EV5). We observed that activating Akt by PTEN reduction increased myelin sheath thickness transiently at postnatal day 18 in CMT1A mice. However, not only large caliber axons were affected but also the already hypermyelinated smaller sized axons. We previously observed that the persistent glial induction of Neuregulin 1 type I accounts for the hypermyelination of small caliber axons in CMT1A (Fledrich et al, 2019). Increased levels of glial Neuregulin 1 type I are observed after peripheral nerve injury and in models of dysmyelinating neuropathies such as CMT1A and CMT1B (Fledrich et al, 2019; Stassart et al, 2013). Here, we demonstrated increased expression of Neuregulin 1 type I and dedifferentiation markers in nerves of *Pmp22^tg* (CMT1A) mice but not *Pmp22^+/-* (HNPP) mice. Moreover, PTEN reduction in *Pmp22^tg* mice did not alter the expression of those markers. Thus, the Neuregulin 1 type I mediated differentiation defect is specific for the PMP22 overexpressing situation and independent of the PI3K/Akt/mTOR signaling pathway. This may explain the positive effects of counteracting the PI3K/Akt/mTOR pathway in *Pmp22^+/-* mice, whereas PTEN reduction slightly increased myelin growth early in development but did not overcome the differentiation defect in *Pmp22^tg* mice. Future research aiming at effective CMT1A therapy strategies may focus on a combination of targeting the differentiation defect and the dysregulated PI3K/Akt/mTOR pathway. In summary, our findings reveal that the PMP22 expression level affects the timing of myelination via the PTEN/PI3K/Akt/mTOR signaling axis. We demonstrated that counteracting the dysregulated signaling pathway is beneficial in HNPP mice and thereby may provide a novel therapy strategy for patients.

## Methods

### Transgenic rats and mice

The generation and genotyping of *Pmp22* transgenic mice (*Tg(PMP22)C61Clh*) (Huxley et al, 1998), *Pmp22* transgenic rats (*SD-Tg(Pmp22)Kan*) (Sereda et al, 1996), *Pmp22^+/-* mice (*Pmp22tm1Ueli*) (Adlkofer et al, 1995), *Pten-flox* mice (*Ptentm1HWu/J*) (Groszer et al, 2001) and the Dhh-Cre driver line (*FVB(Cg)-Tg(Dhh-cre)1Mejr/J*) (Jaegle et al, 2003) have previously been described. In short, for genotyping, genomic DNA was isolated from ear or tail biopsies by incubation in Gitschier buffer with TritonX-100 and proteinase K for a minimum of 2 h at 55 °C following heat inactivation of proteinase K at 90 °C for 10 min. Genotyping primers used for routine genotyping are listed in Table 1.

All animal experiments were conducted according to the Lower Saxony State regulations for animal experimentation in Germany as approved by the Niedersächsische Landesamt für Verbraucherschutz und Lebensmittelsicherheit (LAVES) and in compliance with the guidelines of the Max Planck Institute of Experimental Medicine or the Max Planck Institute for Multidisciplinary Sciences. Rats and mice were kept on a standard diet and drinking water ad libitum at 22 °C temperature, 55% humidity, with a light/dark cycle of 12 h/12 h per day. The health status of the animals was checked on a daily basis.

Inclusion and exclusion criteria were pre-established. Animals were randomly included into the experiment according to genotyping results, age and weight and independent of their sex. Animals were excluded prior to experiments in case of impaired health conditions or weight differences to the average group of >10%. During or after the experiment exclusion criteria comprise impaired health condition of single animals not attributed to genotype or experiment (according to veterinary) or weight loss >10% of the average group. No animals had to be excluded due to illness/weight loss >10% of the average group. Exclusion criteria regarding the outcome assessment were determined with an appropriate statistical test, the Grubbs' test (or ESD method) using the statistic software Graph Pad (Prism). Animal experiments (phenotype analysis, electrophysiology and histology) were conducted in a single blinded fashion towards the investigator. Selection of animal samples out of different experimental groups for molecular biology/ histology/ biochemistry was performed randomly and in a blinded fashion.

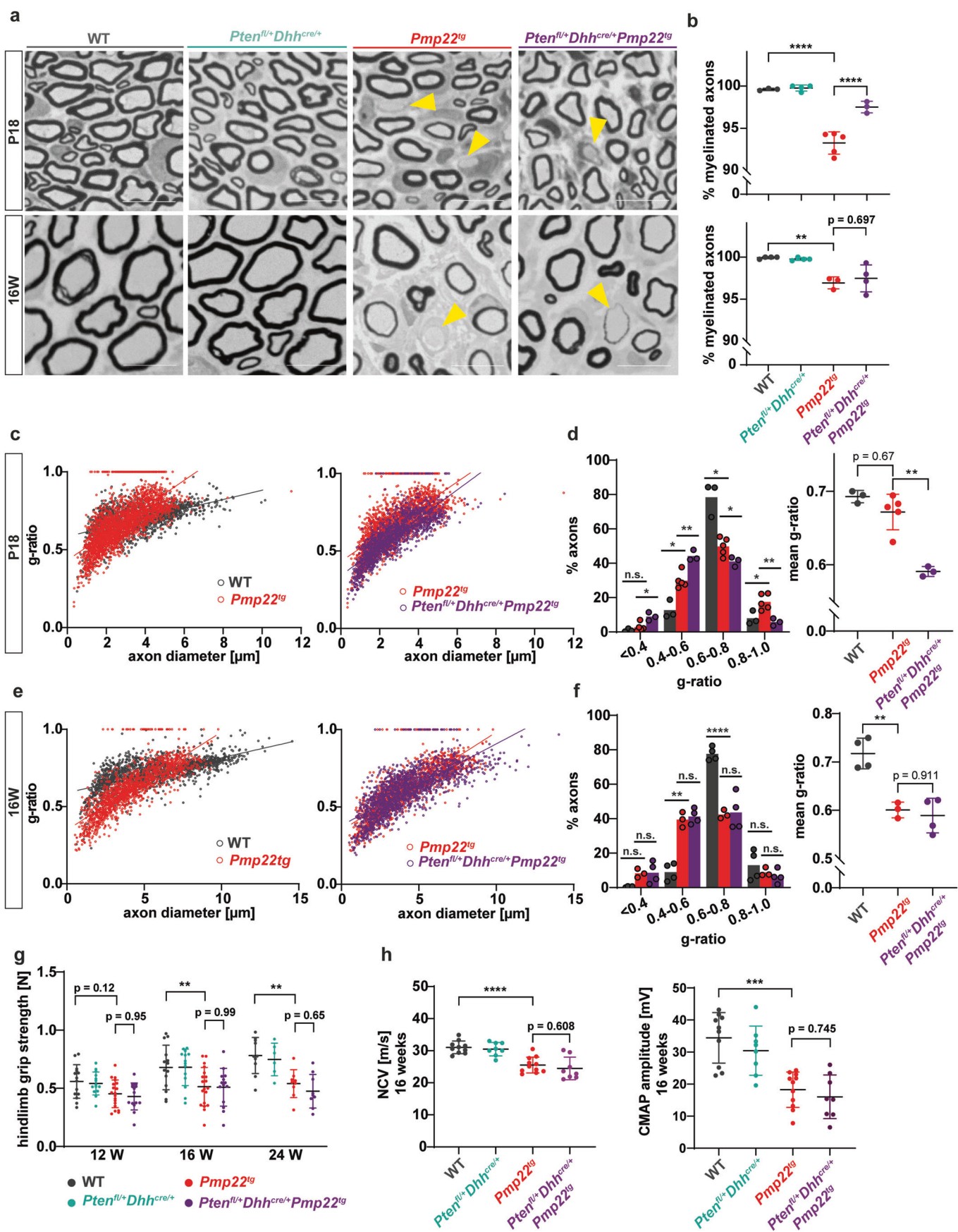

**Figure 6. Reduction of PTEN in *Pmp22^tg* mice increases myelin growth early in development.**

(A) Example images of femoral nerve semi-thin sections from wildtype (WT), *Pten^{fl/+}Dhh^{cre/+}*, *Pmp22^tg* and *Pten^{fl/+}Dhh^{cre/+}Pmp22^tg* mice at P18 (upper panels) and 16 weeks of age (lower panels). Yellow arrowheads indicate amyelinated axons. Scale bar = 10 μm. (B) Quantification of (a) displays a decreased amount of myelinated axons in *Pmp22^tg* whole femoral nerves and an increase in *Pten^{fl/+}Dhh^{cre/+}Pmp22^tg* double mutants (upper panel) at P18 while numbers of myelinated axons are similarly decreased in *Pmp22^tg* and *Pten^{fl/+}Dhh^{cre/+}Pmp22^tg* mice at 16 weeks of age (lower panel). WT $n = 3$–4, *Pten^{fl/+}Dhh^{cre/+}* $n = 3$–4, *Pmp22^tg* $n = 3$–5 and *Pten^{fl/+}Dhh^{cre/+}Pmp22^tg* $n = 3$–4 animals. (C) G-ratio plotted against axon diameter of WT (gray) and *Pmp22^tg* (red) mice (left panel) and *Pmp22^tg* and *Pten^{fl/+}Dhh^{cre/+}Pmp22^tg* (purple) mice (right panel) at P18. WT $n = 3$, *Pmp22^tg* $n = 5$ and *Pten^{fl/+}Dhh^{cre/+}Pmp22^tg* $n = 3$ animals. (D) Distribution of *g*-ratios shown in the left panel displays *Pten^{fl/+}Dhh^{cre/+}Pmp22^tg* femoral nerves have more axons with low *g*-ratios (0.4–0.5) and less axons with higher *g*-ratios (0.8–0.9, 0.9–1.0) compared to *Pmp22^tg* nerves. Mean *g*-ratio (right panel) showed a trend to be decreased in *Pmp22^tg* femoral nerves compared to WT and were significantly decreased in *Pten^{fl/+}Dhh^{cre/+}Pmp22^tg* femoral nerves. WT $n = 3$, *Pmp22^tg* $n = 5$ and *Pten^{fl/+}Dhh^{cre/+}Pmp22^tg* $n = 3$ animals. Groups were compared using two-way ANOVA with Tukey's multiple comparison test (*$p \leq 0.05$, **$p \leq 0.01$, ****$p \leq 0.0001$). (E) G-ratio plotted against axon diameter of WT (gray) and *Pmp22^tg* (red) mice (left panel) and *Pmp22^tg* and *Pten^{fl/+}Dhh^{cre/+}Pmp22^tg* (purple) mice (right panel) at 16 weeks of age. WT $n = 4$, *Pmp22^tg* $n = 3$ and *Pten^{fl/+}Dhh^{cre/+}Pmp22^tg* $n = 4$ animals. (F) No alteration in the distribution of axons over the g-ratio (left panel) as well as mean *g*-ratio (right panel) is observed comparing femoral nerves from *Pmp22^tg* and *Pten^{fl/+}Dhh^{cre/+}Pmp22^tg* mice. WT $n = 4$, *Pmp22^tg* $n = 3$ and *Pten^{fl/+}Dhh^{cre/+}Pmp22^tg* $n = 4$ animals. Groups were compared using two-way ANOVA with Tukey's multiple comparison test (*$p \leq 0.05$, **$p \leq 0.01$, ****$p \leq 0.0001$). (G) The grip strength of hindlimbs is reduced in *Pmp22^tg* and *Pten^{fl/+}Dhh^{cre/+}Pmp22^tg* mice compared to WT controls and *Pten^{fl/+}Dhh^{cre/+}* mice. Behavioral analysis was done at 12, 16 and 24 weeks of age. WT $n = 10$–14, *Pten^{fl/+}Dhh^{cre/+}* $n = 6$–14, *Pmp22^tg* $n = 9$–19 and *Pten^{fl/+}Dhh^{cre/+}Pmp22^tg* $n = 9$–14 mice were analyzed. (H) Nerve conduction velocity (NCV, left panel) and compound muscle action potential amplitudes (CMAP, right panel) are similarly lower in *Pmp22^tg* and *Pten^{fl/+}Dhh^{cre/+}Pmp22^tg* mice compared to WT controls. For electrophysiology measurements WT $n = 10$, *Pten^{fl/+}Dhh^{cre/+}* $n = 8$, *Pmp22^tg* $n = 11$ and *Pten^{fl/+}Dhh^{cre/+}Pmp22^tg* $n = 8$ mice were analyzed. Data information: Means are displayed ±standard deviation. Statistical analysis was performed using one-way ANOVA with Sidak's multiple comparison test (*$p \leq 0.05$, **$p \leq 0.01$, ***$p \leq 0.001$, ****$p \leq 0.0001$) if not indicated otherwise. Source data are available online for this figure.

## Rapamycin therapy

Rapamycin (LC laboratories) was dissolved in vehicle solution containing 5% polyethyleneglycol 400, 5% Tween 80 and 4% ethyl alcohol. HNPP and control mice were i.p. injected with 5 mg Rapamycin per kg bodyweight two times per week from postnatal day 21 until postnatal day 148 with either Rapamycin or placebo (vehicle solution). The weight was continuously controlled and animals were subjected to motor phenotyping tests and electrophysiology at the end of the study.

## Motor phenotyping

The same investigator per experimental group who was blinded towards the genotype performed all phenotyping analyses. Motor performance was assessed by standardized grip strength test and elevated beam test. To assess grip strength, the animals were held by their tail and placed with their forelimbs on a horizontal bar. By gently pulling the animal away, the maximum force was measured in a connected gauge (FMI-210B2 Force Gauge, Alluris). To assess hind limb grip strength, the animal's forelimbs were supported and their hindlimbs placed on the bar. Again, the animal was retracted from the bar and the gauge detected the maximum force applied. All measurements were repeated seven times per animal and the mean calculated. In between fore- and hindlimb grip strength analysis, the animals had a minimum break of 10 min. The elevated beam is an 80 cm long, 14 mm wide bar approximately 60 cm above the ground. The animals were habituated to the beam one day prior to the experimental day. The animals walked the elevated beam for three times and the mean walking time and number of slips was calculated.

## Electrophysiology

For anesthesia, mice were injected i.p. with 6 mg Ketamine (Bayer Vital) and 90 mg Xylazine (WDT) per kg bodyweight. When no toe reflexes were observed anymore, electrophysiological measurements on the sciatic nerve and tail were performed (Evidence 3102evo (Schreiber und Tholen) for CMTxPTEN mutants and

Toennies Neuroscreen (Jaeger, Hoechsberg) for Rapamycin study). To this end, needle electrodes were subcutaneously applied close to the sciatic notch (proximal stimulation) and in close proximity to the ankle (distal stimulation). Motor recording electrodes were inserted in the small muscle on the plantar surface. After proximal and distal supramaximal stimulation the compound muscle action potential (CMAP) was recorded. The distance between the stimulation sites [$m$] divided by the difference of latencies [$s$] allowed to calculate the motor nerve conduction velocity (mNCV). For sensory measurements in the tail, needle electrodes were applied close to the tip of the tail as stimulation electrodes and electrodes in the proximal tail branch served as recording electrodes. Averaged compound sensory nerve potentials were measured after stimulation with 5 mA. According to the distance of electrodes the sensory nerve conduction velocity (sNCV) was calculated.

## Histology

Peripheral nerves were fixed in 2.5% glutaraldehyde and 4% PFA in 0.1 M phosphate buffer for 1 week. Then, they were contrasted with osmium tetraoxide and embedded in epoxy resin (Serva) and 0.5 μm semi-thin sections were cut (Leica RM 2155) using a diamond knife (Histo HI 4317, Diatome). Afterwards sections were stained according to Gallyas (1971) and with Methylene blue/Azur II for 1 min. Analysis was performed on total nerve cross sections using FIJI (NIH).

## Cell culture

Schwann cell (SC)-dorsal root ganglia (DRG) co-cultures were prepared from either E15.5 rat or E13.5 mouse embryos according to standard procedure (Taveggia and Bolino, 2018). After DRGs were digested with 0.25% Trypsin (Invitrogen) at 37 °C for 45 min, the reaction was stopped by adding deactivated fetal calf serum (FCS; HyClone) and basic medium (1% Penicillin/ Streptomycin (Lonza), 10% FCS, 50 ng/ml nerve growth factor (NGF; Alomone Labs) in minimum essential medium (MEM; Gibco)) and cells were plated on collagen-coated coverslips. On day 7, myelination was

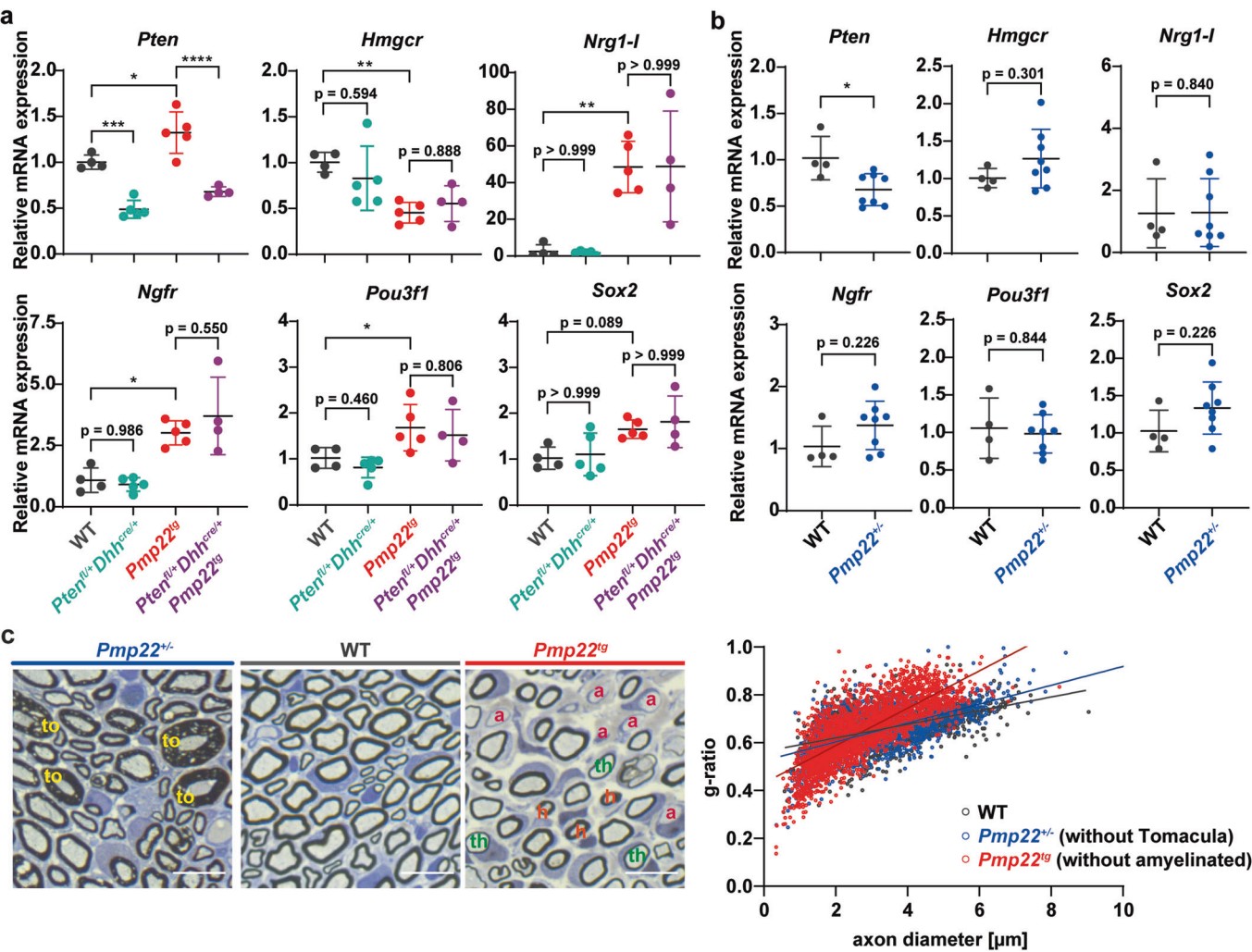

**Figure 7. Dedifferentiation in CMT1A mice is not rescued by PTEN reduction.**

(A) Quantitative RT-PCR from P18 tibial nerves of wildtype (WT) ($n = 4$), $Pten^{fl/+}Dhh^{cre/+}$ ($n = 5$), $Pmp22^{tg}$ ($n = 5$) and $Pten^{fl/+}Dhh^{cre/+}Pmp22^{tg}$ ($n = 5$) mice shows relative mRNA expression of $Pten$, $Hmgcr$, $Nrg1-I$, $Pou3f1$, $Ngfr$, and $Sox2$. $Rplp0$ and $Ppia$ were used as housekeeping genes. (B) Quantitative RT-PCR from P18 tibial nerves of WT ($n = 4$) and $Pmp22^{+/-}$ ($n = 8$) mice shows relative mRNA expression of $Pten$, $Hmgcr$, $Nrg1-I$, $Pou3f1$, $Ngfr$, and $Sox2$. $Rplp0$ and $Ppia$ were used as housekeeping genes. (C) Semi-thin sections of P18 femoral nerves of $Pmp22^{+/-}$ (left), WT (middle) and $Pmp22^{tg}$ (right) mice. $Pmp22^{+/-}$ nerves show myelin overgrowth, the so-called tomacula (to, yellow) and $Pmp22^{tg}$ nerves are characterized by amyelinated (a, pink) and thinly myelinated (th, green) big axons as well as hypermyelinated (h, orange) small axons. Quantification shows g-ratios of $Pmp22^{+/-}$ mice (blue, without tomacula) are similar to the WT (gray) distribution, while $Pmp22^{tg}$ mice (red, without amyelinated) show the hypermyelination of small axons and demyelination of big axons. Data information: Means are displayed ±standard deviation. Statistical analysis was performed using one-way ANOVA with Sidak's multiple comparison test (*$p \leq 0.05$, **$p \leq 0.01$, ***$p \leq 0.001$, ****$p \leq 0.0001$). Source data are available online for this figure.

**Table 1. Genotyping primers.**

| PCR | Primer | Sequence |
|---|---|---|
| Pmp22 tg rat | fwd | 5'-CCAGAAAGCCAGGGAACTC-3' |
| Pmp22 tg rat | rev | 5'-GACAAACCCCAGACAGTTG-3' |
| Pmp22 tg mouse | fwd | 5'-TCAGGATATCTATCTGATTCTC-3' |
| Pmp22 tg mouse | rev | 5'-AAGCTCATGGAGCACAAAACC-3' |
| Dhh-Cre | fwd | 5'-CAGCCCGGACCGACGATGAA-3' |
| Dhh-Cre | rev | 5'-CCTGCGGAGATGCCCAATTG-3' |
| Pten-flox | fwd | 5'-ACTCAAGGCAGGGATGAGC-3' |
| Pten-flox | rev | 5'-CAGAGTTAAGTTTTTGAAGGCAAG-3' |

induced by culturing the cells in basic medium with 50 ng/ml ascorbic acid (AA; Sigma). For PTEN inhibition in $Pmp22^{tg}$ cultures, cells were treated with 1% DMSO (Sigma) as a control or with the PTEN inhibitor VO-OHpic (Rosivatz et al, 2006) (Sigma) at 50 nM, 500 nM and 5 μM, respectively. In $Pmp22^{+/-}$ cultures, cells were treated with either 1% DMSO as control, 20 nM mTOR inhibitor Rapamycin (LC Laboratories) or 10 μM PI3K inhibitor LY294002 (Cell Signaling, #9901). Medium was changed every 2–3 days for 2 weeks. HEK293T cells (Sigma) were maintained in DMEM and regularly tested for mycoplasma contamination. Cells were transiently transfected with ALFA-tagged (Gotzke et al, 2019) PMP22 and Flag-tagged or untagged PTEN expression plasmids (Vectorbuilder) via polyethylenimine.

## Immunocytochemistry

Cells from SC-DRG co-cultures were fixed in 4% paraformaldehyde (PFA) in PBS for 10 min and then permeabilized in a mixture of 95% ice-cold methanol and 5% acetone at −20 °C for 5 min. Afterwards cells were incubated in blocking solution (2% horse serum, 2% bovine serum albumin (BSA), 0.1% porcine gelantine) for 1 h at room temperature before incubation in primary antibodies (mouse anti-MBP 1:500 (Covance), rabbit anti-TUJ1 1:250 (Covance)) overnight at 4 °C. The next day, cells were washed in PBS for three times and incubated in secondary antibodies (Alexa 488 donkey anti mouse 1:1000 (Invitrogen) and Alexa 568 donkey anti rabbit 1:1000 (Invitrogen) diluted in blocking solution with 0.2 µg/µl 4',6'-diamidino-2-phenylindole (DAPI; Sigma) for 1 h at room temperature. Following three washing steps in PBS cells were mounted on slides in Mowiol mounting solution (9.6% Mowiol (Sigma), 24% Glycerol, 0.1 M Tris-HCl). Fluorescent images were taken using a Axiophot Observer Z (Zeiss) with a Colibri light source (Zeiss) and MRM camera (Zeiss). Acquisition and processing of the images was carried out using Zen2.6 blue software (Zeiss), FIJI (NIH) and Illustrator 2020 (Adobe).

## Immunohistochemistry

For immunohistochemistry analysis of cross sections, femoral nerves were dissected, immersion fixed in 4% PFA for 24 hours and imbedded in paraffin. 5 µm cross sections were deparaffinized using a standard xylol and ethanol series. After target retrieval was induced by heating in citrate buffer samples were blocked in 10% BSA and 20% goat serum in PBS for 20 min. Incubation with primary antibodies (rabbit anti PTEN 1:50 (CST, #9188), mouse anti TUJ1 1:250 (Covance), mouse anti P0 1:200 (provided by J.J. Archelos (Archelos et al, 1993)), rabbit anti Phospho-S6 (CST, #4858)) in PBS was carried out over night at 4 °C. To quantify the internodal length, teased fibers were prepared from P18 or 16 weeks old sciatic nerves. Following dissection, connective tissue was removed and nerves teased into single fibers using fine forceps, after 15 min of drying fibers were fixed in 4% PFA for 5 min and permeabilized in ice-cold 95% methanol and 5% acetone for another 5 min. Furthermore, fibers were three times washed in PBS and incubated in blocking solution (10% horse serum, 1% BSA, 0.025% Triton-X-100 in PBS) for 1 h at room temperature. After incubation with primary antibodies (mouse anti MAG 1:50 (Chemicon clone 513), rabbit anti NaV1.6 1:250 (Alomone Labs #ASC-009)) in blocking solution, three washing steps in PBS followed. Secondary antibody (donkey anti mouse Alexa 488 1:1000, donkey anti rabbit Alexa 568 1:1000 (Invitrogen)) incubation was proceeded in the same way in cross-sections and teased fibers, they were incubated with 0.2 µg/µl 4',6'-diamidino-2-phenylindole (DAPI; Sigma) for 1 h at room temperature. After three washing steps in PBS cells were mounted on slides in Mowiol mounting solution (9.6% Mowiol (Sigma), 24% Glycerol, 0.1 M Tris-HCl) and fluorescent images were obtained using a Axiophot Observer Z (Zeiss) with a Colibri light source (Zeiss) and MRM camera (Zeiss). Acquisition, processing and analysis of the images was carried out using Zen2.6 blue software (Zeiss), FIJI (NIH) and Illustrator 2020 (Adobe).

## Myelin purification

Myelin purification was performed as previously described (Larocca and Norton, 2006). In short, 6 dissected sciatic nerves of rats at postnatal day 18 were homogenized in 0.27 M sucrose (PreCellys, 6000 rpm, 2 × 15 s). Part of the homogenate was kept as "lysate control" and the remaining homogenate was layered over 0.83 M sucrose in centrifugation tubes and centrifuged at 75,000 × $g$ for 30 min at 4 °C. The interface between the sucrose gradients (Myelin enriched fraction) was carefully transferred to a new centrifugation tube with 20 mM Tris-Cl buffer and again centrifuged at 75,000 × $g$ for 15 min at 4 °C. The pellet was resuspended in 20 mM Tris-Cl buffer solution and centrifuged at 12,000 × $g$ for 15 min. The final pellet (purified myelin) was taken up in 20 mM Tris-Cl buffer and further processed for protein analysis.

## Protein analysis

Sciatic nerves were homogenized using a Precellys24 homogenizer (VWR) in sucrose lysis buffer (270 nM sucrose, 10 mM Tris-HCl, 1 mM NaHCO$_3$, 1 mM MgCl$_2$, protease inhibitor (cOmplete Mini, Roche), phosphatase inhibitor (PhosphoStop, Roche)) and dissolved in loading buffer (40% glycine [w/v], 240 mM Tris-HCl, 8% SDS [w/v], 0.04% bromophenol blue [w/v], 1 mM DTT). Proteins were separated by electrophoresis using 4–16% or 4-20% pre-cast polyacrylamide gels (Mini-PROTEAN®TGX™, Biorad or Novex™ Tris-Glycin, Thermo Fisher) and PageRuler Plus Prestained Protein Ladder (Thermo Fisher) as loading and size control. Afterwards, proteins were transferred to a methanol-activated Amersham Hybond PVDF membrane (GE Healthcare). Fast Green staining was used to detect the whole protein content on the membrane. Shortly, membrane were rinsed in water to remove transfer buffer residues, incubated in Fast Green staining solution (0.5% Fast Green [w/v] (Sigma), 30% methanol, 6.7% glacial acetic acid) for 5 min, following a two times washing step in washing solution (30% methanol, 6.7% glacial acetic acid) before fluorescent imaging on a ChemoStar (INTAS Science Imaging Instruments). After blocking in Milk-TBST (non-phospho antibodies, 5% milkpowder [w/v], 25 mM Tris, 75 mM NaCl, 5% Tween 20 [w/v]) or BSA-TBST (phospho antibodies, 5% BSA [w/v], 25 mM Tris, 75 mM NaCl, 5% Tween 20 [w/v]) for 1 h at room temperature, Western Blots were incubated overnight at 4 °C in primary antibodies against PMP22 (1:1000, Assay Biotech), PTEN (1:1000, CST #9188), TUJ1 (1:1000, Covance), P0 (1:000, J.J. Archelos (Archelos et al, 1993), p-S6 (1:1000, CST #5364), p-S6K (1:1000, CST #9208). For the following antibodies, membranes were incubated at room temperature in 5% milk in TBST for 1 h: S6 (1:1000, CST #2317), S6K (1:1000, CST #2708), ALFA-tag (1:3000, NanoTag).). After 3–5 washing steps in TBST, membranes were incubated in HRP secondary antibodies (1:5000, Dianova, 45 min room temperature) against the respective species and developed using Western Lightning Plus-ECL-Kit (Perkin Elmer) and the ChemoStar. For interaction analysis of heterologously expressed proteins, plasmids encoding for PTEN with an N-terminal FLAG-tag (Vectorbuilder, #VB220531-1217bxp), untagged PTEN (#VB220603-1029rjt) or PMP22 with a C-terminal ALFA-tag (#VB210322-1182nwx) were transfected into HE293T cells using Lipofectamine 3000 (Thermo Fisher). For immunoprecipitation, sciatic nerves homogenized as above without phosphatase inhibitor,

or transfected HEK293T cells were lysed in 1% (w/v) dodecyl-β-d-maltoside (DDM) for 1 h at room temperature. DDM was diluted to 0.15% and insoluble material was pelleted via centrifugation at $100,000 \times g$. Immunoprecipitation was initiated by addition of antibody-coupled magnetic (PMP22, Sigma-Aldrich # SAB4502217; ALFA-tag, ALFA-Selector PE, NanoTag) or agarose beads (Flag-tag, DYKDDDDK Fab-Trap, Chromotek) to the supernatant. As controls, beads coupled to unspecific antibodies or empty beads were used. After 1 h incubation at 4 °C and three washes with buffer containing 0.025% DDM, protein was eluted with loading buffer (endogenous PMP22) or specific peptides (ALFA, Flag-tag). PMP22 for pull-down experiments was produced in HEK293T cells transfected with PMP22-ALFA (plasmid #VB210322-1182nwx) using polyethylenimine. Cells were lysed, solubilized with 2% DDM overnight, DDM was diluted to 1% and ultracentrifuged as above. The supernatant was bound to ALFA-Selector PE agarose beads at 4 °C o.n., washed and PMP22-ALFA eluted with ALFA-peptide. PMP22-ALFA was further purified via size exclusion chromatography (SEC; Superdex 200 Increase, Cytiva), and fractions pooled and concentrated. After a check for homogeneity via SEC, 10 μg of purified, concentrated PMP22-ALFA per biological replicate of sciatic nerve homogenate were bound to ALFA-Selector CE magnetic beads, consecutively for 1 h at room temperature and 2 h at 4 °C. Sciatic nerves homogenized as above (140 μg protein per biological replicate) were treated with 0.23% (w/v) DDM for 3 h at 4 °C. DDM was diluted to 0.047% and ultracentrifuged as above. Lysates were precleared with ALFA-Selector CE magnetic beads for 2 h at 4 °C and then incubated with PMP22-bound (pull-down) or empty (control) ALFA-Selector CE magnetic beads for 2 h at 4 °C. After six washes, elution of PMP22-ALFA (pull-down) or mock-elution (control) was performed 3 times with ALFA-peptide, first for 10 min at 25 °C, then for 10 min at 37 °C and finally o.n. at 4 °C, and pull-down or control eluates combined.

### RNA analysis

Total RNA was extracted from tibial nerves (without epineurium) using QIAzol lysis reagent and was purified according to manufacturer's instructions using RNeasy Kit (Qiagen). The concentration and purity of the RNA was determined with the ratio of absorption at 260/280 nm using NanoDrop 2000 Spectraphotometer. Following, cDNA was synthesized from total RNA using poly-Thymin and random nonamer primers and Superscript III RNase H reverse transcriptase (Invitrogen). Quantitative real time PCR was carried out using the GoTaq qPCR Master Mix (Promega) and LC480 detection system (Roche). Reactions were carried out in four replicates. The cycle threshold (Ct) value was calculated using LC480 software (Roche) and mean values were normalized as fold changes to the control group. As internal standards peptidyl isomerase A (*Ppia*) and ribosomal protein large P0 (*Rplp0*) were used. Primer sequences are listed in Table 2.

### Statistical analysis

All data are presented as mean ± standard deviation unless indicated otherwise. Data processing and statistical analysis was performed using MS Excel 2016 and GraphPad Prism 8. Statistical tests are indicated in the figure legends. In short, two groups were

**Table 2. qRT-PCR primers.**

| Gene | Primer | Sequence |
| --- | --- | --- |
| *Pmp22* | fwd | 5′-GGCTGTCCCTTTGAACTGAA-3′ |
| | rev | 5′-AACAGGATCCCCAACAAGAGT-3′ |
| *Pten* | fwd | 5′-GAGGCCCTGGATTTTTATGG-3′ |
| | rev | 5′-CGCCTCTGACTGGGAATAGT-3′ |
| *Hmgcr* | fwd | 5′-GACCTTTCTAGAGCGAGTGCAT-3′ |
| | rev | 5′-CGCTATATTCTCCCTTACTTCATCC-3′ |
| *Nrg1-I* | fwd | 5′-GGGAAGGGCAAGAAGAAGG-3′ |
| | rev | 5′-TTTCACACCGAAGCACGAGC-3′ |
| *Ngfr* | fwd | 5′-CGGTGTGCGAGGACACTGAGC-3′ |
| | rev | 5′-TGGGTGCTGGGTGTTGTGACG-3′ |
| *Pou3f1* | fwd | 5′-GCGTGTCTGGTTCTGCAAC-3′ |
| | rev | 5′-AGGCGCATAAACGTCGTC-3′ |
| *Sox2* | fwd | 5′-TCCAAAAACTAATCACAACAATCG-3′ |
| | rev | 5′-GAAGTGCAATTGGGATGAAAA-3′ |
| *Rplp0* | fwd | 5′-CGAGAAGACCTCTTTCTTCCAA-3′ |
| | rev | 5′-AGTCTTTATCAGCTGCACATCG-3′ |
| *Ppia* | fwd | 5′-TGCTGGACCAAACACAAATG-3′ |
| | rev | 5′-CACCTTCCCAAAGACCACAT-3′ |

compared using Student's *t* test. More than two groups were compared using one-way ANOVA with an appropriate post test. For comparing more than two groups for more than one time point (longitudinal analysis) the two-way ANOVA with appropriate post test was used. Statistical differences are marked by stars when $*p < 0.05$, $**p < 0.01$, $***p < 0.001$ and $****p < 0.0001$).

## Data availability

We have no data that require deposition in a public database.

## Peer review information

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

## Acknowledgements

We thank C. Huxley (Imperial College School of Medicine, London) for providing *Pmp22tg* mice, D. Meijer (University Edinburgh) for *DhhCre* mice and U. Suter (ETH Zürich) for *Pmp22+/-* mice. Further we want to acknowledge U. Bode, B. Veith, C. Wieczorek, T. Hoffmeister, S. Schulze, T. Ruhwedel, and W. Möbius (MPI of Multidisciplinary Sciences, Göttingen) for their excellent technical support. We want to thank Prof. Dr. med. K. Toyka for introducing and teaching the in-vivo electrophysiological measurements. MWS is supported by the German Ministry of Education and Research (BMBF, CMT-NET, FKZ: 01GM1511C and CMT-NRG, FKZ: 01GM1605), the German Research Foundation (DFG, SE 1944/3-1 with KAN) and the Volkswagen Foundation

(with DE). MWS heads the Neurogenetics clinic and co-heads the neuromuscular center at UMG. He is member of the Inherited Neuropathy Consortium RDCRN and of the European Reference Network for Rare Neuromuscular Diseases (ERN EURO-NMD).

## Author contributions

**Doris Krauter**: Conceptualization; Data curation; Formal analysis; Validation; Investigation; Visualization; Writing—original draft; Writing—review and editing. **Daniela Stausberg**: Data curation; Formal analysis; Visualization; Writing—review and editing. **Timon J Hartmann**: Data curation; Formal analysis. **Stefan Volkmann**: Data curation; Formal analysis. **Theresa Kungl**: Data curation; Formal analysis; Visualization; Writing—review and editing. **David A Rasche**: Investigation; Data curation. **Gesine Saher**: Writing—review and editing. **Robert Fledrich**: Data curation; Formal analysis; Writing—review and editing. **Ruth M Stassart**: Writing—review and editing. **Klaus-Armin Nave**: Supervision; Writing—review and editing. **Sandra Goebbels**: Supervision; Writing—review and editing. **David Ewers**: Conceptualization; Data curation; Formal analysis; Funding acquisition; Visualization; Writing—review and editing. **Michael W Sereda**: Resources; Supervision; Funding acquisition; Writing—review and editing.

## Funding

## Disclosure and competing interests statement

The authors declare no competing interests.

# Expanded View Figures

**Figure EV1. PMP22 gene dosage dependent alteration of PTEN abundance in PMP22$^{+/-}$ HNPP mice and PMP22$^{tg}$ CMT1A rat sciatic nerve lysates.** ▶

(A) Western Blot analysis showing a decrease of PTEN protein levels in sciatic nerve lysates of Pmp22$^{+/-}$ mice at 9 weeks (n = 4, left panel) and at postnatal day 21 (n = 4, right panel) compared to wildtype (WT) control. Fast green whole protein staining was used as loading control for the quantification. (B) Western Blot analysis showing a decrease of PTEN protein levels in sciatic nerve lysates of Pmp22$^{tg}$ rats at 9 weeks (n = 4, left panel) and at postnatal day 18 (n = 3, right panel) compared to wildtype (WT) control. Whole protein staining was used as loading control for the quantification. (C) Sciatic nerve semi-thin sections of WT (n = 3) and Pmp22$^{+/-}$ mice (n = 3) at postnatal day 6; myelin aberrations are highlighted with yellow arrowheads (left panel). Quantification shows increased percentage of axons with myelin aberrations in sciatic nerves from PMP22$^{+/-}$ mice (left panel). Scale bar is 50 μm. (D) Quadriceps motor nerve semi-thin sections of WT (n = 3) and Pmp22$^{+/-}$ mice (n = 3) at postnatal day 18; myelin aberrations are highlighted with yellow arrowheads (left panel). Quantification shows increased percentage of axons with myelin aberrations in sciatic nerves from PMP22$^{+/-}$ mice (left panel). Scale bar is 25 μm. Data information: Means are displayed ± standard deviation. Statistical analysis was performed using Student's t test (*p < 0.05, **p < 0.01).

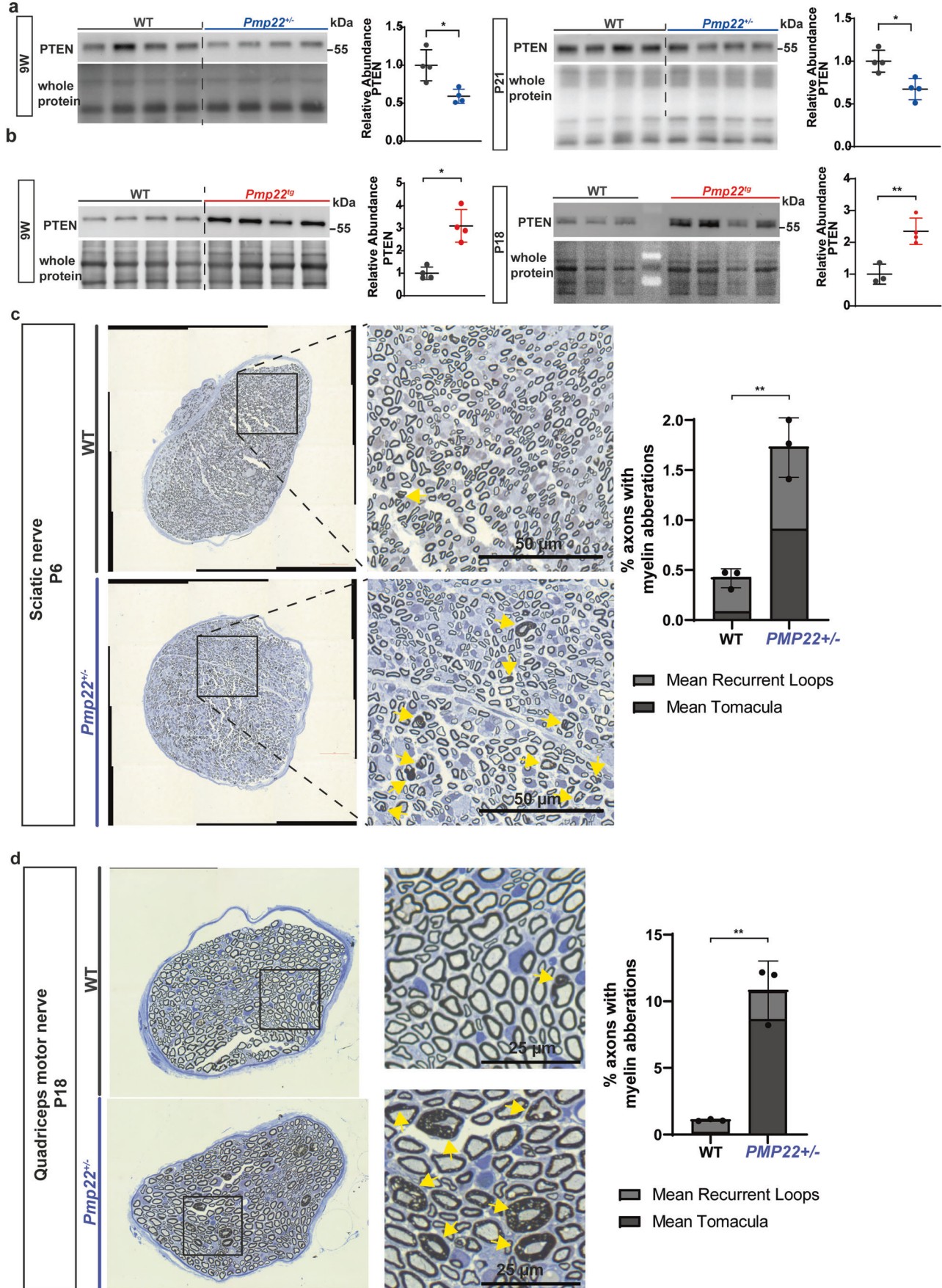

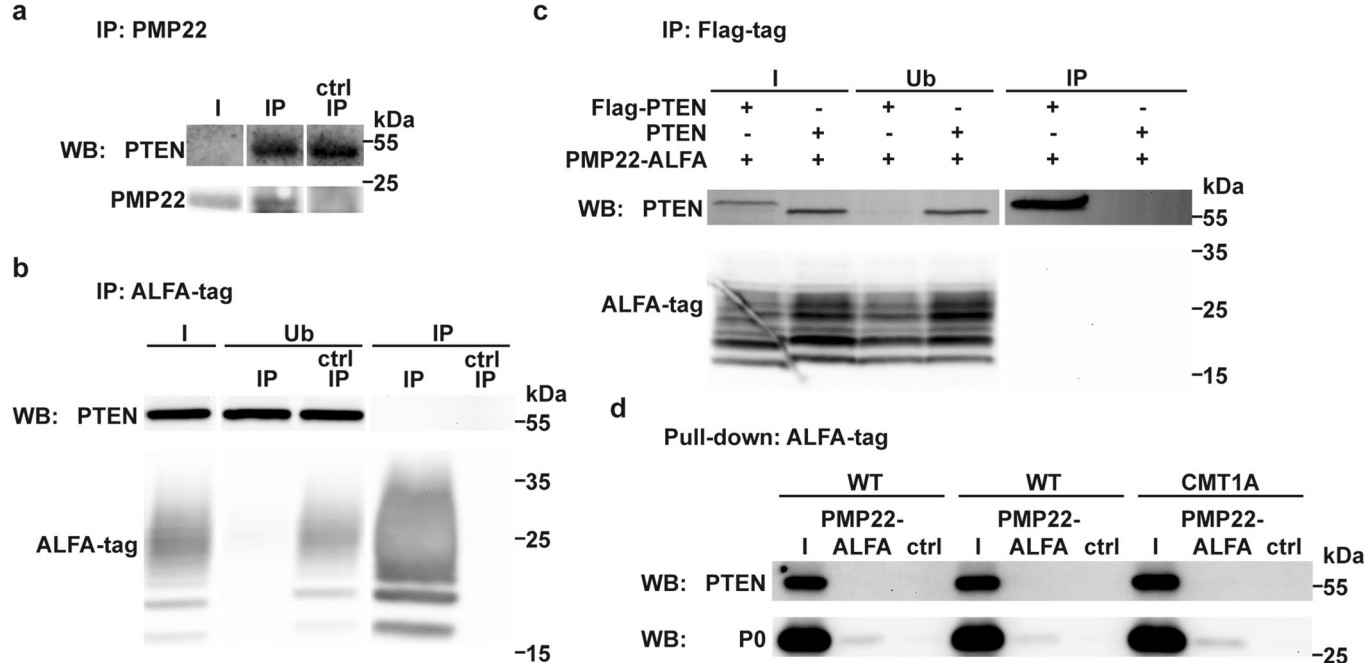

**Figure EV2.  No evidence for molecular interaction between PMP22 and PTEN in peripheral nerve or in cell culture.**

(A) Immunoprecipitation of PMP22 from rat sciatic nerve (P18). Western Blot (WB) analysis shows unspecific binding of PTEN, while PMP22 was specifically detected in PMP22 immunoprecipitation eluate (IP) and not in control eluate (ctrl IP). Input nerve homogenate (I) was diluted 50× for WB analysis. (B) Immunoprecipitation from HEK293T cells after transfection of PMP22-ALFA. WB analysis shows endogenous PTEN in the cell lysate (I) and in the supernatant after the binding step (unbound (Ub)), but not in the immunoprecipitation eluate. I and Ub were diluted 10× for WB analysis. (C) Immunoprecipitation from HEK293T cells after co-transfection of PMP22-ALFA with FLAG-PTEN or untagged PTEN. WB analysis shows specific immunoprecipitation of FLAG-PTEN but not of PMP22-ALFA or untagged PTEN. (D) Pull-down assay on rat sciatic nerve (P18) using purified PMP22-ALFA as prey. WB analysis shows PTEN in the nerve lysate (I) but not in the PMP22-ALFA eluate (PMP22-ALFA) or control eluate (ctrl) in both WT and CMT1A, while P0 was pulled down by PMP22-ALFA. I was diluted 3.33× for WB analysis.

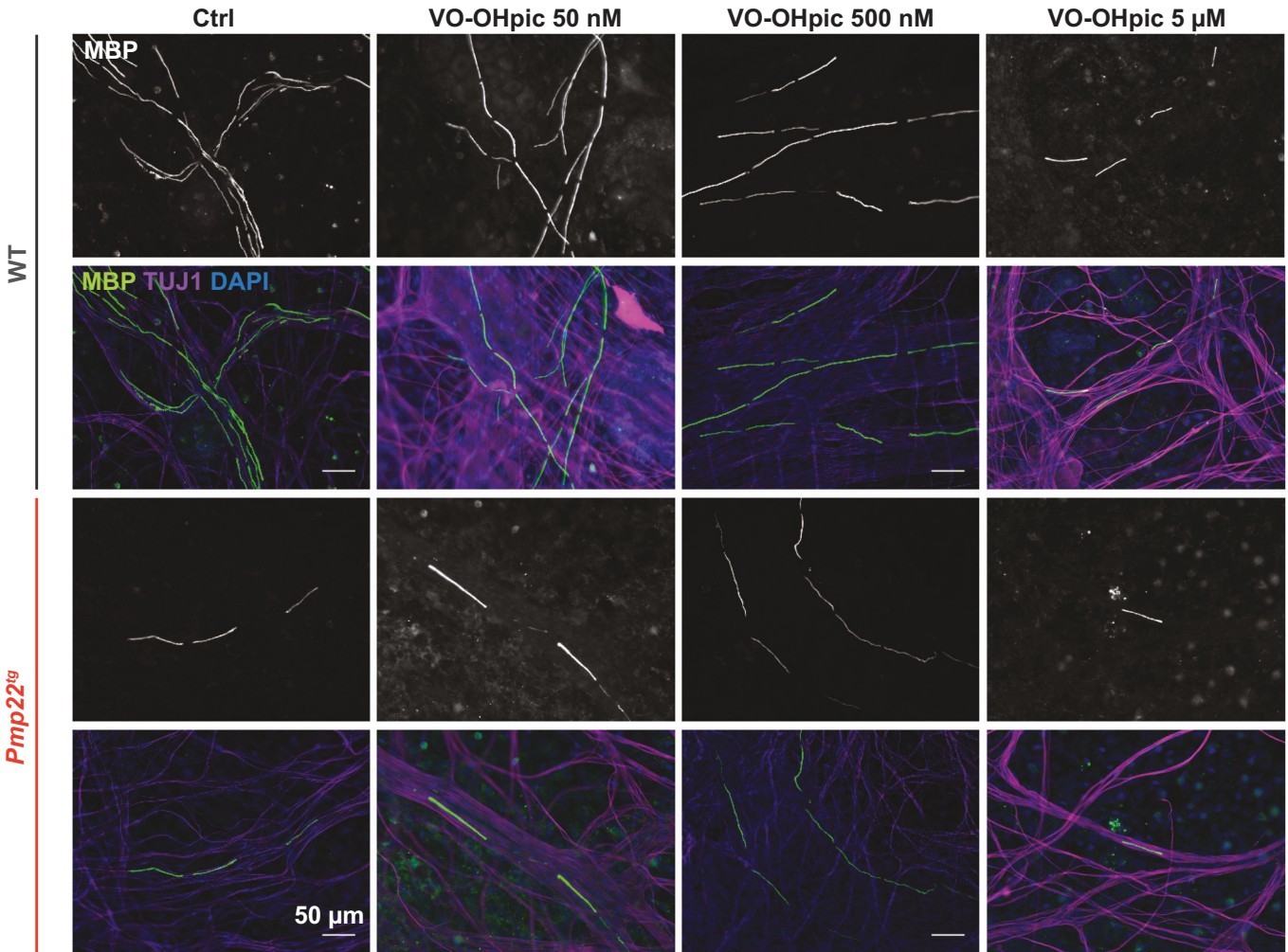

**Figure EV3.  Dose-dependent response of myelination upon PTEN inhibition in *Pmp22^tg* co-cultures in vitro.**

Example images of SC-DRG co-cultures from wildtype (WT) and *Pmp22^tg* rats, treated with different dosages of the PTEN inhibitor VO-OHpic for 14 days. The number of myelinated segments (MBP; gray/green) decreases in WT cultures with increasing inhibitor dosage. In *Pmp22^tg* co-cultures an increase is observed up to 500 nM VO-OHpic but a decrease with 5 μM VO-OHpic. Scale bar is 50 μm. Images for 500 nM VO-OHpic treatment are the same as used in Fig. 4B.

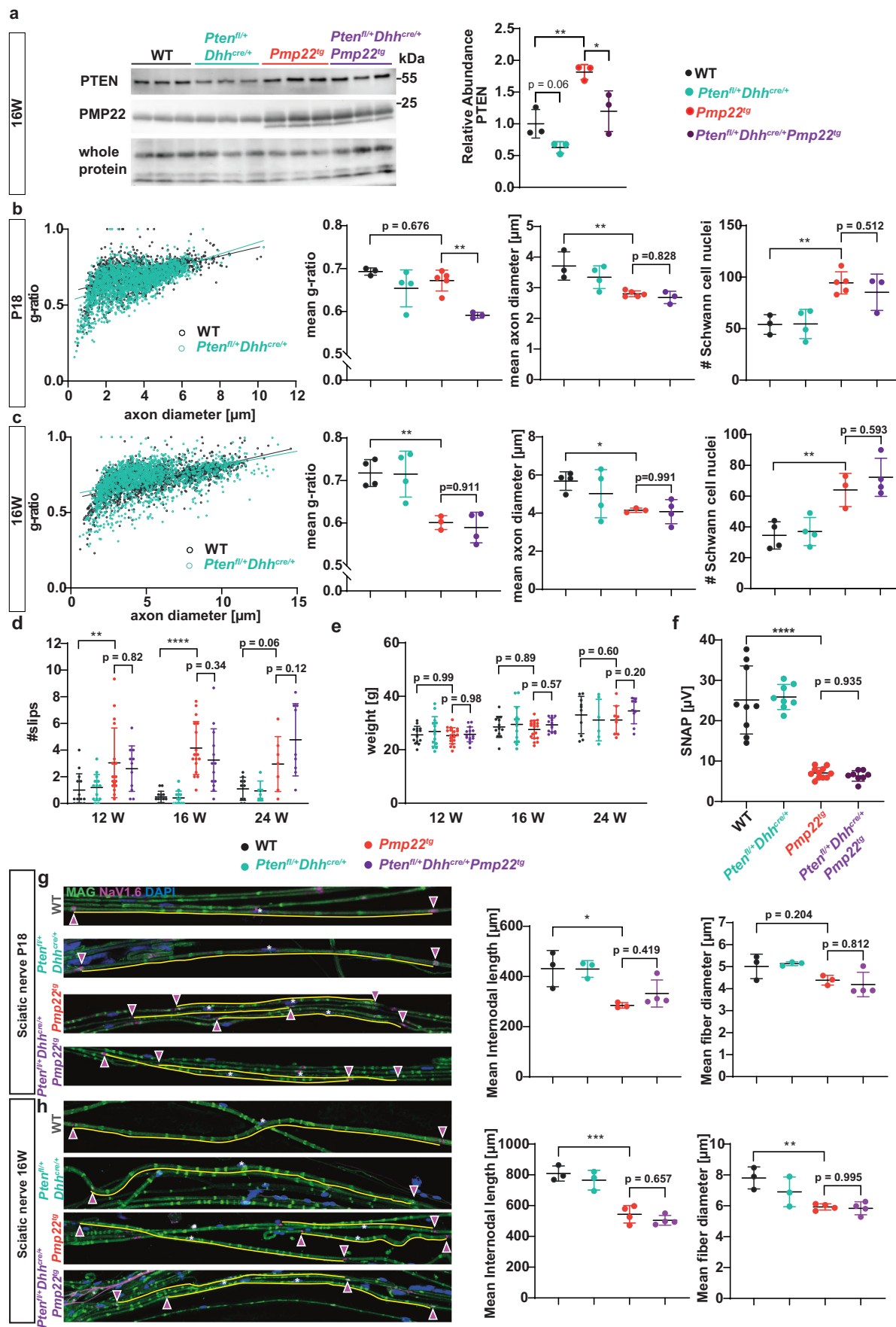

**Figure EV4.   Unaltered internodal length and myelin sheath thickness in *Pten^{fl/+}Dhh^{cre/+}* mice.**

(A) Western Blot analysis shows PTEN and PMP22 protein amounts in whole sciatic nerve lysates from 16 weeks old WT, PTEN heterozygous knockout (*Pten^{fl/+}Dhh^{cre/+}*), CMT1A (*Pmp22^{tg}*) and double mutant (*Pten^{fl/+}Dhh^{cre/+}Pmp22^{tg}*) mice using whole protein staining as loading control. (B) *G*-ratio plotted against axon diameter of wildtype (WT, gray) and *Pten^{fl/+}Dhh^{cre/+}* (turquoise) femoral nerves at P18. Mean *g*-ratio is unaltered in *Pten^{fl/+}Dhh^{cre/+}* and *Pmp22^{tg}* mice compared to WT controls and decreased in *Pten^{fl/+}Dhh^{cre/+}Pmp22^{tg}* mice (left panel). Mean axon diameters are reduced in *Pmp22^{tg}* and *Pten^{fl/+}Dhh^{cre/+}Pmp22^{tg}* mice (middle panel). The number of Schwann cell nuclei per femoral nerve cross section is increased in *Pmp22^{tg}* and *Pten^{fl/+}Dhh^{cre/+}Pmp22^{tg}* mice. WT $n = 3$, *Pten^{fl/+}Dhh^{cre/+}* $n = 4$, *Pmp22^{tg}* $n = 5$ and *Pten^{fl/+}Dhh^{cre/+}Pmp22^{tg}* $n = 3$ animals. (C) *G*-ratio plotted against axon diameter of WT (gray) and *Pten^{fl/+}Dhh^{cre/+}* (turquoise) femoral nerves at 16 weeks of age. Mean g-ratio is unaltered in *Pten^{fl/+}Dhh^{cre/+}* mice compared to WT controls and decreased in *Pmp22^{tg}* and *Pten^{fl/+}Dhh^{cre/+}Pmp22^{tg}* mice (left panel). Mean axon diameters are reduced in *Pmp22^{tg}* and *Pten^{fl/+}Dhh^{cre/+}Pmp22^{tg}* mice (middle panel). The number of Schwann cell nuclei per femoral nerve cross section is increased in *Pmp22^{tg}* and *Pten^{fl/+}Dhh^{cre/+}Pmp22^{tg}* mice. WT $n = 4$, *Pten^{fl/+}Dhh^{cre/+}* $n = 4$, *Pmp22^{tg}* $n = 3$ and *Pten^{fl/+}Dhh^{cre/+}Pmp22^{tg}* $n = 4$ animals. (D) The number of slips on the elevated beam is similarly increased in *Pmp22^{tg}* and *Pten^{fl/+}Dhh^{cre/+}Pmp22^{tg}* mice compared to wildtype controls at all time points. Behavioral analysis was done at 12, 16 and 24 weeks of age. WT $n = 10$–14, *Pten^{fl/+}Dhh^{cre/+}* $n = 6$–14, *Pmp22^{tg}* $n = 9$–19 and *Pten^{fl/+}Dhh^{cre/+}Pmp22^{tg}* $n = 9$-14 mice were analyzed. (E) Neither the weight of *Pmp22^{tg}* nor *Pten^{fl/+}Dhh^{cre/+}Pmp22^{tg}* mice is altered compared to wildtype controls at 12, 16 and 24 weeks of age. WT $n = 10$–14, *Pten^{fl/+}Dhh^{cre/+}* $n = 6$–14, *Pmp22^{tg}* $n = 9$–19 and *Pten^{fl/+}Dhh^{cre/+}Pmp22^{tg}* n = 9-14 mice were analyzed. (F) Sensory nerve action potential amplitudes (SNAP) are decreased in the tail of *Pmp22^{tg}* and *Pten^{fl/+}Dhh^{cre/+}Pmp22^{tg}* mice compared to wildtype controls. For electrophysiology measurements WT $n = 10$, *Pten^{fl/+}Dhh^{cre/+}* $n = 8$, *Pmp22^{tg}* $n = 11$ and *Pten^{fl/+}Dhh^{cre/+}Pmp22^{tg}* $n = 8$ mice were analyzed. (G) Example images of teased fiber preparations of WT, *Pten^{fl/+}Dhh^{cre/+}*, *Pmp22^{tg}* and *Pten^{fl/+}Dhh^{cre/+}Pmp22^{tg}* double mutants stained for MAG (green), NaV1.6 (magenta) and DAPI (blue) at P18. Internodes between two nodes (magenta arrowheads) are underlined in yellow and respective Schwann cell nuclei are marked by white stars. Mean internodal length (left panel) is significantly reduced in *Pmp22^{tg}* teased fibers compared to wildtype controls at P18, whereas *Pten^{fl/+}Dhh^{cre/+}Pmp22^{tg}* mice do not differ in internodal length compared to *Pmp22^{tg}* mice. Mean fiber diameters are not significantly altered (right panel). Analysis was performed on 100 internodes of $n = 3$–4 animals per group. (H) Example images of teased fiber preparations of WT, *Pten^{fl/+}Dhh^{cre/+}*, *Pmp22^{tg}* and *Pten^{fl/+}Dhh^{cre/+}Pmp22^{tg}* double mutants stained for MAG (green), NaV1.6 (magenta) and DAPI (blue) at 16 weeks of age. Internodes between two nodes (magenta arrowheads) are underlined in yellow and respective Schwann cell nuclei are marked by white stars. Mean internodal length (left panel) and fiber diameter (right panel) are significantly reduced in *Pmp22^{tg}* teased fibers compared to wildtype controls at 16 weeks of age, whereas *Pten^{fl/+}Dhh^{cre/+}Pmp22^{tg}* mice do not differ in internodal length compared to *Pmp22^{tg}* mice. Analysis was performed on 100 internodes of $n = 3$–4 animals per group. Data information: Means are displayed ± standard deviation. Statistical analysis was done using one-way ANOVA with Sidak's multiple comparison test (*$p ≤ 0.05$, **$p ≤ 0.01$, ***$p ≤ 0.001$, ****$p ≤ 0.0001$).

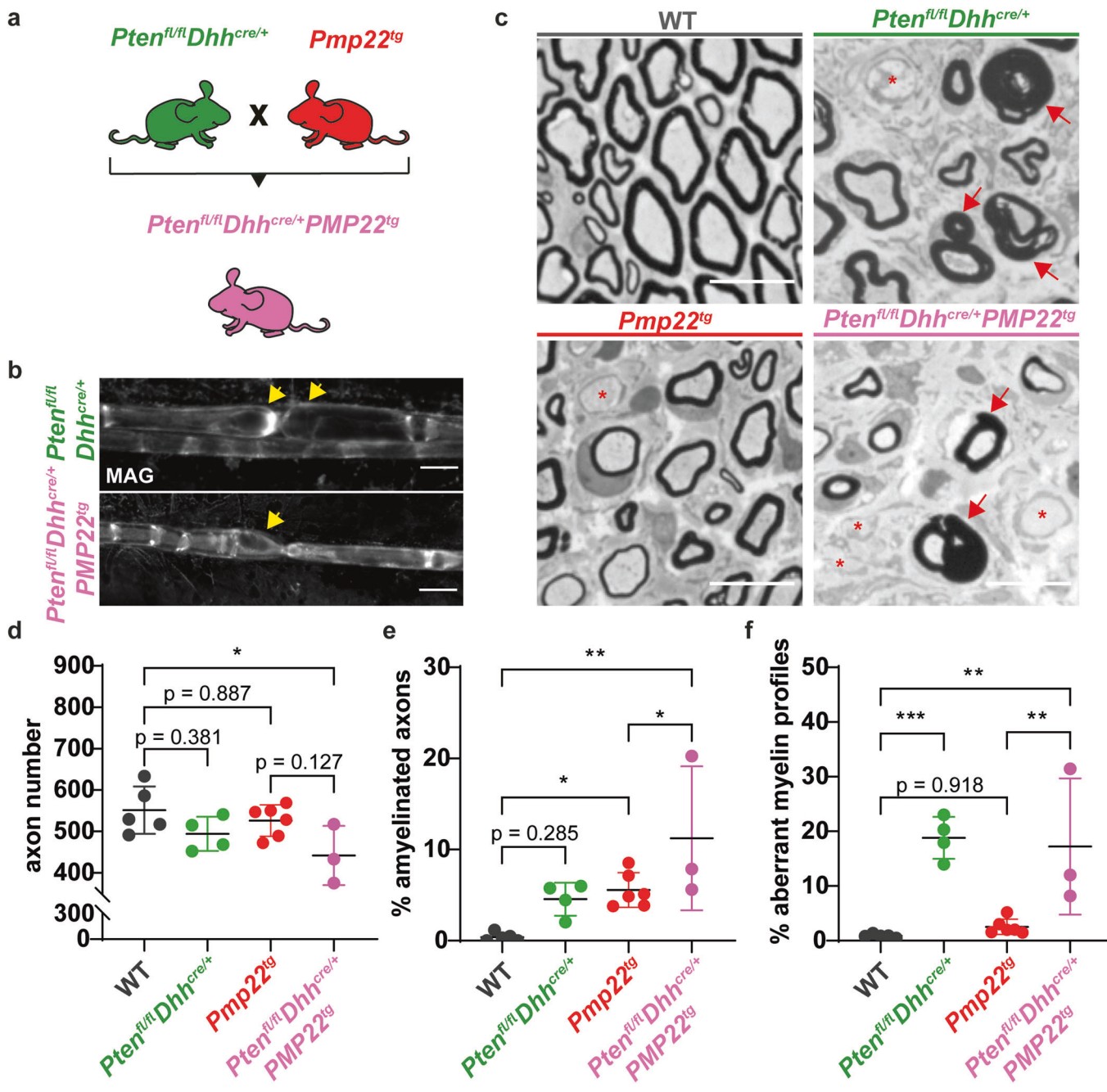

**Figure EV5.** *Pten ablation in Pmp22$^{tg}$ mice leads to myelin abnormalities.*

(A) Crossing scheme of Schwann cell specific full Pten knockout mice (*Pten$^{fl/fl}$Dhh$^{cre/+}$*) with CMT1A mice (*Pmp22$^{tg}$*) to generate a full *Pten* knockout in CMT1A mice (*Pten$^{fl/fl}$Dhh$^{cre/+}$Pmp22$^{tg}$*). (B) Teased fiber preparations of 8 weeks old *PTEN$^{fl/fl}$Dhh$^{cre/+}$* (upper panel) and *Pten$^{fl/fl}$Dhh$^{cre/+}$Pmp22$^{tg}$* mice (lower panel) show focal myelin thickening at paranodal loops as indicated by yellow arrows. Scale bar = 20 μm. (C) Semi-thin cross section of femoral nerves from WT, *Pten$^{fl/fl}$Dhh$^{cre/+}$*, *Pmp22$^{tg}$* and *Pten$^{fl/fl}$Dhh$^{cre/+}$Pmp22$^{tg}$* mice at 8 weeks of age. Red arrows indicate myelin abnormalities such as outfoldings and tomacula, asterisks indicate amyelinated axons. Scale bar = 10 μm. (D–F) Quantification of (C) displays reduced axon numbers in *Pten$^{fl/fl}$Dhh$^{cre/+}$Pmp22$^{tg}$* mice compared to wildtype controls (D). The percentage of amyelinated axons is increased in *Pmp22$^{tg}$* and further elevated in *Pten$^{fl/fl}$Dhh$^{cre/+}$Pmp22$^{tg}$* mice (E). *Pten* depletion alone and in *Pmp22$^{tg}$* leads to an increase in axons with aberrant myelin profiles (F). WT $n = 5$, *PTEN$^{fl/fl}$Dhh$^{cre/+}$* $n = 4$, *Pmp22$^{tg}$* $n = 6$ and *Pten$^{fl/fl}$Dhh$^{cre/+}$Pmp22$^{tg}$* $n = 3$ animals were analyzed. Data information: Means are displayed ± standard deviation. Statistical analysis was done using one-way ANOVA with Sidak's multiple comparison test (*$p \leq 0.05$, **$p \leq 0.01$, ***$p \leq 0.001$).

