## [Peer Review File · EMBO Molecular Medicine]

Targeting PI3K/Akt/mTOR signaling in rodent models of PMP22 gene-dosage diseases

Doris Krauter, Daniela Stausberg, Timon Hartmann, Stefan Volkmann, Theresa Kungl, Gesine Saher, Robert Fledrich, Ruth Stassart, Klaus-Armin Nave, Sandra Goebbels, David Ewers, and Michael Sereda

DOI: 10.15252/emmm.202318047

Corresponding author(s): Michael Sereda (sereda@mpinat.mpg.de)

Review Timeline:

Transfer from Review Commons:	22nd May 23
Editorial Decision:	25th May 23
Revision Received:	25th Aug 23
Editorial Decision:	6th Oct 23
Revision Received:	5th Dec 23
Editorial Decision:	12th Dec 23
Revision Received:	15th Dec 23
Accepted:	15th Dec 23

Editor: Lise Roth

Transaction Report:

This manuscript was transferred to EMBO Molecular Medicine

Review
COMMONS

following peer review at Review Commons.

Review #1

1. Evidence, reproducibility and clarity:

Evidence, reproducibility and clarity (Required)

In this paper the authors report a direct correlation between PMP22 and PTEN expression levels in the nerve of CMT mutants. In CMT1A Pmp22tg rat nerves, PTEN levels are increased, whereas in Pmp22^{+/-} mutants, a model of the HNPP neuropathy, PTEN levels decrease. Consistent with this, Pmp22tg nerves display lower Akt phosphorylation and, viceversa, Pmp22^{+/-} nerves have higher Akt phosphorylation. The authors lowered PTEN in the transgenic and inhibited mTOR using Rapamycin in the Pmp22^{+/-} to support the functional relevance of the PMP22-PTEN correlation.

I have major concerns on the data as shown, which, in my opinion, don't support the main conclusion of this paper. In more detail:

Figure 1

Panel a: the decrease of Pten expression should be quantified with at least n=3 taking into account the variability among different samples at the different time points indicated (the same applies in panel b, even if here the increase of Pten expression level in Pmp22tg nerves is more evident)

Panel a and b: the statement that Pten is more expressed at P18 at the peak of myelination in wildtype nerves is not supported by the blots as shown

Figure S1, page 4: what does it mean "in line with this finding we were unable to detect protein-protein...". Maybe the authors meant: since there is a direct correlation between Pmp22 and Pten expression levels in the mutants, the authors explored the possibility of an interaction between the two. Regarding the co-IPs, in panel a, the co-IP at the endogenous level, the immunoprecipitation efficiency of PMP22 is very low. Maybe a pull-down experiment using either exogenous purified PMP22 or PTEN and nerve lysates can help to rule out the possibility of an interaction. The experiments in b, c are performed in overexpression in a heterologous system (293 cells).

Panel e: maybe with this experiment the authors aim to suggest that Pten and Pmp22 are unlikely to interact directly or indirectly since Pten is cytosolic and Pmp22 myelin-membrane enriched. However, this myelin purification shows that Pmp22 at P0 expression levels are also abundant in the cytosol, maybe also because P18 has been chosen as time point. What about a different type of membrane-cytosol

fractionation experiment and/or another time point?

Page 4: "Taken together, Pmp22 dosage inversely correlates with the abundance of PTEN...": please revise this statement

Additional experiments are needed to support the conclusion of Figure 1 that, in the two mutants, Pten levels reversely correlate with PI3K-Akt-mTOR pathway activation, which represents the rationale of all further experiments. For example, it should be shown systematically in both mutants both Akt and ERK phosphorylation levels (Akt at both T308 and S473), and mTOR activity read outs. In the previously published paper (Fledrich et al.) only increased Akt phosphorylation in Pmp22^{+/-} nerves was reported, whereas Pmp22^{tg} analysis was focused on the interdependence between Akt and ERK without exploring mTOR activation, which is relevant here.

Figure 2

The aberrant myelin figures displayed are similar to myelin ovoids preceding degeneration rather than myelin outfoldings. It is also strange that these alterations are in the wildtype cultures treated with RAPA, that instead, in this system, has been reported to increase myelination as it improves protein homeostasis (autophagy, quality control, etc). Also for this experiment, pulse treatment may be beneficial rather than in continuous.

Is Akt-mTOR phosphorylation-signaling increased also in Pmp22^{+/-} co-cultures as in mutant nerves? Is the treatment reducing the overactivation?

Figure 3

The RAPA treatment seems to increase Pten level in the mutant even above wildtype levels (panel b), which can result in decreased myelin thickness due to downregulation of Akt-mTOR. A different method to normalize expression levels should be used.

Panel c-e: aberrant fibers should be normalized on total number of fibers and on the area, particularly because RAPA is used. Can these data also be reproduced in quadriceps nerves as tomacula are more prominent in these Pmp22^{+/-} nerves showing less variability due to the prevalence of large caliber axons?

Figure 4

A different model, the C61 mouse a Pmp22^{tg} overexpressing PMP22 is used here (rather than the CMT1A rat). This should be explained in the results. Is also this model characterized by increased Pten levels in the nerve? And low Akt-mTOR activation for instance?

The improvement in the number of myelin segments following PTEN inhibition in Pmp22tg co-cultures is very weak.. The 500 nM has instead a consistent effect in reducing myelin segments in the wildtype and I think that these results overall don't support the conclusion that myelination is ameliorated by reducing PTEN activity in Pmp22tg co-cultures.

Similarly to Figure 2, is PTEN level increased in Pmp22tg cultures along with Akt-mTOR downregulation?

Figure 5

As for Figure 4, the use of the mouse transgenic instead of the CMT1A rat should be specified and PTEN, Akt-mTOR expression/activation levels should be checked biochemically also in this model. And quantified (panel c).

In panel d overactivation of mTOR (PS6 staining) in Schwann cells is not evident.

Panel e: co-cultures are established using ex vivo Dhh-Cre recombination. The downregulation of Pten in the cultures should be documented. Pten Fl/+ Dhh-Cre cultures seem to have axonal fasciculation.

Figure 6

G-ratio analysis: which are the mean values (numbers) with SEM in the three groups analyzed wildtype, Pmp22tg and Pmp22tg; Pten fl/+; Dhh-Cre? How is Akt-mTOR signaling in the double mutant as compared to Pmp22tg? Is that increased at P18? If more fibers are committed to myelinate in the double mutant as compared to the single Pmp22tg at P18 ,particularly, it is unclear why there is no difference in differentiation marker expression in Figure 7 (Oct6 and Hmgcr).

2. Significance:

Significance (Required)

In conclusion, the correlation between PMP22 and PTEN is a potential interesting observation. However, in my opinion, experiments as shown don't support the conclusion that PMP22 controls PTEN expression level and activity, which is suggested at the basis of the pathogenesis of PMP22 dosage-related neuropathies.

3. How much time do you estimate the authors will need to complete the suggested revisions:

Estimated time to Complete Revisions (Required)

(Decision Recommendation)

More than 6 months

Yes

Review #2

1. Evidence, reproducibility and clarity:

Evidence, reproducibility and clarity (Required)

This study investigates the modulation, both genetically and pharmacologically, of the PI3K/Akt/mTOR signaling in preclinical animal models for the inherited peripheral neuropathies HNPP and CMT1A. These conditions result from a gene dosage abnormality of the peripheral myelin protein gene PMP22. The exact biological molecular mechanisms remain enigmatic despite it having been over 30 years since the major genetic lesions, the CMT1A duplication and HNPP deletion, were described. With respect to myelin biology one observes focally slowed nerve conduction at pressure palsies and local/segmental hypermyelination in HNPP whereas hypomyelination occurs in CMT1A.

The study is nicely conducted, data illustrations very informative, and writing clear and concise. This paper will likely be of great interest to your readers. A few things the authors may want to consider:

1. Regarding in the Introduction: "...the molecular mechanisms causative for the abnormal myelination remain largely unknown and still no therapy is available." Suggest consider modifying to perhaps: '...no small molecule or pharmacological

- therapeutic intervention exist.' To say "no therapy" exist is 'myopic' and untrue.
2. Suggest adding question mark to end of sentence or changing 'asked' to "investigated" for following thought: "Here, we asked whether PI3K/Akt/mTOR signaling provides therefore a therapeutic target to treat the consequences of altered Pmp22 gene-dosage."
 3. Rather than attempt to establish PRIORITY perhaps 'softening' the INTRODUCTION concluding statement "Our results thus identify a potential pharmacological target for this inherited neuropathy.
 4. [This makes thePI3K/Akt/mTOR pathway a promising target for a preventive treatment of affected nerves also in human patients.] Does this belong in RESULTS? Or rather DISCUSSION?

2. Significance:

Significance (Required)

The authors provide convincing evidence that the HNPP pathobiology is ameliorated by PI3K/Akt/mTOR inhibitors. Interestingly they found radial myelin growth was most affected by this approach and suggest an interesting transdermal approach in injured nerves in the acute prevention of pressure palsies.

3. How much time do you estimate the authors will need to complete the suggested revisions:

Estimated time to Complete Revisions (Required)

(Decision Recommendation)

Less than 1 month

No

Review #3

1. Evidence, reproducibility and clarity:

Evidence, reproducibility and clarity (Required)

In this paper Sareda and co-workers demonstrate that the PTEN/mTOR pathway is indirectly involved in regulating myelin thickness and wrapping in models of altered PMP22 gene dosage both in vitro and in vivo. Inhibition of this pathway decreases myelin thickness in models of HNPP, while increasing myelin thickness in models of CMT1A. The evidence for these conclusions is complex but reasonably presented, and the conclusions mainly supported by the data. The abstract for this paper, however, presents a somewhat oversimplified conclusion that the PTEN pathway mainly modifies models of HNPP, where the paper clearly demonstrates that models of CMT1A are also affected by this same pathway. This should be clarified.

2. Significance:

Significance (Required)

These data are significant, since they would provide new targets for treating inherited neuropathy associated with altered PLP22 gene dosage.

3. How much time do you estimate the authors will need to complete the suggested revisions:

Estimated time to Complete Revisions (Required)

(Decision Recommendation)

Less than 1 month

4. Review Commons values the work of reviewers and encourages them to get credit for their work. Select 'Yes'

below to register your reviewing activity at Web of Science Reviewer Recognition Service (formerly Publons); note that the content of your review will not be visible on Web of Science.

Yes

Revision Plan

Manuscript number: RC-2023-01910

Corresponding author(s): Michael W. Sereda

1. General Statements

Reviewer #1:

In this paper the authors report a direct correlation between PMP22 and PTEN expression levels in the nerve of CMT mutants. In CMT1A Pmp22tg rat nerves, PTEN levels are increased, whereas in Pmp22+/- mutants, a model of the HNPP neuropathy, PTEN levels decrease. Consistent with this, Pmp22tg nerves display lower Akt phosphorylation and, vice versa, Pmp22+/- nerves have higher Akt phosphorylation. The authors lowered PTEN in the transgenic and inhibited mTOR using Rapamycin in the Pmp22+/- to support the functional relevance of the PMP22-PTEN correlation. ... In conclusion, the correlation between PMP22 and PTEN is a potential interesting observation. However, in my opinion, experiments as shown don't support the conclusion that PMP22 controls PTEN expression level and activity, which is suggested at the basis of the pathogenesis of PMP22 dosage-related neuropathies.

We thank Reviewer #1 for this detailed feedback. We appreciate the Reviewer's assessment that our observation that PMP22 and PTEN *are correlated* in CMT1A and HNPP is of potential interest. In the revised manuscript we addressed this key point by adding additional quantifications (**Figure 1a, d; Figure 5d**) and novel Western Blot analyses (**Figure 1a, d**). Regarding the *pathophysiological significance* of the correlation, we point out that both the original as well as the partially revised manuscript contain multiple pieces of evidence demonstrating that altered PTEN activity is critical for both PMP22 gene-dosage related neuropathies:

- i) The inhibition of the PI3K/PTEN/AKT/mTOR axis upstream (LY294002) or downstream (Rapamycin) of decreased PTEN ameliorates myelin defects in an *in vitro* HNPP model (**Figure 2b, c**).
- ii) Downstream of PTEN, Rapamycin treatment ameliorates myelin defects, motor behavior and electrophysiology in the HNPP mouse model *in vivo* (**Figure 3c, d, e, g, i**)
- iii) Targeting of increased PTEN directly by inhibiting its activity pharmacologically (VO-OHpic) in a CMT1A rat model or by depleting it genetically in a CMT1A model leads to ameliorated myelination *in vitro* (**Figure 4b, c; Figure 5f, g**).
- iv) The genetic depletion of PTEN in a CMT1A mouse model increases myelination *in vivo*, albeit not in the long term (**Figure 6a, b, c, d**).

We therefore feel that any additional evidence to show that "PMP22 controls PTEN activity" is not vital for supporting the major claims of the manuscript, i.e. that the observed correlation of PTEN levels with PMP22 gene dosage has relevance for the etiology of PMP22 dosage diseases and

Revision Plan

and that targeting the PI3K-PTEN-AKT-mTOR axis downstream of PTEN provides a potential pharmacological therapy of HNPP (while directly targeting PTEN ultimately fails to rescue CMT1A). However, we agree that the activity of PTEN on the molecular level is interesting, and such evidence would further strengthen our conclusions. Therefore, in the final revised version, we plan to add further Western Blots and explore possible downstream effects of altered PTEN levels.

Reviewer #2:

This study investigates the modulation, both genetically and pharmacologically, of the PI3K/Akt/mTOR signaling in preclinical animal models for the inherited peripheral neuropathies HNPP and CMT1A. These conditions result from a gene dosage abnormality of the peripheral myelin protein gene PMP22. The exact biological molecular mechanisms remain enigmatic despite it having been over 30 years since the major genetic lesions, the CMT1A duplication and HNPP deletion, were described. With respect to myelin biology one observes focally slowed nerve conduction at pressure palsies and local/segmental hypermyelination in HNPP whereas hypomyelination occurs in CMT1A. The study is nicely conducted, data illustrations very informative, and writing clear and concise. This paper will likely be of great interest to your readers. The authors provide convincing evidence that the HNPP pathobiology is ameliorated by PI3K/Akt/mTOR inhibitors. Interestingly they found radial myelin growth was most affected by this approach and suggest an interesting transdermal approach in injured nerves in the acute prevention of pressure palsies.

We thank Reviewer #2 for this positive evaluation.

Reviewer #3:

In this paper Sareda and co-workers demonstrate that the PTEN/mTOR pathway is indirectly involved in regulating myelin thickness and wrapping in models of altered PMP22 gene dosage both in vitro and in vivo. Inhibition of this pathway decreases myelin thickness in models of HNPP, while increasing myelin thickness in models of CMT1A. The evidence for these conclusions is complex but reasonably presented, and the conclusions mainly supported by the data. The abstract for this paper, however, presents a somewhat oversimplified conclusion that the PTEN pathway mainly modifies models of HNPP, where the paper clearly demonstrates that models of CMT1A are also affected by this same pathway. This should be clarified.

We thank Reviewer #3 for the feedback on the manuscript. We agree with the Reviewer that the same pathway (PI3K/Akt/mTOR) also affects CMT1A, but it is of importance for us to highlight that the disease mechanisms are -at least partly- different between HNPP and CMT1A. This is supported by our observation that PTEN reduction in CMT1A only transiently improves myelination *in vivo* (**Figure 6**) and the persistent alteration of differentiation markers despite PTEN reduction, which is not observed in HNPP (**Figure 7**).

2. Description of the planned revisions

Reviewer #1

Regarding the activity of PTEN

Figure 1

- 1) *Additional experiments are needed to support the conclusion of Figure 1 that, in the two mutants, Pten levels reversely correlate with PI3K-Akt-mTOR pathway activation, which represents the rationale of all further experiments. For example, it should be shown systematically in both mutants both Akt and ERK phosphorylation levels (Akt at both T308 and S473), and mTOR activity read outs. In the previously published paper (Fledrich et al.) only increased Akt phosphorylation in Pmp22+/- nerves was reported, whereas Pmp22tg analysis was focused on the interdependence between Akt and ERK without exploring mTOR activation, which is relevant here. 2) (Figure 4) A different model, the C61 mouse a Pmp22tg overexpressing PMP22 is used here (rather than the CMT1A rat). This should be explained in the results. Is also this model characterized by increased Pten levels in the nerve? And low Akt-mTOR activation for instance? 3) (Figure 5) How is Akt-mTOR signaling in the double mutant as compared to Pmp22tg? Is that increased at P18?*

Response: We fully agree with the Reviewer that further exploration of PTEN downstream effects will add value to the manuscript. We already justified the usage of the C61 mouse model more clearly, added P-S6 staining of wildtype in addition to an improved representation in **Figure 5e**, and performed extra Western Blot analysis of PTEN expression (described in the next section “*Incorporated revisions*”). Moreover, we will further evaluate the downstream signaling components of PTEN and will perform additional Western Blot analyses of peripheral nerves of HNPP mice, CMT1A rats as well as C61 and C61xPTENhKO mice.

Figure S1

- 2) *Figure S1, page 4: what does it mean "in line with this finding we were unable to detect protein-protein...". May be the authors meant: since there is a direct correlation between Pmp22 and Pten expression levels in the mutants, the authors explored the possibility of an interaction between the two. Regarding the co-IPs, in panel a, the co-IP at the endogenous level, the immunoprecipitation efficiency of PMP22 is very low. May be a pull-down experiment using either exogenous purified PMP22 or PTEN and nerve lysates can help to rule out the possibility of an interaction. The experiments in b, c are performed in overexpression in a heterologous system (293 cells).*

Response: We agree with the Reviewer that we might have missed a possible interaction between PMP22 and PTEN in the experiments performed so far. Indeed, pull-down experiments may prove helpful to rule out / reveal protein-protein interaction. Therefore, we will use purified PMP22 and perform pull-down experiments using nerve lysates of wildtype and CMT1A rats.

Figure 5

3) *Pten* *Fl/+ Dhh-Cre* cultures seem to have axonal fasciculation.

Response: We thank the Reviewer for this observation. We will systematically inspect all recorded images for features of fasciculation. We will also assess whether fasciculation is a representative feature in cultures derived from any of the genotypes, and if so, whether the genotypes differ in this regard.

3. Description of the revisions that have already been incorporated in the transferred manuscript

Changes in the text are highlighted in **green** in the revised manuscript

Reviewer #1:

Figure 1

1) *Panel a: the decrease of Pten expression should be quantified with at least n=3 taking into account the variability among different samples at the different time points indicated (the same applies in panel b, even if here the increase of Pten expression level in Pmp22tg nerves is more evident).*

Response: We agree with the Reviewer that the timeline is not sufficient to demonstrate alteration in PTEN expression in PMP22 gene dosage diseases CMT1A and HNPP. Therefore, we performed new Western Blot experiments evaluating PTEN expression in (i) HNPP mice, (ii) CMT1A rat (iii) C61 mice and (iv) C61xPTENhKO mice with minimum n = 3 biological replicates and performed the respective quantification which is shown in **Figure 1 (i, ii)** and **Figure 5 (iii, iv)**. The results of the Western Blot analysis and quantification show an increase in PTEN abundance in CMT1A rat (**Figure 1d**) and C61 mice (**Figure 5d**) while a decrease is observed in HNPP mice (**Figure 1a**) and PTENhKOxC61 mice (**Figure 5d**) when compared to wildtype controls.

2) *Panel a and b: the statement that Pten is more expressed at P18 at the peak of myelination in wildtype nerves is not supported by the blots as shown.*

Response: We agree that this observation is only partly supported by the Western Blot analysis, as seen in the HNPP mouse model, and deleted this part in the results section.

3) *Figure S1, page 4: what does it mean "in line with this finding we were unable to detect protein-protein...". May be the authors meant: since there is a direct correlation between Pmp22 and Pten expression levels in the mutants, the authors explored the possibility of an interaction between the two.*

Response: We thank the Reviewer for pointing out the lack of clarity here. We changed the respective sentence accordingly:

“Since there is a direct correlation between PMP22 and PTEN expression levels in the mutants, we explored the possibility of an interaction between the proteins. By immunoprecipitation experiments we were unable to detect protein-protein interaction between PMP22 and PTEN (**Figure S1**).” (Page 4)

- 4) *Page 4: "Taken together, Pmp22 dosage inversely correlates with the abundance of PTEN...": please revise this statement*

Response: We thank the reviewer for spotting this mistake. We changed the sentence accordingly, which now reads:

“Taken together, Pmp22 dosage directly correlates with the abundance of PTEN and presumably the activation level of the PI3K/Akt/mTOR pathway in myelinating Schwann cells (**Figure 1i**).” (Page 4, Line 23)

Figure 2:

- 1) *The aberrant myelin figures displayed are similar to myelin ovoids preceding degeneration rather than myelin outfoldings. It is also strange that these alterations are in the wildtype cultures treated with RAPA, that instead, in this system, has been reported to increase myelination as it improves protein homeostasis (autophagy, quality control, etc).*

Response: We thank the Reviewer for pointing this out. Indeed, in the way the images have been presented the aberrant myelin profiles can be mistaken for ovoids. However, a close inspection of the TUJ1 channel images revealed continuity of the axons below the aberrant myelin, thereby excluding ovoid formation. In the partially revised manuscript, we now also show the TUJ1 channel individually (**Figure 2**), so that it can be appreciated that the defects are confined to the myelin. Concerning the incidence of the myelin defects in RAPA treated wildtype cultures, our analysis can have missed a potential amelioration due to the rather high variability in the data.

Figure 3

Panel c-e: aberrant fibers should be normalized on total number of fibers and on the area, particularly because RAPA is used.

Response: We agree with the Reviewer that number of tomacula and recurrent loops should be normalized to the total number of fibers on the area. We have quantified the total number of fibers in the whole sciatic nerve and normalized the tomacula and recurrent loops number accordingly. Results show a decrease in both tomacula and recurrent loops after Rapamycin treatment in the HNPP mice (**Figure 3c, d, e, f**).

Figure 4

The improvement in the number of myelin segments following PTEN inhibition in Pmp22tg co-cultures is very weak. The 500 nM has instead a consistent effect in reducing myelin segments in the wildtype and I think that these results overall don't support the conclusion that myelination is ameliorated by reducing PTEN activity in Pmp22tg co-cultures.

Response: We thank the Reviewer for this important point. We like to emphasize that we treated whole cultures with the PTEN inhibitor and we cannot rule out a (probably) negative effect on axonal PTEN, resulting in only weak improvement of myelination in PMP22tg cultures and strong effects also on the wildtype co-cultures. Therefore, we decided against a treatment of CMT1A models *in vivo* and further explored the effects of PTEN reduction specifically in Schwann cells using the genetic model as described **Figure 5**. The Reviewer made clear to us that this is inappropriately explained in the results section and we therefore adapted this in the manuscript on page 6:

“Similarly, the prolonged inhibition of PTEN with VO-OHpic (for 14 days) caused a dosage-dependent reduction in myelinated segments in wildtype co-cultures (**Figure 4c, Figure S2**). The mechanism is currently unexplained but cannot rule out a negative effect of PTEN inhibition on DRG neurons and myelination.”

Figure 5:

4) *A different model, the C61 mouse a Pmp22tg overexpressing PMP22 is used here (rather than the CMT1A rat). This should be explained in the results. Is also this model characterized by increased Pten levels in the nerve? And low Akt-mTOR activation for instance?*

Response: We agree with the Reviewer that it has not been clear in the text why we changed here to the C61 mouse model. We clarified this in the Results section which now reads on page 6:

“To reduce *Pten* function in CMT1A models also *in vivo*, we applied a genetic approach (**Figure 5a**). As the genetic tools to specifically target Schwann cells were only available in the mouse and not the rat, we used the C61 mouse model of CMT1A. We reduced PTEN by about 50% selectively in CMT1A Schwann cells by crossbreeding *Pmp22* transgenic mice with floxed *Pten* and *Dhh-cre* mice, yielding *PTEN^{fl/+}Dhh^{cre/+}PMP22^{tg}* experimental mutants (**Figure 5b**). Western blot analyses of sciatic nerve lysates confirmed the increase of PTEN in *PMP22^{tg}* mice and the reduction of PTEN in the double mutants (**Figure 5c, d**).”

Moreover, regarding the PTEN expression we added Western Blot analysis and quantification in **Figure 5c, d** showing increased PTEN expression in the C61 mouse model of CMT1A and decreased PTEN in the PTENhKOxC61 double mutants. Further analysis of the downstream signaling is planned (see “*planned revision*”).

- 5) *PTEN, Akt-mTOR expression/activation levels should be checked biochemically also in this model. And quantified (panel c).*

Response: We added an explanation for the use of the C61 mouse model (see point Figure 5.1 above). Moreover, we quantified the Western Blot analysis and added it in **Figure 5d**. The expression of PTEN was included in the Western Blot analysis (**Figure 5c**) showing increased PTEN expression also in the C61 mouse model. Further biochemical analysis of the C61 mouse model is planned (see “planned revision”).

- 6) *In panel d overactivation of mTOR (PS6 staining) in Schwann cells is not evident.*

Response: We agree with the Reviewer that the way the image was displayed is not sufficient to show P-S6 activation in the double mutants. We have now split the image (**Figure 5e**) to better visualize the P-S6 staining alone compared to the co-staining with P0 (marker for compact myelin) and DAPI (nuclei). Further, we added staining of wildtype nerve. We hope this way the differences in P-S6 activation can be easier appreciated.

Figure 6:

G-ratio analysis: which are the mean values (numbers) with SEM in the three groups analyzed wildtype, Pmp22tg and Pmp22tg; Pten fl/+; Dhh-Cre?

Response: We thank the Reviewer for pointing this out. We added the quantification of the mean g-ratios in **Figure 6d, f**.

Figure 7:

- 1) *If more fibers are committed to myelinate in the double mutant as compared to the single Pmp22tg at P18, particularly, it is unclear why there is no difference in differentiation marker expression in Figure 7 (Oct6 and Hmgcr).*

Response: We thank the reviewer for this comment. We do not necessarily expect to see a strong difference in the expression of differentiation markers given the mild increase in myelination in the double mutants. Similarly, we do not observe alterations in the expression of differentiation markers in HNPP mice, while these fibers produce more myelin. Therefore, we concluded that alterations in PTEN-PI3K/Akt/mTOR signaling do not influence differentiation in the mouse models while in the PMP22 overexpressing situation of CMT1A other mechanisms alter differentiation of the Schwann cells. We also note that experiments were performed at postnatal day 18 and we cannot rule out possible alterations in differentiation marker expression at earlier time points in development in the double mutants.

- 2) *In conclusion, the correlation between PMP22 and PTEN is a potential interesting observation. However, in my opinion, experiments as shown don't support the conclusion*

that PMP22 controls PTEN expression level and activity, which is suggested at the basis of the pathogenesis of PMP22 dosage-related neuropathies.

Response: Please also see section 1. In order to avoid any overstatement that "*PMP22 controls PTEN expression level and activity*", in our revised version we have clarified this point and changed the wording in the main text:

"The mechanisms that link the abundance of PMP22 to that of PTEN are still unclear and we here neither show direct nor indirect control of PTEN expression by PMP22." (Page 8)

Reviewer #2:

1. *Regarding in the Introduction: "...the molecular mechanisms causative for the abnormal myelination remain largely unknown and still no therapy is available." Suggest consider modifying to perhaps: '...no small molecule or pharmacological therapeutic intervention exist.' To say "no therapy" exist is 'myopic' and untrue.*
2. *Suggest adding question mark to end of sentence or changing 'asked' to "investigated" for following thought: "Here, we asked whether PI3K/Akt/mTOR signaling provides therefore a therapeutic target to treat the consequences of altered Pmp22 gene-dosage."*
3. *Rather than attempt to establish PRIORITY perhaps 'softening' the INTRODUCTION concluding statement "Our results thus identify a potential pharmacological target for this inherited neuropathy.*
4. *[This makes thePI3K/Akt/mTOR pathway a promising target for a preventive treatment of affected nerves also in human patients.] Does this belong in RESULTS? Or rather DISCUSSION?*

Response: We thank the Reviewer for the suggestions. We changed the sentences accordingly in the manuscript (1.: Page 3, Line 23; 2.: Page 3, Line 26; highlighted in green). Regarding point 3, we are convinced that identifying pharmacological targets for peripheral neuropathies should be given priority. Indeed, the aspect concerning point 4 is already highlighted in the discussion therefore we removed the sentence from the result section.

Reviewer #3:

The abstract for this paper, however, presents a somewhat oversimplified conclusion that the PTEN pathway mainly modifies models of HNPP, where the paper clearly demonstrates that models of CMT1A are also affected by this same pathway. This should be clarified.

We agree with the Reviewer that the same pathway (PI3K/Akt/mTOR) also affects CMT1A, but it is of importance for us to highlight that the disease mechanisms are -at least partly- different

between HNPP and CMT1A. This is supported by our observation that PTEN reduction in CMT1A only transiently improves myelination *in vivo* (**Figure 6**) and the persistent alteration of differentiation markers despite PTEN reduction, which is not observed in HNPP (**Figure 7**). For clarification we have altered the wording in the abstract which now reads: "In contrast, we found that CMT1A pathogenesis was only transiently ameliorated by altered PI3K/Akt/mTOR signaling, which drives radial but not longitudinal growth of peripheral myelin sheaths".

3. Description of analyses that authors prefer not to carry out

Reviewer #1:

Figure 1:

Figure 1, Panel e: may be with this experiment the authors aim to suggest that Pten and Pmp22 are unlikely to interact directly or indirectly since Pten is cytosolic and Pmp22 myelin-membrane enriched. However, this myelin purification shows that Pmp22 as P0 expression levels are also abundant in the cytosol, may be also because P18 has been chosen as time point. What about a different type of membrane-cytosol fractionation experiment and/or another time point?

Response: We want to clarify that in this experiment not myelin and cytosol fractions were separated but myelin and whole sciatic nerve lysate (which is the input before isolation of the myelin fraction, called "lysate"). Therefore, the analysis aimed at showing an enrichment of PMP22 and P0 in the myelin fraction while PTEN and TUJ (as a control) are not, which makes it more unlikely for PTEN and PMP22 to interact directly. This experiment, together with the immunohistochemical analysis in **Figure 1h** should highlight the location of PMP22 and PTEN in the Schwann cell. Together with the newly suggested experiments of the Reviewer for **Figure S1** (see planned Revision point 1) we do not see the need for extra membrane-cytosol fractionations and/ or another timepoint as the more detailed as the improved experiment on protein-protein interaction using nerve lysate (not only cell culture) is the experiment of choice to clarify whether we have a direct interaction or not.

Regarding in vitro Schwann cell- DRG co-culture experiments:
(Figure 2, Figure 4 and Figure 5e)

1. *(Figure 2) For this experiment, pulse treatment may be beneficial rather than in continuous. Is Akt-mTOR phosphorylation-signaling increased also in Pmp22+/- co-cultures as in mutant nerves? Is the treatment reducing the overactivation?*
2. *(Figure 4) Similarly to Figure 2, is PTEN level increased in Pmp22tg cultures along with Akt-mTOR downregulation?*
3. *(Figure 5) Panel e: co-cultures are established using ex vivo Dhh-Cre recombination. The downregulation of Pten in the cultures should be documented.*

Response: We agree with Reviewer #1 that a deeper analysis of the co-culture system regarding the downstream signaling of PTEN would increase the value of the experiments. Unfortunately, the experiments were designed in a very small scale with the intention of only evaluating myelin alterations on a histological level and we did not have enough tissue to collect cells for deeper protein expression analysis. Moreover, we tried to use the co-culture system as a proof-of-principle experiment in parallel to our in vivo studies which we value more important due to the still quite artificial co-culture setup. We hope that the Reviewer can understand our approach with the focus we set on the in vivo work.

Figure 3:

1. *The RAPA treatment seems to increase Pten level in the mutant even above wildtype levels (panel b), which can result in decreased myelin thickness due to downregulation of Akt-mTOR. A different method to normalize expression levels should be used.*

Response: Comparing the mean, relative expression levels resulting from our quantification as plotted in the graph (panel b) revealed no increase above wildtype level after Rapamycin treatment in the HNPP mouse. Further, we decided for whole protein staining as the superior approach to loading control because we have observed alterations in the expression of other frequently used “housekeepers” such as GAPDH, Actin and Vinculin in the CMT1A rodent models.

2. *Panel c-e: Can these data also be reproduced in quadriceps nerves as tomacula are more prominent in these Pmp22^{+/-} nerves showing less variability due to the prevalence of large caliber axons?*

Response: Unfortunately, quadriceps nerves were not collected for histology in the experiment and therefore we cannot redo the quantification. Nevertheless, we agree that the quadriceps nerves have less variability than the sciatic nerve and will definitely include the tissue in our future experiments.

25th May 2023

Dear Dr. Sereda,

Thank you for the submission of your manuscript to our editorial offices. I have now had the opportunity to read it, together with the referees' reports and your rebuttal letter, and to discuss them with the other members of our editorial team.

We agree that the study fits the scope of the journal, and we appreciate that you are willing to address most of the points raised by the reviewers. We thus encourage you to submit a revised version of your manuscript, including the modifications and revisions described in your point-by-point letter. Acceptance of the manuscript will entail a second round of review. EMBO Molecular Medicine encourages a single round of revision only and therefore, acceptance or rejection of the manuscript will depend on the completeness of your responses included in the next, final version of the manuscript. For this reason, and to save you from any frustrations in the end, I would strongly advise against returning an incomplete revision.

When submitting your revised manuscript, please carefully review the instructions that follow below. Failure to include requested items will delay the evaluation of your revision:

We require:

4) A .docx formatted letter INCLUDING the reviewers' reports and your detailed point-by-point responses to their comments. As part of the EMBO Press transparent editorial process, the point-by-point response is part of the Review Process File (RPF), which will be published alongside your paper.

5) A complete author checklist, which you can download from our author guidelines (<https://www.embopress.org/page/journal/17574684/authorguide#submissionofrevisions>). Please insert information in the checklist that is also reflected in the manuscript. The completed author checklist will also be part of the RPF.

6) Please note that all corresponding authors are required to supply an ORCID ID for their name upon submission of a revised manuscript.

7) It is mandatory to include a 'Data Availability' section after the Materials and Methods. Before submitting your revision, primary datasets produced in this study need to be deposited in an appropriate public database, and the accession numbers and database listed under 'Data Availability'. Please remember to provide a reviewer password if the datasets are not yet public (see <https://www.embopress.org/page/journal/17574684/authorguide#dataavailability>).

8) For data quantification: please specify the name of the statistical test used to generate error bars and P values, the number (n) of independent experiments (specify technical or biological replicates) underlying each data point and the test used to calculate p-values in each figure legend. The figure legends should contain a basic description of n, P and the test applied. Graphs must include a description of the bars and the error bars (s.d., s.e.m.). Please provide exact p values.

- the medical issue you are addressing,

- the results obtained and

- their clinical impact.

13) Author contributions: CRedit has replaced the traditional author contributions section because it offers a systematic machine readable author contributions format that allows for more effective research assessment. Please remove the Authors Contributions from the manuscript and use the free text boxes beneath each contributing author's name in our system to add specific details on the author's contribution. More information is available in our guide to authors.

16) As part of the EMBO Publications transparent editorial process initiative (see our Editorial at <http://embomolmed.embopress.org/content/2/9/329>), EMBO Molecular Medicine will publish online a Review Process File (RPF) to accompany accepted manuscripts.

In the event of acceptance, this file will be published in conjunction with your paper and will include the anonymous referee reports, your point-by-point response and all pertinent correspondence relating to the manuscript. Let us know whether you agree with the publication of the RPF and as here, if you want to remove or not any figures from it prior to publication.

I look forward to receiving your revised manuscript.

Yours sincerely,

Lise Roth

Rev_Com_number: RC-2023-01910
New_manu_number: EMM-2023-18047
Corr_author: Sereda
Title: Targeting PI3K/Akt/mTOR signaling in rodent models of PMP22 gene-dosage diseases

Rev_Com_number: RC-2023-01910

New_manu_number: EMM-2023-18047

Corr_author: Sereda

Title: Targeting PI3K/Akt/mTOR signaling in rodent models of PMP22 gene-dosage diseases

In our response to the reviewers, we adhered to the outline of our revision plan that accompanied our initial submission to EMBO Mol Med including new experiments that were performed since then.

1. General Statements

Reviewer #1:

In this paper the authors report a direct correlation between PMP22 and PTEN expression levels in the nerve of CMT mutants. In CMT1A Pmp22tg rat nerves, PTEN levels are increased, whereas in Pmp22+/- mutants, a model of the HNPP neuropathy, PTEN levels decrease. Consistent with this, Pmp22tg nerves display lower Akt phosphorylation and, vice versa, Pmp22+/- nerves have higher Akt phosphorylation. The authors lowered PTEN in the transgenic and inhibited mTOR using Rapamycin in the Pmp22+/- to support the functional relevance of the PMP22-PTEN correlation. ... In conclusion, the correlation between PMP22 and PTEN is a potential interesting observation. However, in my opinion, experiments as shown don't support the conclusion that PMP22 controls PTEN expression level and activity, which is suggested at the basis of the pathogenesis of PMP22 dosage-related neuropathies.

Response: We thank Reviewer #1 for this detailed feedback. We appreciate the Reviewer's assessment that our observation that PMP22 and PTEN *are correlated* in CMT1A and HNPP is of potential interest. Regarding the *pathophysiological significance* of the correlation, we point out that both the previous manuscript versions already contained multiple pieces of evidence demonstrating that altered PTEN activity is critical for both PMP22 gene-dosage related neuropathies:

- i) The inhibition of the PI3K/PTEN/AKT/mTOR axis upstream (LY294002) or downstream (Rapamycin) of decreased PTEN ameliorates myelin defects in an *in vitro* HNPP model (**Figure 2b, c**).
- ii) Downstream of PTEN, Rapamycin treatment ameliorates myelin defects, motor behavior and electrophysiology in the HNPP mouse model *in vivo* (**Figure 3c, d, e, f, g, i**).
- iii) Targeting of increased PTEN directly by inhibiting its activity pharmacologically (VO-OHpic) in a CMT1A rat model or by depleting it genetically in a CMT1A mouse model leads to ameliorated myelination *in vitro* (**Figure 4b, c; Figure 5h, i**).
- iv) The genetic depletion of PTEN in a CMT1A mouse model increases myelination *in vivo*, albeit not in the long term (**Figure 6a, b, c, d**).

We therefore felt that any additional evidence to show that "PMP22 controls PTEN activity" is not vital for supporting the major claims of the manuscript, i.e. that the observed correlation of PTEN levels with PMP22 gene dosage has relevance for the etiology of PMP22 dosage diseases and that targeting the PI3K-PTEN-AKT-mTOR axis downstream of PTEN provides a potential pharmacological therapy of HNPP (while directly targeting PTEN ultimately fails to rescue CMT1A).

However, we agree that the activity of PTEN on the molecular level is interesting, and such evidence would further strengthen our conclusions. Therefore, in this revised version, we performed multiple new Western Blots (see point-by-point section) showing that:

- i) The activities of canonical downstream effectors of mTOR were altered in all examined PMP22 dosage-related neuropathy models: increased at decreased PTEN levels in HNPP (**Figure 1b**); decreased at increased PTEN levels in the rat (**Figure 1d**) and the mouse model (**Figure 5e**) of CMT1A.
- ii) Rapamycin treatment normalized mTOR activity in the HNPP mouse model (**Figure 3b**).
- iii) The genetic depletion of PTEN in the CMT1A mouse model resulted in a trend towards normalized mTOR activity (**Figure 5e, f**).

We thank the reviewer again for the valuable input, as we indeed feel that these new Western Blot results, together with an improved presentation of the immunofluorescent stainings (now **Figure 5g**) on the mechanistic impact of altered PTEN levels in both HNPP and CMT1A added significant value to the manuscript.

Reviewer #2:

This study investigates the modulation, both genetically and pharmacologically, of the PI3K/Akt/mTOR signaling in preclinical animal models for the inherited peripheral neuropathies HNPP and CMT1A. These conditions result from a gene dosage abnormality of the peripheral myelin protein gene PMP22. The exact biological molecular mechanisms remain enigmatic despite it having been over 30 years since the major genetic lesions, the CMT1A duplication and HNPP deletion, were described. With respect to myelin biology one observes focally slowed nerve conduction at pressure palsies and local/segmental hypermyelination in HNPP whereas hypomyelination occurs in CMT1A. The study is nicely conducted, data illustrations very informative, and writing clear and concise. This paper will likely be of great interest to your readers. The authors provide convincing evidence that the HNPP pathobiology is ameliorated by PI3K/Akt/mTOR inhibitors. Interestingly they found radial myelin growth was most affected by this approach and suggest an interesting transdermal approach in injured nerves in the acute prevention of pressure palsies.

Response: We thank Reviewer #2 for this positive evaluation.

Reviewer #3:

In this paper Sareda and co-workers demonstrate that the PTEN/mTOR pathway is indirectly involved in regulating myelin thickness and wrapping in models of altered PMP22 gene dosage both in vitro and in vivo. Inhibition of this pathway decreases myelin thickness in models of HNPP, while increasing myelin thickness in models of CMT1A. The evidence for these conclusions is complex but reasonably presented, and the conclusions mainly supported by the data. The abstract for this paper, however, presents a somewhat oversimplified conclusion that the PTEN pathway mainly modifies models of HNPP, where the paper clearly demonstrates that models of CMT1A are also affected by this same pathway. This should be clarified.

Response: We thank Reviewer #3 for the feedback on the manuscript. We agree with the Reviewer that the same pathway (PI3K/Akt/mTOR) also affects CMT1A, but it is of importance for us to highlight that the disease mechanisms are -at least partly- different between HNPP and CMT1A. This is supported by our observation that PTEN reduction in CMT1A only transiently improves myelination *in vivo* (**Figure 6**) and the persistent alteration of differentiation markers despite PTEN reduction, which is not observed in HNPP (**Figure 7**).

2. Point-by-point

Changes in the text are highlighted in **green** (Revision#1 for Review Commons) and **pink** (Revision#2, now) in the final revised manuscript.

Reviewer #1

- 1) (Figure 1) Panel a: the decrease of Pten expression should be quantified with at least n=3 taking into account the variability among different samples at the different time points indicated (the same applies in panel b, even if here the increase of Pten expression level in Pmp22tg nerves is more evident).

Response: We agree with the Reviewer that the timeline is not sufficient to demonstrate alteration in PTEN expression in PMP22 gene dosage diseases CMT1A and HNPP. Therefore, we performed new Western Blot experiments evaluating PTEN expression in HNPP and CMT1A with n = 4 biological replicates and performed the respective quantification which are shown in **Figure 1a** and **Figure 1c**. The results of the Western Blot analysis and quantification show an increase in PTEN abundance in CMT1A rat (**Figure 1c**) while a decrease is observed in HNPP mice (**Figure 1a**) when compared to wildtype controls.

- 2) (Figure 1) Panel a and b: the statement that Pten is more expressed at P18 at the peak of myelination in wildtype nerves is not supported by the blots as shown.

Response: We agree that this observation is only partly supported by the Western Blot analysis, as seen in the HNPP mouse model, and deleted this part in the results section.

- 3) (Figure S1) page 4: what does it mean "in line with this finding we were unable to detect protein-protein...". May be the authors meant: since there is a direct correlation between Pmp22 and Pten expression levels in the mutants, the authors explored the possibility of an interaction between the two.

Response: We thank the Reviewer for pointing out the lack of clarity here. We changed the respective sentence accordingly on page 4:

"Since there is a direct correlation between PMP22 and PTEN expression levels in the mutants, we explored the possibility of an interaction between the proteins. By immunoprecipitation experiments we were unable to detect protein-protein interaction between PMP22 and PTEN (**Figure S1**)."

- 4) (Figure S1) Regarding the co-IPs, in panel a, the co-IP at the endogenous level, the immunoprecipitation efficiency of PMP22 is very low. May be a pull-down experiment using either exogenous purified PMP22 or PTEN and nerve lysates can help to rule out the possibility of an interaction. The experiments in b, c are performed in overexpression in a heterologous system (293 cells).

Response: We thank the Reviewer for this helpful suggestion. We used highly purified PMP22 to perform a stringent pull-down from sciatic nerves of wildtype and CMT1A rats at p18. However, as shown in the new panel b in Figure S1 (now Figure EV1), we were not able to detect interacting PTEN, while P0, a published interaction partner of PMP22, was found specifically in the PMP22 elution fractions. In the manuscript, the corresponding passages now read in the results section (page 4):

"Furthermore, we performed pull-down experiments using purified PMP22. While we were able to confirm interaction of PMP22 with P0 (D'Urso et al., 1999, Hasse et al., 2004), we could not detect PTEN in PMP22 eluates (Figure EV1b), making a direct, high-affinity interaction unlikely."

and the discussion section (page 9):

"Further, we have not been able to pull down PTEN from sciatic nerve lysate with purified PMP22 (Figure EV1b)."

- 5) (Figure 1) Panel e: may be with this experiment the authors aim to suggest that Pten and Pmp22 are unlikely to interact directly or indirectly since Pten is cytosolic and Pmp22 myelin-membrane enriched. However, this myelin purification shows that Pmp22 as P0 expression levels are also abundant in the cytosol, may be also because P18 has been chosen as time point. What about a different type of membrane-cytosol fractionation experiment and/or another time point?

Response: We want to clarify that in this experiment not myelin and cytosol fractions were separated but myelin and whole sciatic nerve lysate (which is the input before isolation of the myelin fraction, called "lysate"). Therefore, the analysis aimed at showing an enrichment of PMP22 and P0 in the myelin fraction while PTEN and TUJ (as a control) are not, makes it more unlikely for PTEN and PMP22 to interact directly. This experiment, together with the immunohistochemical analysis in **Figure 1h** should highlight the location of PMP22 and PTEN in the Schwann cell. Together with the newly suggested experiments of the Reviewer for **Figure S1** (now **Figure EV1**, see point 4) we do not see the need for extra membrane-cytosol fractionations and/ or another timepoint as the improved experiment on protein-protein interaction using nerve lysate (not only cell culture) is the experiment of choice to clarify whether we have a direct interaction or not.

- 6) *(Figure 1) Page 4: "Taken together, Pmp22 dosage inversely correlates with the abundance of PTEN...": please revise this statement*

Response: We thank the reviewer for spotting this mistake. We changed the sentence accordingly, which now reads on page 4:

"Taken together, Pmp22 dosage directly correlates with the abundance of PTEN and correlates inversely with the activation level of the PI3K/Akt/mTOR pathway (**Figure 1b, d**) (18), presumably in myelinating Schwann cells (**Figure 1i**)."

- 7) *(Figure 1) Additional experiments are needed to support the conclusion of Figure 1 that, in the two mutants, Pten levels reversely correlate with PI3K-Akt-mTOR pathway activation, which represents the rationale of all further experiments. For example, it should be shown systematically in both mutants both Akt and ERK phosphorylation levels (Akt at both T308 and S473), and mTOR activity read outs. In the previously published paper (Fledrich et al.) only increased Akt phosphorylation in Pmp22+/- nerves was reported, whereas Pmp22tg analysis was focused on the interdependence between Akt and ERK without exploring mTOR activation, which is relevant here.*

Response: We agree with the Reviewer that a deeper analysis of the downstream pathway beyond Akt phosphorylation (as shown by Fledrich et al. 2014) is vital to support our conclusion. Therefore we performed Western Blots analyzing the activation of ribosomal protein S6 and its kinase (S6K) which are downstream of mTOR signaling. We observed an increase in S6 phosphorylation in sciatic nerve lysates of HNPP mice early in development at postnatal day 6 (**Figure 1a**) and a decrease in S6K phosphorylation in CMT1A sciatic nerve lysates at postnatal day 6 (**Figure 1c**), strongly supporting our conclusion that PTEN levels reversely correlate with PI3K-Akt-mTOR pathway activation.

- 8) *(Figure 2) The aberrant myelin figures displayed are similar to myelin ovoids preceding degeneration rather than myelin outfoldings. It is also strange that these alterations are in the wildtype cultures treated with RAPA, that instead, in this system, has been reported to increase myelination as it improves protein homeostasis (autophagy, quality control, etc).*

Response: We thank the Reviewer for pointing this out. Indeed, in the way the images have been presented the aberrant myelin profiles can be mistaken for ovoids. However, a close inspection of the TUJ1 channel images revealed continuity of the axons below the aberrant myelin, thereby excluding ovoid formation. In the revised manuscript, we now show the TUJ1 channel individually (**Figure 2**), clarifying that the defects are confined to myelin. Regarding the frequency of the myelin defects in RAPA treated wildtype cultures, our analysis may have missed a potential amelioration due to the rather high variability in the data.

- 9) *(Figure 2) For this experiment, pulse treatment may be beneficial rather than in continuous. Is Akt-mTOR phosphorylation-signaling increased also in Pmp22+/- co-cultures as in mutant nerves? Is the treatment reducing the overactivation?*

Response: We agree with Reviewer #1 that a deeper analysis of the co-culture system regarding the downstream signaling of PTEN would increase the value of the experiments. Unfortunately, the experiments were designed in a small scale with the intention of only evaluating myelin alterations on a histological level and we did not have enough tissue to collect cells for deeper protein expression analysis. Moreover, we like to point out that we merely performed the co-culture system as a proof-of-principle experiment for our *in vivo* studies where we have addressed the question of Akt/mTOR signaling in PMP22+/- mouse mutants.

- 10) *(Figure 3) The RAPA treatment seems to increase Pten level in the mutant even above wildtype levels (panel b), which can result in decreased myelin thickness due to downregulation of Akt-mTOR. A different method to normalize expression levels should be used.*

Response: We decided for whole protein staining as the superior approach to loading control because we have observed alterations in the expression of other frequently used “housekeepers” such as GAPDH, Actin and Vinculin in the CMT1A rodent models. When comparing the mean, relative expression levels resulting from our quantification plotted in Figure 3b, we observed no increase above wildtype level after Rapamycin treatment in HNPP mice.

- 11) *(Figure 3) Panel c-e: Can these data also be reproduced in quadriceps nerves as tomacula are more prominent in these Pmp22+/- nerves showing less variability due to the prevalence of large caliber axons?*

Response: Unfortunately, quadriceps nerves were not collected for histology in the experiment and therefore we cannot redo the quantification. Nevertheless, we agree that the quadriceps nerves have less variability than the sciatic nerve and will definitely include the tissue in our future experiments.

- 12) *(Figure 3) Panel c-e: aberrant fibers should be normalized on total number of fibers and on the area, particularly because RAPA is used.*

Response: We agree with the Reviewer that number of tomacula and recurrent loops should be normalized to the total number of fibers on the area. We have quantified the total number of fibers in the whole sciatic nerve and normalized the tomacula and recurrent loops number accordingly. Results show a decrease in both tomacula and recurrent loops after Rapamycin treatment in the HNPP mice (**Figure 3c, d, e, f**).

13) *(Figure 4) The improvement in the number of myelin segments following PTEN inhibition in Pmp22tg co-cultures is very weak. The 500 nM has instead a consistent effect in reducing myelin segments in the wildtype and I think that these results overall don't support the conclusion that myelination is ameliorated by reducing PTEN activity in Pmp22tg co-cultures.*

Response: We thank the Reviewer for this important point. We like to emphasize that we treated whole cultures with the PTEN inhibitor and we cannot rule out a (probably) negative effect on axonal PTEN, resulting in only weak improvement of myelination in PMP22tg cultures and strong effects also on the wildtype co-cultures. Therefore, we decided against a treatment of CMT1A models *in vivo* and further explored the effects of PTEN reduction specifically in Schwann cells using the genetic model as described in **Figure 5**. The Reviewer made clear to us that this is inappropriately explained in the results section and we therefore adapted this in the results section page 6:

“Similarly, the prolonged inhibition of PTEN with VO-OHPic (for 14 days) caused a dosage-dependent reduction in myelinated segments in wildtype co-cultures (**Figure 4c, Figure S2**). The mechanism is currently unexplained but cannot rule out a negative effect of PTEN inhibition on DRG neurons and myelination.”

14) *(Figure 4) Similarly to Figure 2, is PTEN level increased in Pmp22tg cultures along with Akt-mTOR downregulation?*

Response: Please see point 9.

15) *(Figure 5) A different model, the C61 mouse a Pmp22tg overexpressing PMP22 is used here (rather than the CMT1A rat). This should be explained in the results. Is also this model characterized by increased Pten levels in the nerve? And low Akt-mTOR activation for instance? PTEN, Akt-mTOR expression/activation levels should be checked biochemically also in this model. And quantified (panel c).*

Response: We agree with the Reviewer that it has not been clear in the text why here we used the C61 mouse model. We clarified this in the Results section which now reads on page 6:

“To reduce *Pten* function in CMT1A models also *in vivo*, we applied a genetic approach (**Figure 5a**). As the genetic tools to specifically target Schwann cells were only available in the mouse and not the rat, we used the C61 mouse model of CMT1A. We reduced PTEN by about 50% selectively in CMT1A Schwann cells by crossbreeding *Pmp22* transgenic mice

with floxed *Pten* and *Dhh-cre* mice, yielding *PTEN^{fl/+}Dhh^{cre/+}PMP22^{tg}* experimental mutants (**Figure 5b**). Western blot analyses of sciatic nerve lysates confirmed the increase of PTEN in *PMP22^{tg}* mice and the reduction of PTEN in the double mutants (**Figure 5c, d**)."

Moreover, regarding the PTEN expression and downstream signaling of Akt/mTOR we added Western Blot analysis and quantification in **Figure 5c, d** showing increased PTEN expression in the C61 mouse model of CMT1A and decreased PTEN in the PTENhKOxC61 double mutants and in **Figure 5e, f** showing a decrease in S6 phosphorylation in nerves of the C61 mouse, as well as a trend towards normalization in the PTENhKOxC61 double mutants.

16) (Figure 5) In panel d overactivation of mTOR (PS6 staining) in Schwann cells is not evident.

Response: We agree with the Reviewer that the way the image was displayed is not sufficient to show P-S6 activation in the double mutants. For clarification, we have now split the image (now **Figure 5g**) to better visualize the P-S6 staining alone compared to the co-staining with P0 (marker for compact myelin) and DAPI (nuclei). Further, we added staining of wildtype nerve to easier visualize the differences in P-S6 activation.

17) (Figure 5) Panel e: co-cultures are established using ex vivo *Dhh-Cre* recombination. The downregulation of *Pten* in the cultures should be documented.

Response: See point 9)

18) (Figure 5) *Pten* *Fl/+* *Dhh-Cre* cultures seem to have axonal fasciculation.

Response: We thank the Reviewer for this observation. We systematically inspected all recorded images for features of fasciculation and observed that fasciculation is rarely found in cultures derived from any of the genotypes and the example image of *Pten^{fl/+}Dhh-cre* cultures was not well selected. We have now picked a better representative image in **Figure 5h**.

19) (Figure 5) How is Akt-mTOR signaling in the double mutant as compared to *Pmp22^{tg}*? Is that increased at P18?

Response: We added P-S6 staining of wildtype in addition to an improved representation in **Figure 5e**, and performed novel Western Blot analysis of S6 phosphorylation at postnatal day 18, showing a decrease in C61 mice and an increase in the PTENhxC61 double mutants (**Figure 5 e, f**) (see also point 15).

20) (Figure 6) G-ratio analysis: which are the mean values (numbers) with SEM in the three groups analyzed wildtype, *Pmp22^{tg}* and *Pmp22^{tg}; Pten fl/+; Dhh-Cre*?

Response: We thank the Reviewer for pointing this out. We added the quantification of the mean g-ratios in **Figure 6d, f**.

21) *(Figure 7) If more fibers are committed to myelinate in the double mutant as compared to the single Pmp22tg at P18, particularly, it is unclear why there is no difference in differentiation marker expression in Figure 7 (Oct6 and Hmgcr).*

Response: We thank the reviewer for this comment. We do not necessarily expect to see a strong difference in the expression of differentiation markers given the mild increase in myelination in the double mutants. Similarly, we do not observe alterations in the expression of differentiation markers in HNPP mice, while these fibers produce more myelin. Therefore, we concluded that alterations in PTEN-PI3K/Akt/mTOR signaling do not influence differentiation in the mouse models while in the PMP22 overexpressing situation of CMT1A other mechanisms alter differentiation of the Schwann cells. We also note that experiments were performed at postnatal day 18 and we cannot rule out possible alterations in differentiation marker expression at earlier time points in development in the double mutants.

22) *In conclusion, the correlation between PMP22 and PTEN is a potential interesting observation. However, in my opinion, experiments as shown don't support the conclusion that PMP22 controls PTEN expression level and activity, which is suggested at the basis of the pathogenesis of PMP22 dosage-related neuropathies.*

Response: In order to avoid any overstatement that "*PMP22 controls PTEN expression level and activity*", in our revised version we have clarified this point and changed the wording in the results section accordingly (page 8):

"The mechanisms that link the abundance of PMP22 to that of PTEN are still unclear and we here neither show direct nor indirect control of PTEN expression by PMP22."

With the addition of Western Blots for the activation of downstream Akt/ mTOR signaling and additional quantifications, we have now provided additional evidence that PTEN level inversely correlate with the activation of Akt/mTOR downstream signaling.

Reviewer #2:

1. *Regarding in the Introduction: "...the molecular mechanisms causative for the abnormal myelination remain largely unknown and still no therapy is available." Suggest consider modifying to perhaps: '...no small molecule or pharmacological therapeutic intervention exist.' To say "no therapy" exist is 'myopic' and untrue.*
2. *Suggest adding question mark to end of sentence or changing 'asked' to "investigated" for following thought: "Here, we asked whether PI3K/Akt/mTOR signaling provides therefore a therapeutic target to treat the consequences of altered Pmp22 gene-dosage."*

3. *Rather than attempt to establish PRIORITY perhaps 'softening' the INTRODUCTION concluding statement "Our results thus identify a potential pharmacological target for this inherited neuropathy.*
4. *[This makes the PI3K/Akt/mTOR pathway a promising target for a preventive treatment of affected nerves also in human patients.] Does this belong in RESULTS? Or rather DISCUSSION?*

Response: We thank the Reviewer for these valuable suggestions. We changed the sentences accordingly in the manuscript (1.: page 3; 2.: page 3; highlighted in green). Regarding point 3, we are convinced that identifying pharmacological targets for peripheral neuropathies should be given priority. Indeed, the aspect concerning point 4 is already highlighted in the discussion. Therefore we removed the sentence from the result section.

Reviewer #3:

The abstract for this paper, however, presents a somewhat oversimplified conclusion that the PTEN pathway mainly modifies models of HNPP, where the paper clearly demonstrates that models of CMT1A are also affected by this same pathway. This should be clarified.

We agree with the Reviewer that the same pathway (PI3K/Akt/mTOR) also affects CMT1A, but we like to highlight that the disease mechanisms are -at least partly- different between HNPP and CMT1A. This is supported by our observation that PTEN reduction in CMT1A only transiently improves myelination *in vivo* (**Figure 6**) and the persistent alteration of differentiation markers despite PTEN reduction, which is not observed in HNPP (**Figure 7**). For clarification we have altered the wording in the abstract which now reads: "In contrast, we found that CMT1A pathogenesis was only transiently ameliorated by altered PI3K/Akt/mTOR signaling, which drives radial but not longitudinal growth of peripheral myelin sheaths".

6th Oct 2023

Dear Prof. Sereda,

Thank you for the submission of your revised manuscript to EMBO Molecular Medicine, and please accept my apologies for the delay in getting back to you as I sought external advice on your manuscript.

Indeed, as you will see below, your manuscript was re-reviewed by referees #1 and #2 (who had initially reviewed your manuscript for Review Commons). While referee #2 is satisfied with the revisions and supportive of publication, referee #1 still raises major concerns on your work. I thus consulted an independent expert in the field, who provided further comments on the report from referee #1:

"1- PMP22 haploinsufficient mutant. Phosphorylation of S6 is shown only at P6 which is very early in myelination in postnatal development, and this is the data that supports mTORC1 activation. In the co-culture experiments shown in Fig. 2, the claimed myelin outfoldings are not corresponding to what is displayed. The altered myelin seems to be ovoids of degenerated segments that actually are also present in the WT Rapamycin-treated.

The in vivo rescue using Rapamycin shows only a mild amelioration of the phenotype, from 4.5% to 3%, and this percentage at 5 months of age is underestimated. In this mutant quadriceps motor nerves should be used, where tomacula are more abundant (up to 20% of alterations). Finally, the neurophysiological analysis reveals defects that are not typical of this mutant, particularly at that age and with this low abundance of myelin alterations that authors show (e.g. decreased MNCV and CMAP, please refer to the paper from Prof. Jun Li where conduction blocks typical of this neuropathy are reproduced).

Advisor:

I notice that the phospho-antibody used in Figure 1b and 1d, to measure the ribosomal protein S6/phosphor-S6(K) ratio, is different between mouse Pmp22^{+/-} (HNPP model) and the Pmp22^{tg} (CMT1A model) overexpressing rat. The quality of the antibodies (CST #5364 versus CST #9208 in materials & methods) seems to differ between mouse and rat, respectively. In Figures 1b and 1d, the authors showed the reduction of PTEN in 9W animals and below in the same panel, the effect of S6 or S6K phosphorylation respectively at postnatal day P6. It would be more logical to have a direct comparison in the Pmp22^{+/-} (Fig 1b) (or Pmp22^{tg} in Fig 1d), in the same panel, of the PTEN expression with the S6 (or S6K) phosphorylation at the same time point. The complete panel at 9W can go in the main figure, and the panel at P6 in supplementary.

It is not clear why the authors have tested the phosphorylation status of the ribosomal protein S6 in the HNPP model, while for the CMT1A model, they have analyzed the phosphorylation status of the p70 S6 kinase (S6K/ p70S6K1) to measure the activity of the PI3K/Akt/mTOR pathway. However, among the two analyzed targets, only S6K/p70S6K1 is known as one of the most characterized mTOR targets, but not S6. Why testing S6 phosphorylation for the HNPP model?

I agree that it is difficult to discriminate on the Figure 2 (= microscopy images) if these are myelin outfoldings or ovoids in the co-culture experiments. However, I tend to accept with the authors that these resemble tomaculae that are typical for an HNPP pathology (caused by a heterozygous PMP22 deletion). I do agree with the reviewer that these degenerating segments are also present in the wild-type rapamycin treated co-cultures (data shown are indicated as non-significant in Figure 2d).

I agree that there is a mild amelioration of the rapamycin treatment on the HNPP phenotype in the Pmp22^{+/-} animals (the p-values vary between one and three stars). In particular, in Figure 3j the Pmp22^{+/-} RAPA NCV is not ameliorated compared to the Pmp22^{+/-} Placebo. I am not sure if the authors can repeat these experiments for the mutant quadriceps motor nerves in the revised manuscript (instead measuring CMAP and motor NCV of the mouse sciatic nerve as mentioned in the materials & methods). The authors may refer to the paper of Jun Li (Bai Y, Zhang X, Katona I, Saporta MA, Shy ME, O'Malley HA, Isom LL, Suter U, Li J. *J Neurosci.* 2010 Jan 13;30(2):600-8. doi: 10.1523/JNEUROSCI.4264-09.2010.PMID: 20071523).

2- CMT1A, transgenic. Again, decreased activation of mTORC1 pathway is shown at P6 only in Figure 1, actually using a different antigen P6SK with non-specific bands. In the co-culture experiments in Figure 4 authors use a PTEN inhibitor that dramatically affects myelination also in the control culture. Such a toxic or non-specific effect makes the interpretation of results very difficult and can be misleading. For example, less myelin means reduced Schwann cell proliferation, increased apoptosis etc. The in vivo experiment, the Pten Schwann cell-specific ablation in the Pmp22 transgenic: the increase of myelinated axons is transient only at P18 and there is a concomitant increase of myelin thickness beyond normal value, as a gain of function due to overactivation of mTORC1. Schwann cell de-differentiation is not ameliorated at all, last experiment and Figure.

Advisor:

In Figure 5, the authors show the genetic approach to reduce PTEN expression in the CMT1A model (here in the mouse model). In Figure 5 c-d, PTEN is reduced in the double mutants in sciatic nerve lysate from 16-week-old mice. If the authors want to show the link between PTEN expression and the Akt/mTOR pathway, why in the western blot shown in Figure 5e they used lysates from 18-days old mice? It would be nice to see the PTEN expression in Figure 5e from the same animals.

Again, regarding the mTOR targets in the Pmp22^{tg} mouse model, Figure 5e-f reports the S6 phosphorylation status, as an effect of the genetic rescue. However, in Figure 1, where initially the Pmp22^{tg} rat model was characterized, the S6K phosphorylation was used. This gives the impression that the authors have used the protein marker S6 or S6K in an interchangeable way, but

they are not both directly linked to mTOR, they are in a cascade. I would rather be consistent and use the same markers as in Figure 1. Also in the main text, the description of the PI3K/Akt/mTOR pathway with their downstream targets should be more accurate to avoid confusion by the reader.

Indeed, the PTEN inhibitor Vanadium complex VO-OHpic seems to dramatically affect myelination also in the control culture shown in Figure 4. If this gives a toxic or non-specific effect, possibly leading to reduced Schwann cell proliferation or increased apoptosis, one could recommend the authors to document with a western blotting with apoptotic markers (as a supplementary figure).

I agree that the Schwann cell de-differentiation is not ameliorated. This is illustrated in Figure 7a/b. The p-values are not significant for the transcription factors Ngfr, Pou3fr1 and Sox2 (= RT-qPCR experiments). Note that Figure 7d is mentioned on page 8, but it is not shown. Figure 7d is not added in this revised manuscript; I assume Figure 7d was deleted during the revision of the manuscript. Figure 7c (= microscopy images of myelination in Pmp22+/-, wild type, Pmp22tg) is the last figure/experiment shown for g-ratio and axon diameter."

Based on the comments from this advisor, we would like to invite further revisions of the study, in which these points should be addressed. As EMBO Press usually encourages one single round of revisions, please be aware that this will be the last chance for you to address these issues.

The revised manuscript will once again be subjected to review, and we cannot guarantee a positive outcome at this stage.

Moreover, please address the following editorial requests:

1/ Manuscript text:

- Please address the queries from our data editors in the related "Data edited MS file". Please only keep in track changes mode any new modification.
- Acknowledgements: Please make sure that the information provided in the acknowledgements matches the information provided in the submission system.
- Disclosure statement and competing interests: We updated our journal's competing interests policy in January 2022 and request authors to consider both actual and perceived competing interests. Please review the policy <https://www.embopress.org/competing-interests> and update your competing interests if necessary. Please rename the section accordingly, and place it after the acknowledgements.
- References: please reformat to have references in alphabetical order, with 10 authors before et al.
- "The file includes" paragraph should be removed from the manuscript.
- "Supplemental information" should be renamed "Expanded View Figure Legends".

2/ Figures:

- We do not accept statistics and error bars when n=2. Please correct Figure 5i accordingly.
- Please clearly indicate in the figure legends of Figure 4B and EV2A if the same pictures were used.
- Figure 7d is referenced in the text, but no 7d panel can be found, please check.
- Tables EV1-EV2 can remain in the ms as Table 1 and 2 but they need to stay editable and black and white and need to be placed between the main and EV figure legends; otherwise, they should be removed from the manuscript and uploaded separately as Expanded View.

3/ Synopsis:

Please resize the visual abstract as a file 550 px wide x 300-600 px high.

4/ Please upload "The Paper Explained" as a separate file.

5/ As part of the EMBO Publications transparent editorial process initiative (see our Editorial at <http://embomolmed.embopress.org/content/2/9/329>), EMBO Molecular Medicine will publish online a Review Process File (RPF) to accompany accepted manuscripts.

In the event of acceptance, this file will be published in conjunction with your paper and will include the anonymous referee reports, your point-by-point response and all pertinent correspondence relating to the manuscript. Let us know whether you agree with the publication of the RPF and as here, if you want to remove or not any figures from it prior to publication. Please note that the Authors checklist will be published at the end of the RPF.

I look forward to receiving your revised manuscript.

Yours sincerely,

Lise Roth

***** Reviewer's comments *****

Referee #1 (Comments on Novelty/Model System for Author):

Models are adequate

Referee #1 (Remarks for Author):

Unfortunately, I still have serious concerns on conclusions made that are not adequately supported by the results/data as shown.

In detail:

1- PMP22 haploinsufficient mutant. Phosphorylation of S6 is shown only at P6 which is very early in myelination in postnatal development, and this is the data that supports mTORC1 activation. In the co-culture experiments shown in Fig. 2, the claimed myelin outfoldings are not corresponding to what is displayed. The altered myelin seems to be ovoids of degenerated segments that actually are also present in the WT Rapamycin-treated.

The in vivo rescue using Rapamycin shows only a mild amelioration of the phenotype, from 4.5% to 3%, and this percentage at 5 months of age is underestimated. In this mutant quadriceps motor nerves should be used, where tomacula are more abundant (up to 20% of alterations). Finally, the neurophysiological analysis reveals defects that are not typical of this mutant, particularly at that age and with this low abundance of myelin alterations that authors show (e.g. decreased MNCV and CMAP, please refer to the paper from Prof. Jun Li where conduction blocks typical of this neuropathy are reproduced).

2- CMT1A, transgenic. Again, decreased activation of mTORC1 pathway is shown at P6 only in Figure 1, actually using a different antigen P6SK with non-specific bands. In the co-culture experiments in Figure 4 authors use a PTEN inhibitor that dramatically affects myelination also in the control culture. Such a toxic or non-specific effect makes the interpretation of results very difficult and can be misleading. For example, less myelin means reduced Schwann cell proliferation, increased apoptosis etc.

The in vivo experiment, the Pten Schwann cell-specific ablation in the Pmp22 transgenic: the increase of myelinated axons is transient only at P18 and there is a concomitant increase of myelin thickness beyond normal value, as a gain of function due to overactivation of mTORC1. Schwann cell de-differentiation is not ameliorated at all, last experiment and Figure.

Referee #2 (Comments on Novelty/Model System for Author):

This revision and response to reviews is excellent.

Referee #2 (Remarks for Author):

Very thorough study and thoughtful responses to Reviewer(s) input.

Rev_Com_number: RC-2023-01910

New_manu_number: EMM-2023-18047-V2

Corr_author: Sereda

Title: Targeting PI3K/Akt/mTOR signaling in rodent models of PMP22 gene-dosage diseases

EMM-2023-18047-V2

Corr_author: Sereda

Title: Targeting PI3K/Akt/mTOR signaling in rodent models of PMP22 gene-dosage diseases

REVISION #2

We thank the Reviewer and the Advisor for their valuable comments. We have grouped the comments according to the associated figures and here respond to them together. All according **changes are highlighted in yellow** throughout the manuscript.

Point by Point Response

Reviewer 1, Advisor, Response

Figure 1:

Reviewer: "1- PMP22 haploinsufficient mutant. Phosphorylation of S6 is shown only at P6 which is very early in myelination in postnatal development, and this is the data that supports mTORC1 activation. CMT1A, transgenic. Again, decreased activation of mTORC1 pathway is shown at P6 only in Figure 1, actually using a different antigen P6SK with non-specific bands."

Advisor: "In Figures 1b and 1d, the authors showed the reduction of PTEN in 9W animals and below in the same panel, the effect of S6 or S6K phosphorylation respectively at postnatal day P6. It would be more logical to have a direct comparison in the Pmp22+/- (Fig 1b) (or Pmp22tg in Fig 1d), in the same panel, of the PTEN expression with the S6 (or S6K) phosphorylation at the same time point. The complete panel at 9W can go in the main figure, and the panel at P6 in supplementary. I notice that the phospho-antibody used in Figure 1b and 1d, to measure the ribosomal protein S6/phosphor-S6(K) ratio, is different between mouse Pmp22+/- (HNPP model) and the Pmp22tg (CMT1A model) overexpressing rat. The quality of the antibodies (CST #5364 versus CST #9208 in materials & methods) seems to differ between mouse and rat, respectively. It is not clear why the authors have tested the phosphorylation status of the ribosomal protein S6 in the HNPP model, while for the CMT1A model, they have analyzed the phosphorylation status of the p70 S6 kinase (S6K/ p70S6K1) to measure the activity of the PI3K/Akt/mTOR pathway. However, among the two analyzed targets, only S6K/p70S6K1 is known as one of the most characterized mTOR targets, but not S6. Why testing S6 phosphorylation for the HNPP model?"

Response: P6 is indeed very early in development of PNS myelination, which is why we chose this timepoint to assess mTOR signaling. The PI3K/Akt/mTOR pathway drives early myelination downstream of Neuregulin 1 type III (Maurel and Salzer, 2000; Michailov et al., 2004; Ogata et al., 2004), and the activation level of mTOR around the onset of myelination (i.e. P6) is critical (reviewed by Figlia et al., 2017). Moreover, we have previously shown that Akt phosphorylation is highest at P6 in wildtype rats and strongly reduced in the CMT rat model, while the abundances and differences are less prominent in later timepoints (Fledrich et al., Nat Medicine 2014, Fig. 1g).

We now emphasize the significance of early PI3K/Akt/mTOR signaling in the revised version of the manuscript (Introduction, page 2 line 21, page 3 line 6, line 8; Results, page 4 line 12, page 5 line 11).

To clarify that the onset of disease is visible already early in myelination, we added histological images and quantification of disease hallmarks in the sciatic nerve of PMP22^{+/-} mice at P6 in the new supplementary figure EV1c. The CMT rat was already described in detail at this early timepoint (Fledrich et al., 2014).

We agree with the Advisor that it is more logical to show cause (PTEN abundance) and effect (mTOR activation) at the same timepoint, and in light of the particular importance of early timepoints we decided to additionally assess PTEN expression at P6. We changed Figure 1 accordingly and

- added Western Blot analysis of PTEN at P6 showing a decrease in the HNPP mouse model PMP22^{+/-} (Figure 1b, upper panel), in line with the increased phosphorylation of S6 (Figure 1b, lower panel)
- added Western Blot analysis of PTEN in CMT rat at P6 (Figure 1d, upper panel), in line with the decreased phosphorylation of S6K (Figure 1d, lower panel).
- moved Western Blot analysis of PTEN at later timepoints (p18/p21 and 9 weeks) to the new Figure EV1a, b.

We also agree with the Advisor that of the two phospho-antigens we used to measure mTOR activity, only S6K is a direct target of mTOR. However, phosphorylation of S6, which is a direct target of S6K, also seems an appropriate measure of mTOR activity. In fact, it is even among the most frequently used mTOR activity readouts. We checked the recent literature (since 2019) via Google Scholar using the key phrase “mTOR activity” and found in the first 10 resulting, original research articles (when sorted by “relevance”) that

- 7 used phosphorylation of S6 to measure activity of mTOR (PMIDs 35271573, 31527800, 33150381, 34988408, 33794244, 35326535, 31105702)
- two used phosphorylation of S6K (PMIDs 32807902, 32686662)
- one used another readout (PMID 31874162).

We conclude that phosphorylation of S6 is a *bona fide* measure of activity of mTOR. When using the rat tissue for assessing S6 phosphorylation we had technical problems and therefore preferred the comparably well-established phosphorylation of S6K. Moreover, we added S6K phosphorylation next to only S6 phosphorylation in Figure 5c for the CMT1A mouse model to underline the feasibility of both antigens as a readout for the pathway activation (details see point 5).

Figure 2:

Reviewer: “In the co-culture experiments shown in Fig. 2, the claimed myelin outfoldings are not corresponding to what is displayed. The altered myelin seems to be ovoids of degenerated segments that actually are also present in the WT Rapamycin-treated.”

Advisor: “I agree that it is difficult to discriminate on the Figure 2 (= microscopy images) if these are myelin outfoldings or ovoids in the co-culture experiments. However, I tend to accept with the authors that these resemble tomaculae that are typical for an HNPP pathology (caused by a heterozygous PMP22 deletion). I do agree with the reviewer that these degenerating segments are also present in the wild-type rapamycin treated co-cultures (data shown are indicated as non-significant in Figure 2d).”

Response: We are convinced that the observed structures of excess myelin in our Schwann cell DRG co-cultures resemble tomacula in the artificial *in vitro* system. Other forms of CMT, i.e. due to mutations in *Mtmr2* cause a similar histological phenotype as that observed in PMP22^{+/-} mice (Bolino et al., 2000). Similar to our *in vitro* experiments, Bolino et al. (2016, *EMBO Mol Med*)

showed the existence of myelin outfoldings in *Mttr2*^{-/-} Schwann-cell DRG co-cultures. Regarding the concerns of the Reviewer and Advisor as to the Rapamycin treated wildtype cultures, we agree that the example image does not display the outcome of the analysis. We therefore selected another, better representative image in Figure 2b. Nevertheless, minor fractions (below 10 %) of segments in DMSO, rapamycin and LY294002 treated wildtype cultures also display outfoldings, however we found no significant differences among them.

Figure 3:

Reviewer: “The *in vivo* rescue using Rapamycin shows only a mild amelioration of the phenotype, from 4.5% to 3%, and this percentage at 5 months of age is underestimated. In this mutant quadriceps motor nerves should be used, where tomacula are more abundant (up to 20% of alterations). Finally, the neurophysiological analysis reveals defects that are not typical of this mutant, particularly at that age and with this low abundance of myelin alterations that authors show (e.g. decreased MNCV and CMAP, please refer to the paper from Prof. Jun Li where conduction blocks typical of this neuropathy are reproduced).”

Advisor: “I agree that there is a mild amelioration of the rapamycin treatment on the HNPP phenotype in the *Pmp22*^{+/-} animals (the p-values vary between one and three stars). In particular, in Figure 3j the *Pmp22*^{+/-} RAPA NCV is not ameliorated compared to the *Pmp22*^{+/-} Placebo. I am not sure if the authors can repeat these experiments for the mutant quadriceps motor nerves in the revised manuscript (instead measuring CMAP and motor NCV of the mouse sciatic nerve as mentioned in the materials & methods). The authors may refer to the paper of Jun Li (Bai Y, Zhang X, Katona I, Saporta MA, Shy ME, O'Malley HA, Isom LL, Suter U, Li J. *J Neurosci.* 2010 Jan 13;30(2):600-8. doi: 10.1523/JNEUROSCI.4264-09.2010.PMID: 20071523).”

Response: We agree that the observed incidence of tomacula in the sciatic nerve is relatively low. Nevertheless, we note that the observed reduction from 4.5 % to 3 % tomacula is an improvement of ~30 %, and in addition to tomacula reduction, we see a 40 % reduction of recurrent loops (1.7 % to 0.9 %). Others (Bolino *et al.* 2016, *EMBO Mol Med*) have also used the sciatic nerve as a readout for their studies in *PMP22*^{+/-} mice and reported 8 % of axons showing tomacula at P45, without separating tomacula and recurrent loops. Furthermore, other experimental readouts such as electrophysiology and Western Blot analysis were also performed from the sciatic nerve in our and other studies (Goebbels *et al.* 2012, Adlkofer *et al.* 1995 and 1997). While we do agree that the quadriceps nerve serves as a better tissue for performing the histological quantifications, as it was done in the first HNPP publications (Adlkofer *et al.* 1995 and 1997) as well as in other translational CMT studies (Klein *et al.* 2015, Groh *et al.* 2012), we are therefore convinced that our analysis of sciatic nerves is valid and sufficient to support our claims. Indeed, additional analysis of quadriceps nerves would have enriched the manuscript, but unfortunately, these nerves were not prepared during the study and for time and for ethical reasons we cannot redo the study. In Figure EV1c and d we added histological analyses of both sciatic and quadriceps motor nerves at young ages (P6/P18) to illustrate that while sciatic nerve may not at be the ideal tissue, it displays significant numbers of HNPP hallmarks early on. We also mention this in the text, now stating on page 5:

“Histologically, sciatic nerves from *Pmp22*^{+/-} mice displayed the disease-typical formation of tomacula, i.e. areas of focal hypermyelination, recurrent loops, and myelin infoldings, already starting at an early stage of myelination (Figure 3c, EV1c).”

The Reviewer raises concerns that the neurophysiological defects observed are not typical regarding the age and low abundance of myelin alterations.

In the original publications, electrophysiology was performed at:

- **10 weeks** of age (Adlkofer *et al.*, 1995), showing no alterations in CMAP or NCV of facial and sciatic nerve recordings in PMP22^{+/-} mutants
- **12-14 month** of age (Adlkofer *et al.*, 1997) showing significant decreased CMAP amplitudes in PMP22^{+/-} mutants and only a tendency in NCV measurements from 40±14 in PMP22^{+/+} control mice versus 32±7 m/s in PMP22^{+/-} mice
- **18 months** of age (Zielasek & Toyka, 1999) reporting a significant decrease in both CMAP and NCV in PMP22^{+/-} mice

Our analysis was done in mice at the age of **6.5 months**, meaning a timepoint between the early 10 weeks (no changes observed) and the 12 months (reduced CMAP, only NCV tendency). We observed reduced CMAP amplitudes at this timepoint, which are in line with the previous publications. Regarding conduction velocities, we agree with the Reviewer that the reduced NCV was not observed in early timepoints in the previous publications, only tendencies at 12 months were reported (Adlkofer *et al.*, 1997) and significant reductions at 18 month (Zielasek & Toyka 1999). The numerical difference of our recordings (reduction of 7 m/s in PMP22^{+/-} nerves) is similar to that reported in the Adlkofer 1997 study (reduction of 8 m/s), but the standard deviation of our recordings is lower (+1.42 m/s in WT Placebo and +4.4 m/s in PMP22^{+/-} Placebo) compared to the published recordings from 1997 (+14 m/s in WT and +/-7 m/s in PMP22^{+/-}). Nevertheless, we do not observe an improvement in the NCV after treatment.

Both, the Reviewer and the Advisor mention the Paper from Jun Li (Bai *et al.*, 2010). This paper established a model aiming to resemble the human phenotype of pressure palsies in HNPP. In fact, when we evaluated this model we experienced technical difficulties when applying weights to the nerves and therefore performed recordings without prior induction of pressure palsies.

The Advisor raises the question whether we can repeat the electrophysiological measurements in the quadriceps motor nerve. Unfortunately, electrophysiology cannot be performed in the quadriceps motor nerve, and the studies mentioned above (Adlkofer 1995, 1997, Zielasek & Toyka 1999) also performed the *in vivo* recordings in the sciatic nerve.

Figure 4:

Reviewer: “In the co-culture experiments in Figure 4 authors use a PTEN inhibitor that dramatically affects myelination also in the control culture. Such a toxic or non-specific effect makes the interpretation of results very difficult and can be misleading. For example, less myelin means reduced Schwann cell proliferation, increased apoptosis etc.”

Advisor: “Indeed, the PTEN inhibitor Vanadium complex VO-OHpic seems to dramatically affect myelination also in the control culture shown in Figure 4. If this gives a toxic or non-specific effect, possibly leading to reduced Schwann cell proliferation or increased apoptosis, one could recommend the authors to document with a western blotting with apoptotic markers (as a supplementary figure).”

Response: We agree with the Reviewer and Advisor that the PTEN inhibitor VO-OHpic shows toxic effects at high concentrations, and we discussed in the text (page 6 line 3) that VO-OHpic not only affects Schwann cells but is probably also toxic to neurons. It was because of the suspected VO-OHpic toxicity prevented us from testing VO-OHpic *in vivo*, but instead switch to the genetic model to specifically reduce PTEN only in Schwann cells and not axons. Nevertheless, despite some toxicity we did see that inhibiting PTEN is beneficial specifically in PMP22^{tg} cultures, indicating that increased PTEN is responsible for insufficient myelination in CMT1A. Therefore, we still think that the results shown in Figure 4 should be presented in the

manuscript, as they were instrumental to justify the switch to genetic targeting of PTEN only in Schwann cells *in vivo*. As to the Advisor's recommendation to investigate the toxic effect in the co-cultures, unfortunately, the scale of the cultures is very small and we could not keep tissue for further analysis e.g. Western Blots.

Figure 5:

Reviewer: "The *in vivo* experiment, the Pten Schwann cell-specific ablation in the Pmp22 transgenic: the increase of myelinated axons is transient only at P18 and there is a concomitant increase of myelin thickness beyond normal value, as a gain of function due to overactivation of mTORC1."

Advisor: "In Figure 5, the authors show the genetic approach to reduce PTEN expression in the CMT1A model (here in the mouse model). In Figure 5 c-d, PTEN is reduced in the double mutants in sciatic nerve lysate from 16-week-old mice. If the authors want to show the link between PTEN expression and the Akt/mTOR pathway, why in the western blot shown in Figure 5e they used lysates from 18-days old mice? It would be nice to see the PTEN expression in Figure 5e from the same animals. Again, regarding the mTOR targets in the Pmp22tg mouse model, Figure 5e-f reports the S6 phosphorylation status, as an effect of the genetic rescue. However, in Figure 1, where initially the Pmp22tg rat model was characterized, the S6K phosphorylation was used. This gives the impression that the authors have used the protein marker S6 or S6K in an interchangeable way, but they are not both directly linked to mTOR, they are in a cascade. I would rather be consistent and use the same markers as in Figure 1. Also in the main text, the description of the PI3K/Akt/mTOR pathway with their downstream targets should be more accurate to avoid confusion by the reader."

Response: We completely agree with the Advisor that it is more reasonable to analyze PTEN abundance and mTOR activation at the same timepoint. Therefore, we performed extra Western Blot analysis for PTEN at P18 (Figure 5c, d). Moreover, we also performed Western Blot analysis of PS6K in addition to PS6 and observed an increase of the downstream activation of PI3K/Akt/mTOR signaling when PTEN is genetically reduced in the CMT1A mouse model (Figure 5c, d). In the text we now state (page 6):

"As predicted, this manipulation caused activation of the downstream targets of Akt and mTOR, S6K and ribosomal protein S6, as visualized by protein abundance using Western Blot analysis (Figure 5c, d) and in immunostained peripheral nerve sections (Figure 5e)."

We further added a more detailed explanation of the downstream targets of the PI3K/Akt/mTOR pathway in the introduction to avoid confusion (page 2 line 36).

Figure 7

Reviewer: "Schwann cell de-differentiation is not ameliorated at all, last experiment and Figure."

Advisor: "I agree that the Schwann cell de-differentiation is not ameliorated. This is illustrated in Figure 7a/b. The p-values are not significant for the transcription factors Ngfr, Pou3fr1 and Sox2 (= RT-qPCR experiments). Note that Figure 7d is mentioned on page 8, but it is not shown. Figure 7d is not added in this revised manuscript; I assume Figure 7d was deleted during the revision of the manuscript. Figure 7c (= microscopy images of myelination in Pmp22^{+/-}, wild type, Pmp22tg) is the last figure/experiment shown for g-ratio and axon diameter."

Response: Schwann cell dedifferentiation is indeed not ameliorated in the Pmp22^{tg} model, and this is an important outcome of the experiment, explaining why we only see a transient improvement in myelin sheath thickness with PTEN reduction in CMT1A. Importantly, we only

observe alterations in dedifferentiation markers in CMT1A model and not in HNPP. Thus, dedifferentiation is a PMP22-overexpression, CMT1A but not HNPP, specific alteration. Previously we found that applying Nrg1 in CMT1A improved the overall phenotype and increased PI3K/AKT signaling, while at the same time it ameliorated dedifferentiation (Fledrich *et al.*, 2014). Therefore, we believed that this pathway is crucial in regulating myelin sheath thickness in CMT1A (PMP22 gene dosage diseases in general). The data here now let us conclude that activating the PI3K/AKT/mTOR pathway alone is not sufficient to improve myelination in CMT1A in the long term because it is uncoupled from dedifferentiation, which is the crucial defect in CMT1A. To highlight this important mechanistic insight—in addition to a previous focus in the discussion section—we changed the abstract, now stating:

“In contrast, we found Schwann cell dedifferentiation in CMT1A uncoupled from PI3K/Akt/mTOR, leaving partial PTEN ablation insufficient for disease amelioration.”

Furthermore, we reworked the section covering this aspect in the introduction (page 2 line 21) and now emphasize the crucial finding that we do not see amelioration of the disease phenotype by lowering PTEN levels in CMT1A in the final sentence of the introduction (page 3 line 13).

12th Dec 2023

Dear Prof. Sereda,

Thank you for the submission of your revised manuscript to EMBO Molecular Medicine. We have now received the feedback from referee #3, who had also advised us on the previous version of your manuscript, and as you will see below, he/she is satisfied with the revisions and supportive of publication.

We will thus be able to accept your manuscript once the following minor editorial points will be addressed:

1/ Manuscript text:

- Please remove the highlighted text and only keep in track changes mode any new modification.
- Please provide information for housing/husbandry of the animals.
- Please indicate whether the HEK293T were tested for mycoplasma contamination and adjust the checklist accordingly.

2/ Synopsis: I included minor modifications to your text, please let me know if you agree with the following or amend as you see fit:

Genetic and pharmacologic targeting of the PTEN-PI3K/AKT/mTOR/S6 signaling pathway was used as a potential therapeutic strategy in the peripheral neuropathies hereditary neuropathy with liability to pressure palsies (HNPP) and Charcot Marie Tooth disease type 1A (CMT1A).

- Abundance of PTEN as the major inhibitor of the PI3K/Akt/mTOR/S6 signaling pathway is decreased in HNPP and increased in CMT1A rodent models.
- Inhibiting mTOR downstream of PTEN improves the histological and behavioral disease phenotype in a mouse model of HNPP.
- Genetic targeting of PTEN in Schwann cells only transiently increased myelin growth in CMT1A model mice due to a differentiation defect, which is not present in HNPP nerves.

3/ As part of the EMBO Publications transparent editorial process initiative (see our Editorial at <http://embomolmed.embopress.org/content/2/9/329>), EMBO Molecular Medicine will publish online a Review Process File (RPF) to accompany accepted manuscripts.

This file will be published in conjunction with your paper and will include the anonymous referee reports, your point-by-point response and all pertinent correspondence relating to the manuscript.

Let us know whether you agree with the publication of the RPF and as here, if you want to remove or not any figures from it prior to publication.

I look forward to receiving your revised manuscript as soon as possible.

With kind regards,

Lise

***** Reviewer's comments *****

Referee #3 (Remarks for Author):

The authors have addressed the remaining points, raised by the reviewer and advisor, in this second revision of their manuscript. The authors have modified the text and figures accordingly and where relevant. I have no further comments.

All editorial and formatting issues were resolved by the authors.

15th Dec 2023

Dear Prof. Sereda,

Thank you for submitting your revised files. I am pleased to inform you that your manuscript is accepted for publication and is now being sent to our publisher to be included in the next available issue of EMBO Molecular Medicine!

If you have any questions, please do not hesitate to contact the Editorial Office.

Congratulations on your interesting work!

With kind regards,

Lise Roth

Rev_Com_number: RC-2023-01910

New_manu_number: EMM-2023-18047-V4

Corr_author: Sereda

Title: Targeting PI3K/Akt/mTOR signaling in rodent models of PMP22 gene-dosage diseases